# Dynamic changes in mitochondria support phenotypic flexibility of microglia

Katherine Espinoza[1,2,4], Ari W. Schaler[1,2,4], Daniel T. Gray[1], Arielle R. Sass[1], Adrian Escobar[1,2], Kamilia Moore[1], Megan E. Yu[1], Casandra G. Chamorro[1] & Lindsay M. De Biase[1,3] ✉

Microglial capacity to adapt to tissue needs is a hallmark feature of these cells. New studies show that mitochondria critically regulate the phenotypic adaptability of macrophages. To determine whether these organelles play similar roles in shaping microglial phenotypes, we generated transgenic mouse crosses to accurately visualize and manipulate microglial mitochondria. We find that brain-region differences in microglial attributes and responses to aging are accompanied by regional differences in mitochondrial mass and aging-associated mitochondrial remodeling. Microglial mitochondria are also altered within hours of LPS injections and microglial expression of inflammation-, trophic-, and phagocytosis-relevant genes is strongly correlated with expression of mitochondria-relevant genes. Finally, direct genetic manipulation of microglial mitochondria alters microglial morphology and leads to brain-region specific effects on microglial gene expression. Overall, this study advances our understanding of microglial mitochondria and supports the idea that mitochondria influence basal microglial phenotypes and phenotypic remodeling that takes place over hours to months.

An essential and defining characteristic of microglia is their ability to rapidly remodel their attributes and shift into a wide array of distinct functional states[1–3]. Historically, microglial cellular phenotypes were viewed in a binary fashion as either pro-inflammatory (M1, activated) or anti-inflammatory / tissue-reparative (M2, homeostatic)[4]. Work over the past 10-15 years has refined our understanding of microglial functions, showing that they can adopt a wide spectrum of transcriptional signatures, morphologies, and secretory/ proliferative/ phagocytic/ migratory profiles[5]. Moreover, the particular combination of microglial properties observed depends on brain region, age, and the presence or absence of chronic stress, high fat diet, and pollution exposure. Distinct tissue injury, infection, and disease processes also elicit unique sets of properties in responsive microglia[5–17], meaning that binary categorization of microglial functional states is far too simplistic and inaccurate.

As the field works to catalogue the spectrum of cellular phenotypes that microglia display in distinct circumstances, efforts have also been made to identify central or master regulators of microglial phenotypes. Key examples include transcription factors that turn on entire programs of microglial gene expression and associated downstream functional capabilities. Other examples include chromatin remodelers that can similarly enable or suppress expression of large numbers of genes that will shape downstream microglial attributes and functions[10,18]. Identifying such regulators of microglial phenotypes that have the capacity to modulate numerous microglial attributes is of high utility for the field. It can assist with defining and categorizing microglial functional states, as they can be determined on the basis of the central regulator rather than panels of 10s-100s of microglial genes, proteins, or functional readouts. These central regulators are also of great interest as potential therapeutic targets that could be

[1]Department of Physiology, David Geffen School of Medicine, UCLA, Los Angeles, USA. [2]UCLA Neuroscience Interdepartmental Graduate Program, Los Angeles, USA. [3]Department of Neurobiology, David Geffen School of Medicine, UCLA, Los Angeles, USA. [4]These authors contributed equally: Katherine Espinoza, Ari W. Schaler. ✉e-mail: ldebiase@mednet.ucla.edu

utilized to elicit broad changes in microglial function via coordinated regulation of multiple microglial attributes. Recent findings raise the possibility that mitochondria - as organelles - may have the capability to regulate microglial phenotypes in this fashion.

Classically thought of as the ATP producing powerhouse of the cell, mitochondria are now also recognized as critical signaling hubs[19–22]. Mitochondria can influence calcium-dependent intracellular signaling pathways via their ability to buffer intracellular calcium. They exert strong influence on signaling that involves protein post-translational modifications, as mitochondrial metabolic pathways generate many of the metabolites needed for these modifications. Mitochondria can influence the function of cellular endoplasmic reticulum and lysosomes via physical interactions with these organelles, and mitochondria can signal to the nucleus to impact gene expression[19–22]. Research in macrophages supports the idea that mitochondria act as signaling hubs, demonstrating that mitochondria regulate macrophage responses to pathology and ability to shift into distinct inflammatory states[23–29]. Key observations suggest a similarly important role for these organelles in regulating changes in microglial cellular phenotypes. Regional differences in microglial phenotypes are accompanied by prominent differences in expression of mitochondrial-relevant genes[30], and the state of mitochondria appears to be linked to microglial responses to early life immune challenge, brain injury, and protein aggregates in neurodegenerative disease[31–38]. Yet, direct and rigorous study of these organelles in microglia in vivo is minimal and gaps persist in our knowledge about potential mitochondrial regulation of microglial attributes.

To begin addressing this knowledge gap, we sought to analyze the relationship between mitochondria and key microglial attributes in both physiological and pathological contexts in mice. We used multiple transgenic crosses and multidisciplinary approaches to determine whether differences in microglial properties across brain regions during normative aging were accompanied by regional differences in microglial mitochondria. To explore the relationship between these organelles and microglial properties during CNS challenges, we chose pathological insults that differ in their modality (tissue injury vs inflammatory challenge) and time course over which microglia exhibit phenotypic remodeling (mins vs hours/days). Finally, to probe causal links between mitochondrial state and microglial properties we generated transgenic crosses to selectively manipulate microglial mitochondria and link organelle state to cellular properties. Collectively, these studies advance the field by revealing the mitochondrial landscape within microglia and the relationship of these organelles to microglial phenotypic remodeling across brain regions, in response to inflammatory insult, and during aging.

## Results

### Region-specific phenotypes of microglia are accompanied by regional differences in mitochondrial mass and number

Mitochondrial abundance, morphology, and molecular composition vary dramatically across cell types[39], suggesting that the relationship between mitochondria and cellular function cannot be generalized across cell types and needs to be defined on a cell by cell basis. To unequivocally visualize microglial mitochondria, we crossed mice that express inducible cre recombinase in microglia ($CX3CR1^{CreER/+}$) with mice that express Cre-dependent mitochondrial-targeted GFP (mitoGFP) (Fig. 1A)[40,41]. In fixed tissue from young adult (2mo) double transgenic $CX3CR1^{CreER/+};mitoGFP$ mice ($MG$-$mitoGFP$ mice hereafter), numerous GFP+ structures could be observed throughout the somas and cell processes of microglia in the nucleus accumbens (NAc) and ventral tegmental area (VTA), two interconnected brain regions where we have extensively mapped microglial phenotypes across the lifespan (Fig. 1B and Fig. S1A, B)[30,42,43]. Consistent with reports of tamoxifen-independent recombination when using $CX3CR1^{CreER/+}$ mice and reporters with minimal distance between loxP sites[44,45], recombination was

observed in approximately 80% of microglia in $MG$-$mitoGFP$ mice without tamoxifen administration (Fig. S1A–C) or in oil-injected controls (Fig. S1D, E). FACS-based analysis indicated that 4-hydroxytamoxifen treatment increased recombination to approximately 90% of microglia (Fig. S1F, G). Given the high level of tamoxifen-independent recombination, we elected to carry out analyses without tamoxifen-treatment, unless otherwise noted. In terms of cellular and organelle specificity of mitoGFP, FACS analyses indicated that GFP expression was not observed in non-microglial CNS cells (Fig. S1H, I). Moreover, in acute brain sections from $MG$-$mitoGFP$ mice, GFP+ structures were consistently colocalized with tetramethylrhodamine (TMRM), a mitochondrial membrane potential indicator, confirming that GFP-tagged structures are mitochondria (Fig. S1J, K). Together these results indicate that $MG$-$mitoGFP$ mice accurately label microglial mitochondria.

Microglia show prominent differences in cell density, branching complexity, and gene expression across the NAc and VTA[30]. To determine if these regional differences in baseline microglial attributes were accompanied by regional differences in microglial mitochondria, we carried out high resolution confocal imaging and 3D volumetric reconstruction of both microglia and their mitochondria in the NAc and VTA of young adult mice (Fig. 1B and Fig. S2A). Consistent with previous findings, microglial tissue coverage in the VTA was lower than that observed in NAc (Fig. 1C)[30]. Yet, field of view (FOV) mitochondrial mass (mitochondrial volume relative to microglial volume) and FOV mitochondrial number relative to microglial volume were greater in the VTA compared to the NAc (Fig. 1D, E). The FOV sphericity, median volume, max volume, and aspect ratio (longest axis/shortest axis - measure of elongation) of individual mitochondria were relatively consistent across NAc and VTA (Fig. S2B–E). Together, these observations indicate that regional differences in microglial cellular phenotypes are accompanied by regional differences in some but not all mitochondrial metrics.

### Mitochondrial number is partially correlated with morphological complexity of microglia

Microglial capacity to survey the surrounding brain tissue is influenced by the degree of their cell branching complexity. To define how mitochondrial networks relate to microglial branching structure, we crossed $MG$-$mitoGFP$ mice to Ai14 reporter mice to drive cytosolic expression of TdTomato (TdTom) along with GFP tagging of microglial mitochondria (Fig. 1A)[40,41,46]. In the absence of tamoxifen injection, incidence of mitoGFP+TdTom+ cells was low enough to allow rigorous and unequivocal reconstruction of the branching patterns of individual microglial cells and their mitochondrial networks in VTA and NAc as well as prefrontal cortex (PFC) (Fig. 1F and Fig. S2F). Consistent with previous findings, microglia display morphological differences across these brain regions, with NAc microglia exhibiting greater total process length than VTA microglia (Fig. 1G)[30]. PFC microglia displayed total process length comparable to that of VTA microglia. To relate these regional differences in total process length to mitochondria of the cells, we used regression analyses and found that greater process length was significantly associated with higher total volume of mitochondria in NAc microglia (Fig. 1H). Both VTA and PFC microglia showed trends toward greater process length being correlated with a greater number of mitochondria (Fig. 1H). These findings suggest that a greater quantity/volume of mitochondria are needed to support a greater mass of cell processes.

Microglia also displayed regional differences in morphology in terms of total number of branch points, with this metric being elevated in NAc microglia compared to VTA and PFC microglia (Fig. 1I). Robust regression analyses revealed that greater branching was associated with a greater number of mitochondria for VTA microglia (Fig. 1J). Trends toward greater branching being associated with a larger total volume of mitochondria were also evident for NAc and VTA microglia.

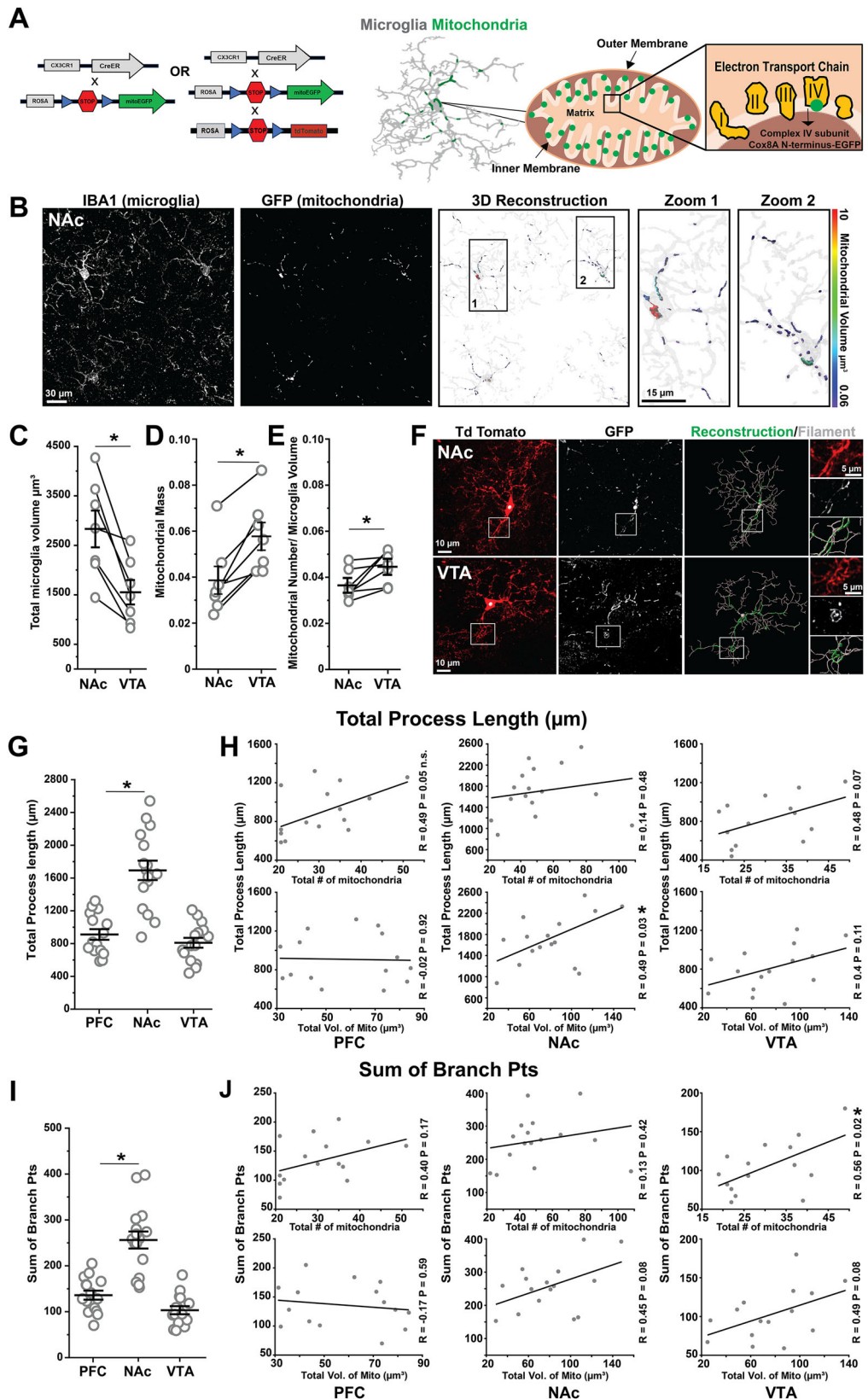

Intriguingly, these findings suggest that the *architecture* of cell processes, rather than simply the total *amount* of cell processes, may be related to mitochondrial networks within microglia (Fig. 1J). Consistent with this idea, Sholl analysis revealed that both the number of intersections (reflecting the number of cell process branches / morphological complexity) and the number of mitochondria reached a maximum at approximately 20–25 μm from the cell soma in the PFC, NAc, and VTA microglia (Fig. S2G–I). Hence, the subcellular distribution of the organelles aligns with branching architecture, as opposed to mitochondria being primarily clustered in the soma or at the motile tips of microglial processes. In addition, the maximum number of Sholl intersections was significantly correlated with the total number of

**Fig. 1 | Regional specialization of microglia is accompanied by regional differences in mitochondrial mass and number. A** Schematic of transgenic crosses used to achieve GFP labeling of microglial mitochondria. **B** Representative high-magnification images from P60 *MG-mitoGFP* mice of NAc microglia (*Iba1*) and mitochondria (*GFP*); panels *at right* show 3D volumetric reconstructions of microglia (*transparent gray*) and mitochondria (*colored according to volume*). **C–E** Field of view (FOV) microglial volume (tissue coverage; *$P = 0.003$*), FOV microglial mitochondrial mass (FOV mitochondrial volume / FOV microglial volume; *$P = 0.001$*), and FOV microglial mitochondrial number (relative to FOV microglial volume; *$P = 0.026$*) in NAc and VTA. Data points from the same mouse connected by lines ($N = 7$ mice, stats via two-tailed paired t-tests). **F** Representative confocal images of microglial cells in NAc and VTA expressing cytosolic TdTomato (TdTom) and mitochondrial targeted GFP (*MG-mitoGFP;Ai14* mice). Reconstruction of cell branching and 3D volumetric reconstruction of mitochondria in Imaris *at right*.

**G** Total process length of individual microglia from PFC ($N = 15$ cells), NAc ($N = 16$ cells), and VTA ($N = 15$ cells). One-way ANOVA with Bonferroni correction, $F_{2,43} = 31.30$, $P = 0.0001$. *$P < 0.05$. **H** Robust regression plots relating mitochondrial number and volume to total process length of individual microglia from PFC ($R = 0.40$ $P = 0.17$), NAc ($R = 0.13$ $P = 0.42$), and VTA ($R = 0.56$ $P = 0.02$). *$P < 0.05$ **I** Sum of branch points of individual microglia from PFC, NAc, and VTA. One-way ANOVA with Bonferroni correction, $F_{2,43} = 36.73$, *$P = 0.0001$. *$P < 0.05$. **J** Robust regression plots relating mitochondrial number and volume to morphological complexity (sum of branch points) of individual microglia from PFC ($R = 0.17$ $P = 0.59$), NAc ($R = 0.45$ $P = 0.08$), VTA ($R = 0.49$ $P = 0.08$). *$P < 0.05$. Mice used for these experiments were not treated with 4-hydroxytamoxifen. Data for panels **C, D, E, G,** and **I** was plotted as mean +/- SEM. Source data for all graphs are provided as a Source Data file.

mitochondria in PFC, with NAc and VTA microglia showing trends toward significance in this metric (Fig. S2J). Altogether, these analyses suggest that baseline microglial morphology and features of their mitochondria are partially interlinked.

## Mitochondrial number and mitochondrial motility are not aligned with microglial tissue surveillance

Microglia engage in continual morphological remodeling as they survey the surrounding tissue. Microglial tissue surveillance can also vary across brain regions and is assumed to be an energy-intensive process[7,47]. However, the relationship between microglial cell process motility and mitochondrial abundance has not been explored. We performed two-photon imaging in acute brain sections prepared from young adult (2-3mo) *MG-mitoGFP;Ai14* mice and recorded motility of NAc and VTA microglial cell processes (TdTom) and attributes of mitochondria (GFP) within these cells (Fig. 2A, B and Fig. S3A–C). Microglia were imaged at least 60 μm below the surface of the brain section. Although microglia show more simplified branching in acute brain sections compared to fixed tissue, total mitochondrial volume and total number of microglial Sholl intersections were positively correlated (Fig. S3D), indicating that alignment of mitochondria with the morphological architecture of the cells was preserved in acute brain sections. Moreover, microglial motility and mitochondrial motility were not correlated with time elapsed since preparation of brain sections (Fig. S3E, F), suggesting that this preparation is well suited for initial exploration of relationships between mitochondria and microglial cell dynamics.

Although NAc and VTA microglia displayed regional differences in morphology and mitochondrial mass in fixed tissue (Fig. 1), microglial cell process motility, defined as percent pixel change over a 10 min period, was similar across NAc and VTA in acute brain sections (Fig. 2B, C and Fig. S3B). In addition, despite several associations between microglial morphology and features of their mitochondria in fixed tissue (Fig. 1H, J and Fig. S2J), there was no correlation between total mitochondrial volume or mitochondrial mass and microglial motility in NAc or VTA (Fig. 2D, E) in acute brain sections. These observations suggest that there is a more robust relationship between mitochondrial abundance and baseline cell morphology than with *dynamic remodeling* of that morphology.

Microglial mitochondria themselves also exhibited motility that included jitter, concerted directional movement, and occasional examples of fission and fusion (Fig. 2F, G and Fig. S3C). These types of mitochondrial motility are consistent with what has been reported in other cell types[29,48]. 3D reconstruction and tracking of mitochondria in Imaris revealed that motility of individual mitochondria (total displacement over 10 mins) was similar in NAc and VTA microglia, with the majority of mitochondria displaying modest movement (approximately 1 μm) and a smaller number of mitochondria showing larger movements of 2-6 μm (Fig. 2H). To begin exploring whether mitochondrial motility was related to cell process remodeling of individual

microglia, we used robust regression analyses and found that mitochondrial motility (percent pixel change for all mitochondria in the cell) was not significantly correlated with microglial motility in either the NAc or the VTA (Fig. 2I). This was also the case at the level of primary branches of microglia. Even within one microglial cell, some primary branches contained few or no mitochondria while others contained multiple mitochondria. However, there was no significant correlation between motility of primary microglial branches and either mitochondrial motility (Fig. 2J) or mitochondrial mass within those branches (Fig. S3G). Altogether, these data further reinforce the idea that mitochondria are not tightly linked to baseline microglial tissue surveillance within a 10 min time frame.

## Mitochondrial distribution is stable during early phases of microglial response to focal injury

In addition to homeostatic tissue surveillance, microglia exhibit rapid and robust cell process extension toward focal sites of CNS injury[49–51]. In acute brain sections from *MG-mitoGFP;Ai14* mice, we induced focal laser lesions and imaged the response of both microglia and their mitochondria to this insult. Consistent with previous findings in the field[49,52], microglia rapidly extended cell processes toward and contacted the lesion within 5 min (Fig. 3A–G). However, these rapidly extending cell processes did not contain any mitochondria and no mitochondria were trafficked into lesion-responsive cell processes during the imaging period (Fig. 3A–E). Quantitative analyses confirmed minimal change in mitochondrial (GFP+) signals within 30 μm of the lesion over a 20 min period (Fig. 3H, I). Qualitative observations suggested minimal remodeling of mitochondria in the somas and non-extending processes of microglia that responded to laser lesion (Fig. 3A–E, zoom).

## Mitochondrial remodeling is evident at the earliest stages of microglial responses to inflammatory challenge

Microglia respond to a wide variety of CNS insults that vary substantially in both the nature of the insult (injury vs. infection vs. disease process vs. pathological electrical activity, etc.) as well as the time course over which microglial responses emerge and evolve (minutes to months). Given lack of clear evidence for mitochondrial involvement in rapid microglial responses to tissue injury, we sought to further explore the relationship between mitochondria and microglial phenotypic remodeling using a pathological insult of distinct modality as well as time course. Intraperitoneal (i.p.) injections of lipopolysaccharide (LPS), a component of the cell wall of gram-negative bacteria, were used to model inflammatory responses that accompany infection. Microglia show prominent changes in gene expression within hours of LPS injection and changes in morphology by 24–48 hours post-injection[53]. In vitro LPS exposure alters microglial mitochondrial structure and function[54–57] and, in macrophages, mitochondrial alterations play an essential role in enabling and regulating changes in macrophage attributes as they respond to inflammatory

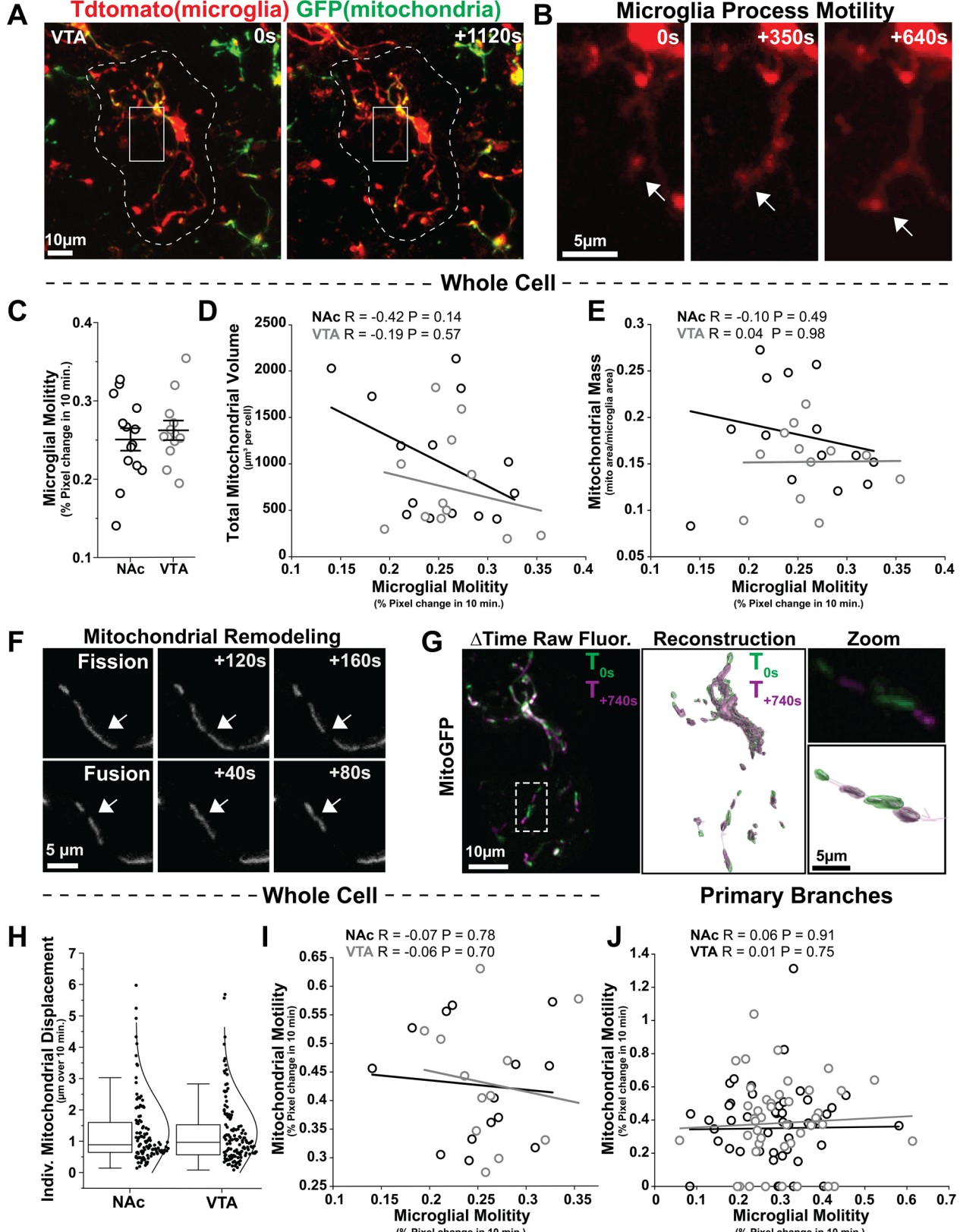

challenges[24–27,29,58,59]. All together, these findings suggest that there may be important relationships between mitochondria, as organelles, and microglial responses to inflammatory challenge.

Young adult (3-4mo) *MG-MitoGFP* mice were given i.p. injections of saline or LPS and microglial gene- and protein-expression were analyzed via FACS, using the previously described gating strategy to

identify and isolate microglia (Fig. S1D), followed by downstream qPCR (Fig. 4A). High expression of key microglial genes (*Cx3cr1* and *P2ry12*) and little to no expression of oligodendrocyte-lineage, astrocyte-, and neuron-specific genes (Fig. S4A) confirmed fidelity of our microglial isolation. Within 4 h of LPS challenge, microglia from both striatum (STR, containing NAc) and midbrain (MB, containing VTA) upregulated

**Fig. 2 | Mitochondrial abundance and mitochondrial motility are not strongly correlated with microglial tissue surveillance. A** Representative images taken during multiphoton live imaging of VTA microglia (*TdTomato*) and their mitochondria (*GFP*) in acute brain sections from 1.5–2mo old *MG-mitoGFP;Ai14* mice. **B** Example of microglial cell process extension indicated by *white arrow*. **C** Overall microglial motility in NAc and VTA during a 10-min period. *N* = 14 NAc cells and *N* = 12 VTA cells, *P* = 0.659, two-tailed t-test. Data was plotted as mean +/- SEM. **D** Robust regression analysis relating mitochondrial volume (total volume in a cell averaged across time points) and microglial motility over a 10-min period. *N* = 14 NAc cells, R = −0.42 *P* = 0.14 and *N* = 12 VTA cells, R = −0.19 *P* = 0.57. **E** Robust regression analysis relating mitochondrial mass and microglial motility over a 10-min period. *N* = 14 NAc cells, R = −0.10 *P* = 0.49 and *N* = 12 VTA cells, R = 0.04 *P* = 0.98. **F** Representative examples of mitochondrial fission and fusion observed in microglia from *MG-mitoGFP* mice. **G** Raw fluorescence from multi-photon imaging (*left*) and 3D reconstruction of microglial mitochondria (*right*), colored by time. **H** Displacement of individual mitochondria across cells over a 10-min period. *N* = 116 NAc mitochondria and *N* = 134 VTA mitochondria. Box plot displaying median, 25-75 percentiles (box), and 5th and 95th percentiles (whiskers). **I** Robust regression analysis relating mitochondrial motility to microglial motility. *N* = 14 NAc cells, R = −0.07 *P* = 0.78 and *N* = 12 VTA cells, R = −0.06 *P* = 0.70. **J** Robust regression analysis relating mitochondrial motility *within* one primary microglial branch to motility of the primary branch itself. *N* = 55 primary branching processes from NAc microglia, R = 0.06 *P* = 0.91 and *N* = 46 primary branching processes from VTA microglia, R = 0.01 *P* = 0.75. Mice used for these experiments were not treated with 4-hydroxytamoxifen. Source data for all graphs are provided as a Source Data file.

transcripts for pro-inflammatory factors (*Il1β, Tnfα*), and down-regulated transcripts for homeostatic microglial genes (*Tgfβ1, P2ry12, Cx3cr1*), consistent with previous studies[53,60]. Upregulation of genes associated with microglial phagocytic function (*Cd45, Itgam*) was evident by 24 h post-injection (Fig. 4B, S4B and Table S1, S2). Overall, STR and MB microglia responded similarly to LPS challenge at the gene expression level.

Microglial expression of mitochondrial-function related genes was also altered at both 4 h and 24 h post injection (Figs. 4B, S4B and Tables S1, S2). Within 4 h of LPS challenge, genes related to mitochondrial ROS handling (*Sod2, Cat*), mitochondrial fission (*Dnm1l*), and mitochondrial biogenesis (*Nrf1*) were significantly altered. Some of these gene expression changes were maintained into 24 h post injection (*Sod2, Dnm1l*), while others appeared more restricted to the 4 h time point (*Cat, Nrf1*). At 24 h post injection, prominent upregulation of electron transport chain component genes *Ndufa12* (Complex I), *Cox4i1* (Complex IV), and *AtpSd* (Complex V) emerged. Genes related to mitochondrial calcium handling (*Vdac1, Micu1*) were also significantly altered. Again, most gene expression changes were similar across STR and MB (Figs. 4B, S4B and Tables S1, S2).

To probe the relationship between mitochondrial state and key features of microglia as they respond to inflammatory challenge, we used Principal Component Analysis (PCA) and found that PCA based solely on expression patterns of mitochondrial function-relevant genes can explain 64.7% of sample variation with the first 2 components and 81% of variation with the first 3 components (Fig. 4C and Fig. S4C). Moreover, 24 h post-injection samples cluster away from Saline samples (Fig. 4C) and 4 h post-injection samples are interspersed among Saline samples or skewed toward the 24 h samples. Hence, without even considering microglial inflammatory genes or homeostatic genes, microglia that are responding to LPS challenge can be clearly identified on the basis of their mitochondrial function profiles. We also used robust regression analyses and found numerous correlations between individual microglial-state and mitochondrial state genes (Fig. 4D and Table S3). For example, expression of homeostatic microglial gene *P2ry12* showed significant correlation with electron transport chain (ETC) genes (*Ndufa12, Sdha*) as well as *Tfam*, which influences expression of mitochondrial DNA encoded ETC genes, in saline-treated animals, but the majority of these correlations were lost at 4 h post-LPS. Microglial pro-inflammatory genes *Il1β* and *Tnfα* showed significant correlation with mitochondrial biogenesis genes (*Nrf1, Tfam*), mitochondrial fusion and fission genes (*Mnf1, Dnm1l*) and the ROS handling gene *Sod2* at 4 h post LPS (Fig. 4D and Table S3). Microglial- and mitochondrial- gene correlations remained abundant and continued to evolve at 24 h post LPS.

Together these data indicate that numerous facets of mitochondrial function are remodeled during microglial responses to inflammatory challenge and that mitochondrial remodeling begins at the earliest stages of microglial responses to inflammatory challenges. Moreover, examination of *Cd68*, a key microglial lysosome gene, hints that mitochondria, as organelles, may be uniquely positioned to regulate microglial functional state during inflammatory challenge. While LPS treatment increased the correlation between most mitochondria-relevant genes and key microglial function genes (*Cd45, Cx3cr1, P2ry12, Tgfβ1*) (Fig. 4D), LPS treatment did not modulate correlations between *Cd68* and *Cx3cr1, P2ry12, Tgfβ1* (correlations between *Cd68* and *Cd45* increased at 24 h post LPS) (Fig. S4D and Table S4). Moreover, while multiple mitochondria-relevant genes were positively correlated with expression of the microglial pro-inflammatory factors *Tnfα* and *Il1β* at 4 h post-LPS, *Cd68* expression was not correlated with *Tnfα* and *Il1β* in any of the conditions examined (Fig. S4D).

The transcript-level, qPCR-based analyses presented thus far show high sensitivity for the detection of early changes in cell and organelle state following LPS challenge. To test protein-level changes following the LPS challenge, we analyzed data recorded during FACS isolation of microglia. These analyses revealed that the intensity of GFP, reflecting mitochondrial mass, was significantly increased at 24 h post-injection in both the STR and MB, with hints of emerging increases at 4 h post-injection (Fig. 4E, F). In contrast, mitochondrial membrane potential, as detected by TMRM intensity, decreased significantly at 24 h post LPS (Fig. 4E, F). These changes in mitochondrial mass and mitochondrial membrane potential were accompanied by significant protein-level increases in CD45 by 24 h in STR microglia and trends toward increased CD45 in MB microglia. Protein level expressions of CX3CR1 and P2RY12 differed across regions but not across treatment (Fig S4E). Microglial size, as indicated by forward scatter, was significantly elevated in both STR and MB by 24 h post LPS, while internal complexity (side scatter) remained unchanged (Fig S4E). These data confirm that changes in mitochondrial state at the protein and functional level are associated with changes in microglial protein-level expression and cell structure.

**Increased mitochondrial mass and mitochondrial elongation are associated with microglial responses to inflammatory challenge**

To investigate how LPS challenge impacts the morphology of microglial mitochondria, we analyzed fixed tissue from 3-4mo *MG-MitoGFP* mice (Fig. 5A, Fig. S5A) at 24 h post-LPS, when protein-level changes in microglia and their mitochondria were most evident via FACS (Fig. 4). 3D reconstruction of both microglia and their mitochondria in NAc and VTA revealed significant increases in mitochondrial mass at 24 h post LPS (Fig. 5B), consistent with FACS-based analyses of GFP and qPCR-based increases in mitochondrial biogenesis factor *Nrf1*. No significant changes in mitochondrial number (relative to cell volume) were detected (Fig. 5C) and, instead, microglial mitochondria showed significant elongation, as detected by increases in aspect ratio (Fig. 5D). Elongation of mitochondria has been associated with increased capacity for oxidative phosphorylation[61–63], which would be consistent with the qPCR-based observation of robust increases in electron transport chain components (*Ndufa12, AtpSd, Cox4i1*) at 24 h post LPS (Fig. 4B). Collectively, these findings indicate that mitochondrial structural remodeling is a key component of microglial responses to inflammatory challenge.

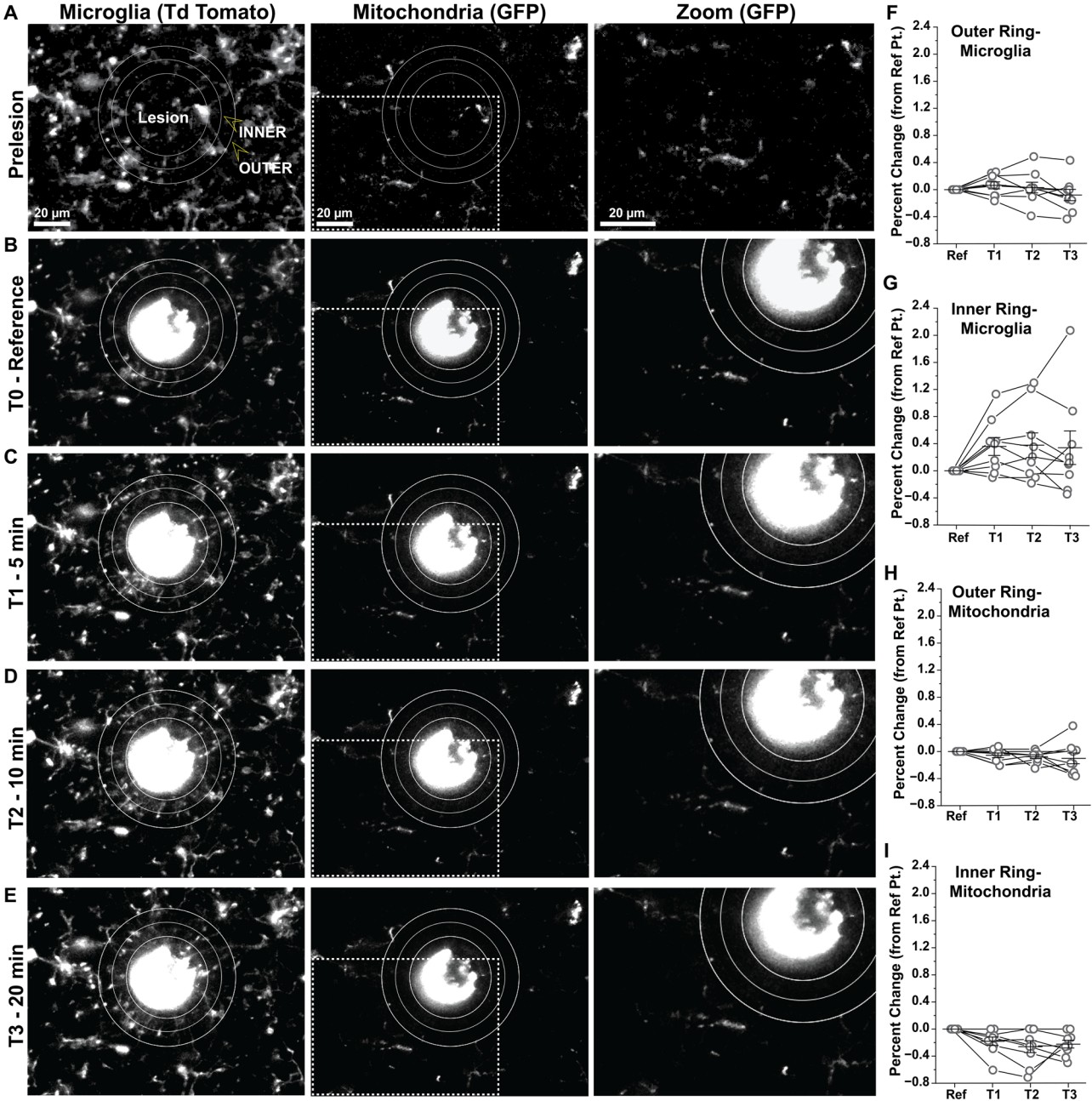

**Fig. 3 | Mitochondrial distribution is stable during early phases of microglial response to focal tissue injury. A** Representative image from multiphoton live imaging of NAc microglia (*TdTomato*) and their mitochondria (*mitoGFP*) in an acute brain slice from young adult *MG-mitoGFP;Ai14* mouse prior to induction of focal laser injury. **B** TdTom+ microglia and GFP+ mitochondria from the same field of view as in *A* immediately following induction of focal laser lesion (T0). Ring 1 encompasses the primary lesion area, with *inner* and *outer* rings at 15 μm and 30 μm from ring 1, respectively. **C–E** TdTom+ microglia and GFP+ mitochondria from the same field of view as in *A* at 5, 10, and 20 min post laser lesions. **F, G** Percent change in TdTomato signal (microglial processes) within the outer and inner rings, respectively, relative to T0. One-way ANOVA with repeated measures: Outer ring within subjects comparison $F_{(3,24)} = 1.6$, $P = 0.215$, Outer ring between subjects

comparison $F_{(1,8)} = 0.01$, $P = 0.942$, Inner ring within subjects comparison $F_{(3,24)} = 2.38$, $P = 0.095$, Inner ring between subjects comparison $F_{(1,8)} = 4.08$, $P = 0.078$. **H, I** Percent change in GFP signal (mitochondria) within the outer and inner rings, respectively, relative to T0. One-way ANOVA with repeated measures: Outer ring within subjects comparison $F_{(3,24)} = 1.2$, $P = 0.318$, Outer ring between subjects comparison $F_{(1,8)} = 3.69$, $P = 0.091$, Inner ring within subjects comparison $F_{(3,24)} = 5.4$, $P = 0.005$, Inner ring between subjects comparison $F_{(1,8)} = 14.5$, $P = 0.005$. For data in **F–I**, $N = 3$ NAc acute brain sections from 3 mice (9 NAc acute brain sections total) and data was plotted as mean +/- SEM. Mice used for these experiments were not treated with 4-hydroxytamoxifen. Source data used to generate all graphs are provided as a Source Data file.

## Microglial mitochondria undergo morphological and functional changes during aging

The results described above indicate that there are key relationships between mitochondria and region-specific microglial morphology (Fig. 1), as well as microglial expression of homeostatic genes and inflammatory factors in the context of an inflammatory insult (Fig. 4).

Microglia exhibit numerous changes during aging, including altered morphology and inflammatory profile[13,43,64–66]. Altered mitochondrial function is heavily implicated in CNS aging, but aging-associated changes in mitochondria have not been investigated with cellular specificity[67–69]. Hence, whether and how mitochondrial remodeling is linked to changing microglial attributes during

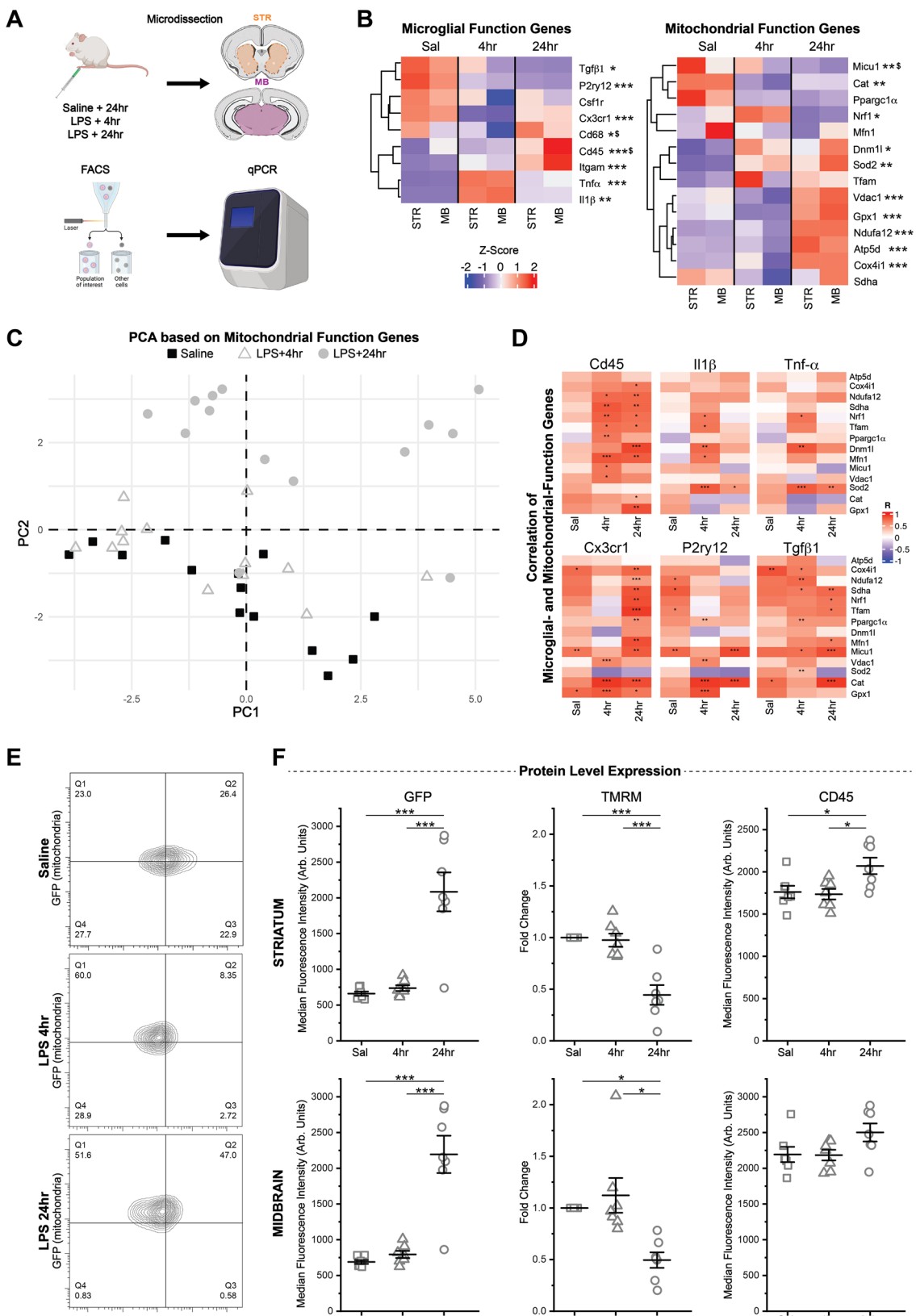

aging is unknown. We also showed previously that microglial responses to aging vary across brain regions, with VTA microglia displaying proliferative and inflammatory changes months before microglia in the NAc[43]. Whether there are links between mitochondrial state and early VTA microglial aging has also not been investigated.

To determine if aging-associated changes in microglial morphology and inflammatory profile are accompanied by changes in their mitochondria, we analyzed NAc and VTA microglial mitochondria in fixed tissue from middle-aged (12-13mo) and late middle-aged (16-18mo) *MG-MitoGFP* mice (Fig. 6A) and compared them to microglial mitochondria in young adult (2-3mo) mice.

**Fig. 4 | Response to inflammatory challenge (LPS) elicits rapid mitochondrial changes in microglia. A** Schematic of the experiment design, created partially in BioRender https://BioRender.com/yedibtm. 3-4mo *MG-MitoGFP* mice were injected with saline or lipopolysaccharide (LPS) prior to FACS isolation of STR/MB microglia and downstream qPCR analysis. **B** Heatmaps showing relative expression levels ($2^{-\Delta Ct}$) of key microglial function and mitochondrial function genes. Z-scores represent average gene expression across treatment groups and brain regions. See Tables S1 and S2 for Two-way ANOVAs and post hoc comparisons with Bonferroni correction. Treatment: * $P < 0.05$, ** $P < 0.01$, *** $P < 0.001$. Brain region difference: \$ $P < 0.05$. (Saline STR $N = 7$ mice, Saline MB $N = 7$ mice, LPS 4 h STR $N = 7$ mice, LPS 4 h MB $N = 6$ mice, LPS 24 h STR $N = 7$ mice, LPS 24 h MB $N = 7$ mice). **C** PC1 and PC2 of Principal Component Analysis of samples using only mitochondrial function genes. **D** Heatmaps depicting the degree of correlation between expression levels of microglial function genes (pro-inflammatory - *Cd45*, *Il1β*, *Tnfα*; homeostatic -

*Cx3cr1*, *P2ry12*, *Tgfβ1*) and mitochondrial function genes as determined by robust regression analysis with false discovery rate correction. See Table S3 for R- and P-values. * $P < 0.05$, ** $P < 0.01$, *** $P < 0.001$. **E** Representative FACS contour plots showing TMRM and GFP fluorescence intensity of microglia from Saline, LPS 4 h, and LPS 24 h mice. **F** Median Fluorescence Intensity of GFP, TMRM (normalized to each cell's GFP to account for mitochondrial mass), and CD45 in STR and MB microglia from Saline ($N = 7$, squares), LPS 4 h ($N = 7$, triangles), and LPS 24 h ($N = 7$, circles) mice. One-way ANOVAs and post hoc comparisons with Bonferroni correction: GFP STR, $F_{(2,18)} = 25.06735$, $P < 0.0001$; GFP MB, $F_{(2,18)} = 29,85264$, $P < 0.0001$; TMRM STR, $F_{(2,18)} = 19.2316$, $P < 0.0001$; TMRM MB, $F_{(2,18)} = 6.61021$, $P = 0.00704$; CD45 STR, $F_{(2,18)} = 5.52274$, $P = 0.01348$; CD45 MB, $F_{(2,18)} = 2.99003$, $P = 0.07565$. * $P < 0.05$, ** $P < 0.01$, *** $P < 0.001$. Data was plotted as mean +/- SEM. Mice used for these experiments were not treated with 4 hydroxytamoxifen. Source data used to prepare all graphs are provided as a Source Data file.

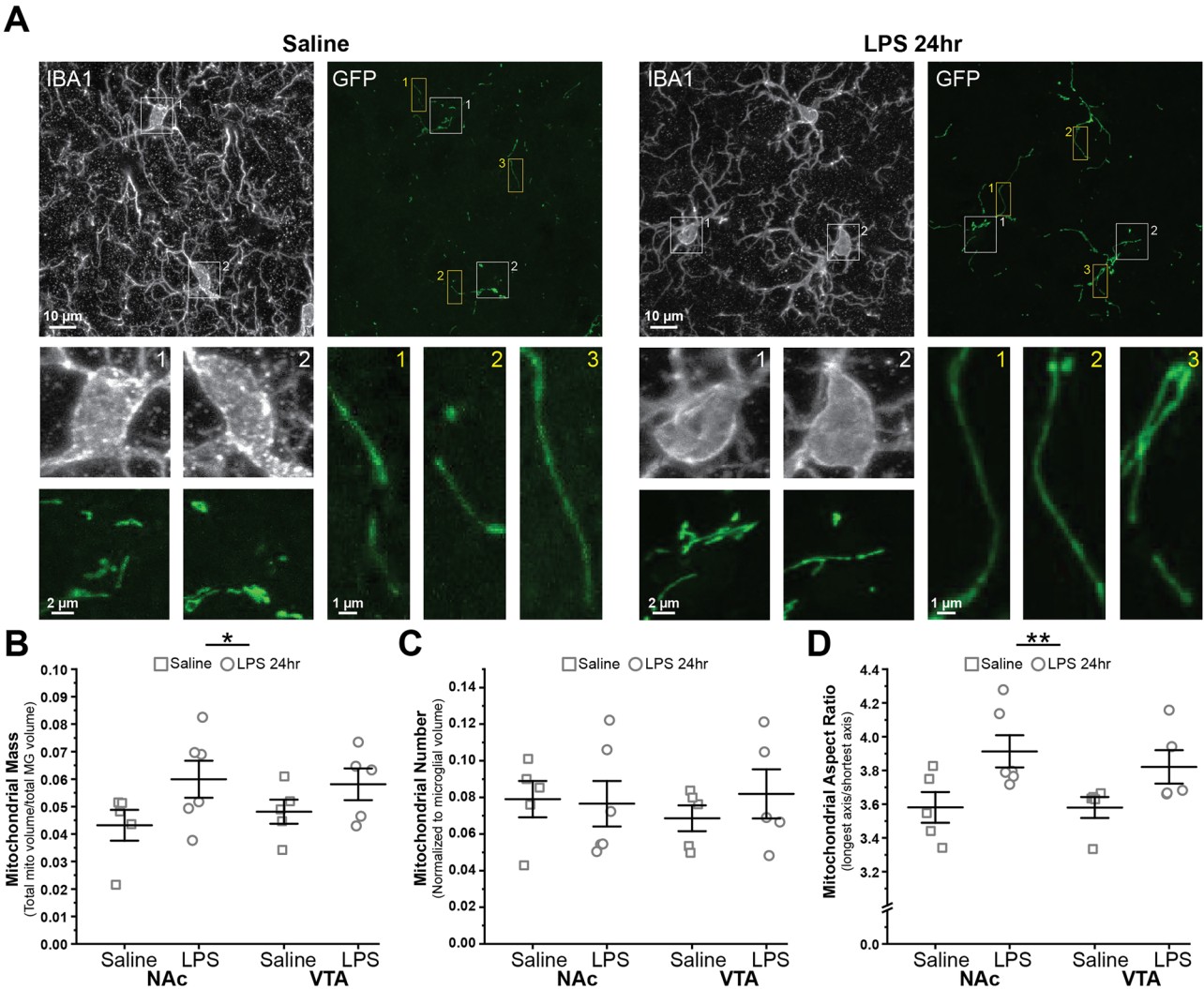

**Fig. 5 | Microglial responses to inflammatory challenge (LPS) are accompanied by mitochondrial elongation and increases in mitochondrial mass.**
**A** Representative images of NAc microglia (IBA1) and their mitochondria (GFP) from 3-4mo *MG-MitoGFP* mice injected with saline (*left*) or LPS 24 h (*right*). *White* and *yellow boxes* highlight microglial somas and cell processes shown at higher magnification in *lower panels*. **B** Field of view (FOV) microglial mitochondrial mass (FOV mitochondrial volume / FOV microglial cell volume) within NAc and VTA microglia of saline (*squares*) or LPS treated mice (*circles*); Two-way ANOVA with Bonferroni correction for post hoc comparisons: main effect of treatment, $F_{(1,17)} = 5.19787$, $P = 0.03579$; main effect of region, $F_{(1,17)} = 0.0713$, $P = 0.079266$. **C** FOV mitochondrial number (relative to FOV microglial cell volume) within NAc

and VTA microglia of saline (*squares*) or LPS treated (*circles*) mice; Two-way ANOVA with Bonferroni correction for post hoc comparisons: main effect of treatment: $F_{(1,17)} = 0.23167$, $P = 0.63642$; main effect of region, $F_{(1,17)} = 0.05079$, $P = 0.82438$. **D** Average mitochondrial aspect ratio (longest axis / shortest axis) within NAc and VTA microglia of saline (*squares*) or LPS treated (*circles*) mice; Two-way ANOVA with Bonferroni correction for post hoc comparisons: main effect of treatment, $F_{(1,17)} = 10.19319$, $P = 0.00533$; main effect of region, $F_{(1,17)} = 0.26571$, $P = 0.61286$. Quantification for *B-D* from $N = 5$ saline-injected mice and $N = 6$ LPS-injected mice. Data for *B-D* was plotted as mean +/- SEM. *$P < 0.05$, **$P < 0.01$. Mice used for these experiments were not treated with 4 hydroxytamoxifen. Source data used to generate all graphs are provided as a Source Data file.

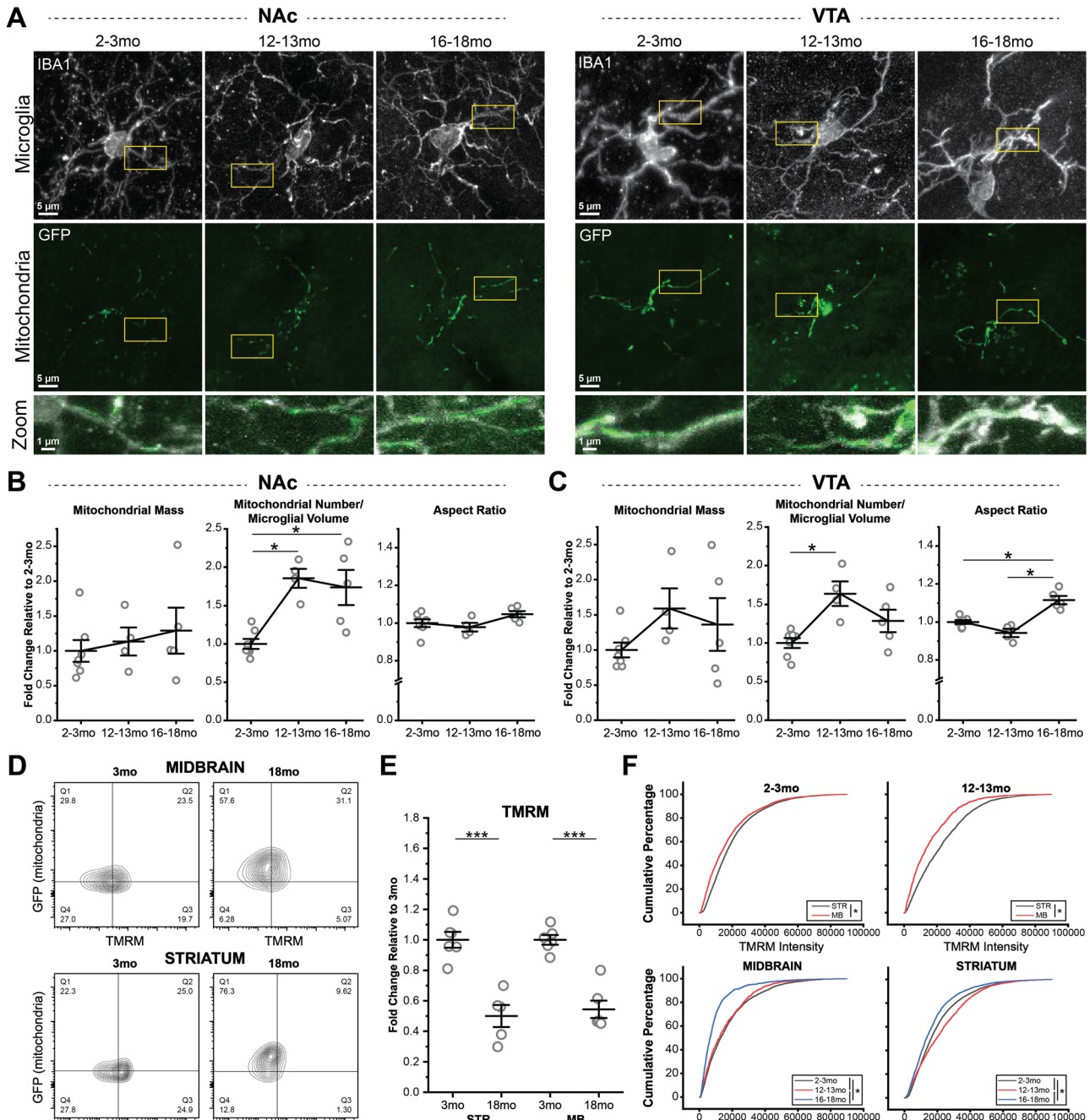

**Fig. 6 | Microglial mitochondria undergo remodeling in middle- and late middle aged mice. A** Representative images of NAc (*left*) and VTA (*right*) microglia (IBA1) and microglial mitochondria (GFP) from 2-3mo, 12-13mo, and 16-18mo *MG-MitoGFP* mice. Regions highlighted by *yellow boxes* shown at higher magnification in *lower panels*. **B, C** Field of view (FOV) mitochondrial mass, number, and aspect ratio of NAc and VTA microglia during aging, normalized to mean values from 2-3mo animals. 2-3mo ($N = 7$ mice), 12-13mo ($N = 4$ mice), and 16-18mo ($N = 5$ mice). One-way ANOVAs with Bonferroni correction for post hoc comparisons: mitochondrial mass NAc, $F_{(2,13)} = 0.43588$, $P = 0.65581$, mitochondrial number NAc, $F_{(2,13)} = 11.5342$, $P = 0.00132$, mitochondria aspect ratio NAc, $F_{(2,13)} = 2.54264$, $P = 0.11695$, mitochondrial mass VTA, $F_{(2,13)} = 1.47803$, $P = 0.26401$, mitochondrial number VTA, $F_{(2,13)} = 7.57294$, $P = 0.0066$, mitochondrial aspect ratio VTA, $F_{(2,13)} = 24.40763$, $P < 0.0001$. * $P < 0.05$. Data plotted as mean +/- SEM. **D** Representative FACS contour plots showing TMRM fluorescence intensity (measure of mitochondrial membrane potential) and GFP fluorescence intensity

(mitochondrial abundance) of microglia from 2-3mo and 16-18mo *MG-MitoGFP* mice. **E** TMRM median fluorescence intensity (normalized to individual cell mitoGFP signal to account for mitochondrial abundance) of STR and MB microglia from 3mo ($N = 6$ STR, 6 MB) and 18mo ($N = 5$ STR, 6 MB) *MG-MitoGFP* mice. Two-way ANOVA with Bonferroni correction for post hoc comparisons: main effect of age, $F_{(1,19)} = 79.60044$, $P < 0.0001$; main effect of region, $F_{(1,19)} = 0.1661$, $P = 0.68816$. *** $P < 0.001$. Data was plotted as mean +/- SEM. **F** Cumulative probability distributions comparing TMRM fluorescence intensity across brain regions (STR *black*, MB *red*) in microglia from 2-3mo and 12-13mo C57Bl6 wildtype mice (*top panels*) and comparing TMRM fluorescence intensity across ages (2-3mo *black*, 12-13mo *red*, 16-18mo *blue*) in MB and STR microglia (*bottom panels*). * $P < 0.05$, Kolmogorov-Smirnov tests. Mice used for these experiments were not treated with 4 hydroxytamoxifen. Source data used to generate all graphs are provided as a Source Data file.

Tissues were treated with TrueBlack to quench autofluorescent aggregates (lipofuscin) that emerge during aging and any dim residual signals in microglia could be clearly distinguished from *MG-MitoGFP* signal during analysis (Fig. S6A)[70]. With age, mitochondrial mass tended to increase, particularly in the VTA, and significant increases in mitochondrial number (relative to cell volume) were observed in both NAc and VTA microglia (Fig. 6B, C). VTA microglia also showed significant increases in mitochondrial elongation at 16-18mo (Fig. 6B, C). Hence, at the level of mitochondrial structure, LPS challenge and aging both elicit similar types of mitochondrial remodeling.

Although aging impacts microglia in both the NAc and VTA, key regional differences in microglial aging are also apparent. VTA microglia show more prominent lysosome swelling, proliferation, and inflammatory profiles, while NAc microglia show more prominent reductions in cell branching complexity[43]. To further explore how regional differences in microglial aging relate to mitochondrial remodeling during aging, within-mouse comparisons were carried out using paired t-tests. This analysis revealed that the elevated mitochondrial mass and mitochondrial number in VTA microglia relative to NAc microglia that were observed at 2-3mo were no longer significant at 12-13mo and 16-18mo, likely due to increased mouse-to-mouse variation (Fig. S6B, C) and increased mitochondrial number (relative to cell volume) in NAc microglia. Finally, VTA microglia exhibited an elevated aspect ratio (greater mitochondrial elongation) relative to NAc microglia at 16-18mo (Fig. S6D). Therefore, some regional differences in microglial mitochondria that are evident in young adulthood (mitochondrial mass and number) are lost during aging, while other regional differences in microglial mitochondria are gained (elongation).

To further explore how aging impacts microglial mitochondria, we used the previously described gating strategy (Fig. S1D) for FACS-based analysis of microglia from 3mo and 18mo *MG-mitoGFP* mice. This analysis revealed that the median fluorescence intensity of GFP was significantly increased with age in both STR (containing NAc) and MB (containing VTA) microglia (Fig. S6E), pointing to increases in mitochondrial mass. Conversely, mitochondrial membrane potential (TMRM intensity normalized to GFP) was significantly reduced with age in both regions (Fig. 6D, E). To confirm changes to mitochondrial membrane potential and further elucidate the time course of such changes, we analyzed TMRM intensity of microglia from WT mice and included an intermediate 12–13mo time point. In this analysis, MB microglia exhibited slightly more depolarized mitochondrial membrane potential (lower TMRM intensity) than STR microglia in 2–3mo old mice (Fig. 6F). These regional differences grew progressively larger in 12-13mo old mice (Fig. 6F) and 16–18mo old mice (Fig. S6F). Examining the time course of TMRM changes within each region revealed subtle but significant depolarization in 12-13mo MB microglia and pronounced depolarization in 16-18mo MB microglia, compared to MB microglia from 2-3mo mice. Interestingly, STR microglia exhibited hyperpolarized mitochondrial membrane potential (higher TMRM intensity) in 12–13mo mice compared to 2–3mo mice. At 16-18mo, however, they showed depolarized mitochondrial membrane potential, relative to microglia from 2-3mo mice (Fig. 6F). Together, these findings indicate that the functional status of microglial mitochondria changes significantly during aging, with key differences in the time course and magnitude of these changes across brain regions. Moreover, they reveal that remodeling of microglial mitochondrial status begins in middle age, rather than in the late stages of aging.

### Targeted genetic manipulation of microglial mitochondria elicits changes in microglial morphology and gene expression

Results presented thus far indicate that there are correlations between region-specific microglial morphology and mitochondrial mass/number. They also reveal correlations between mitochondrial

mass, mitochondrial elongation, mitochondrial membrane potential, and changes in microglial morphology and inflammatory profiles during responses to LPS challenge and aging. To further probe whether there are causal links between mitochondria and these microglial phenotypic attributes, we crossed *CX3CR1*[CreER/+] mice and *CX3CR1*[CreER/+]*;mitoGFP* mice with *Tfam*[fl/fl] mice to achieve targeted deletion of mitochondrial transcription factor A (TFAM) in microglia (*MG-TFAMKO* or *MG-mitoGFP-TFAMKO*) (Fig. 7A). TFAM deletion in diverse cell types leads to reduced expression of electron transport chain subunits encoded by mitochondrial DNA and gives rise to changes in mitochondrial membrane potential and mitochondrial morphology, making this an ideal genetic manipulation to further elucidate the relationship between mitochondrial and microglial phenotypes[71–75].

For these experiments, mice were injected with 4-hydroxytamoxifen and analyzed 2 months later to allow for both TFAM deletion and turnover of existing mitochondrial components that will be impacted by TFAM deletion, as mitochondrial proteins are among the most long-lived within the cell[76]. This interval also allows for the replacement of any peripheral CX3CR1+ cells that experience TFAM deletion in these mice. FACS followed by qPCR confirmed that this treatment paradigm resulted in downregulation of *Tfam* in both STR and MB microglia in *MG-TFAMKO* mice (Fig. 7B). Histology in fixed tissue revealed that microglial TFAM loss resulted in increased mitochondrial volume, mitochondrial number, and mitochondrial sphericity at the tissue (FOV) level in *MG-mitoGFP-TFAMKO* mice (Fig. 7C–F, Fig. S7A). Mean intensity of the mitoGFP signal was also elevated in *MG-mitoGFP-TFAMKO* mice in both NAc and VTA (Fig. S7A–C). This increase in GFP intensity appeared to vary substantially from cell-to-cell in *MG-mitoGFP-TFAMKO* mice and may reflect incomplete TFAM recombination in some microglia. Analysis of individual cells confirmed increases in mitochondrial number and trends toward increased mitochondrial sphericity in microglia from *MG-mitoGFP-TFAMKO* mice (Fig. S7D, E). This analysis also highlighted increased cell-cell variation in microglia from *MG-mitoGFP-TFAMKO* mice. Some properties of microglial mitochondria, such as FOV median and max mitochondrial volume, were unchanged following TFAM loss (Fig. S7F). Together, these findings confirm that this genetic manipulation induces changes to some of the same mitochondrial attributes (number, sphericity, GFP intensity) that differ across brain regions and that are remodeled during microglial responses to LPS and aging.

To determine how genetically-induced changes in mitochondrial status relate to microglial properties, we analyzed microglial density, tissue coverage, and individual microglial morphological complexity. Microglial density did not differ significantly between *MG-mitoGFP-TFAMKO* and control mice (Fig. 7G). However, microglial tissue coverage was significantly increased in both NAc and VTA microglia from *MG-mitoGFP-TFAMKO* mice (Fig. 7H), suggesting that this manipulation caused changes to microglial morphology. To investigate this possibility, we also reconstructed individual microglial cells and assessed branching complexity via Sholl analysis. The total number of Sholl intersections was elevated in microglia from *MG-mitoGFP-TFAMKO* mice, particularly in the NAc (Fig. 7I), supporting the idea that there are important links between mitochondrial state and microglial morphology. Moreover, this *increase* in morphological complexity suggests that TFAM deletion does not simply make microglia "reactive" or compromise their health. Finally, no significant changes were observed in FOV mitochondrial number (relative to FOV microglial volume) or mitochondrial mass in *MG-mitoGFP-TFAMKO* mice (Fig. 7J–K), which would be expected if both microglial volume and mitochondrial number/volume are increased in these mice.

To further elucidate how genetically targeted manipulation of microglial mitochondria affects microglial properties, we used FACS analysis of STR and MB microglia (gated via the previously described strategy, Fig. S1D) as well as qPCR for a panel of microglial- and

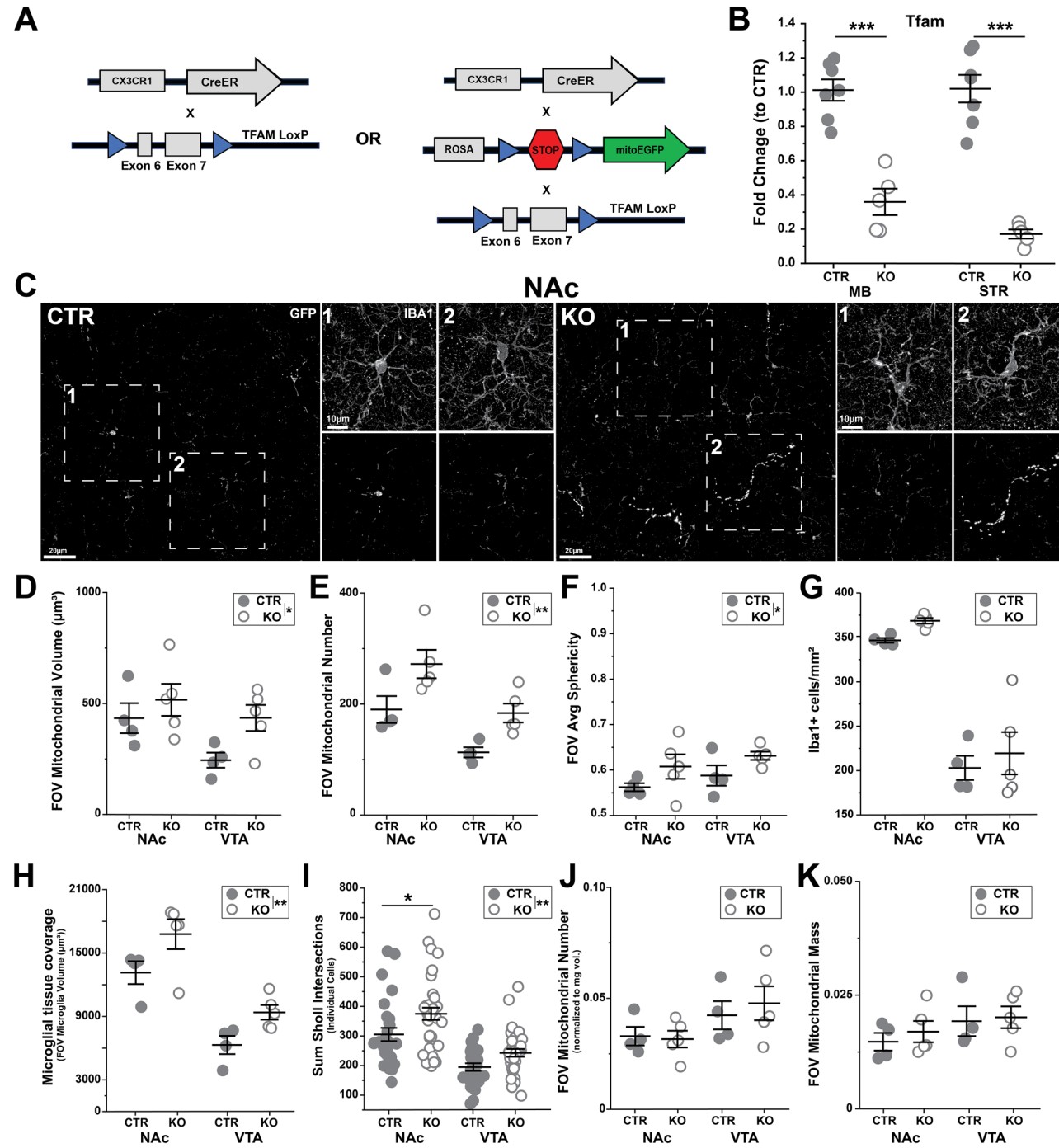

mitochondrial-function related genes. FACS analysis revealed significant increases in microglial forward scatter (FSC, cell size) and side scatter (SSC, internal cellular complexity) in *MG-TFAMKO* samples (Fig. 8A), consistent with observed increases in microglial tissue coverage/volume (Fig. 7H). Protein level expression of CD45 and CX3CR1 (median fluorescence intensity, MFI, measured during FACS) differed across brain regions, but was not significantly altered by TFAM loss. qPCR analysis revealed that multiple mitochondria-related genes were altered in response to TFAM knockout. This included increased expression of genes coding for components of the electron transport chain (*Ndufa12, Atp5d, Cox4i1, Sdha*), as well as genes related to mitochondrial ROS (*Sod2* and *Cat*) and calcium handling (*Micu1*, Fig. 8B, Fig. S8A and Table S5). Mitochondrial biogenesis genes *Nrf1* and *Ppargc1a* also showed significant changes, but in opposite directions (Fig. 8B, Fig. S8A and Table S5). Mitochondrial fusion gene *Dnm1l*

was significantly decreased and there were no significant changes in expression of mitochondrial fission-associated gene *Mfn1* (Fig. 8B, Fig. S8A and Table S5). Together, these data indicate that loss of microglial TFAM induces broad changes to expression of nuclear-encoded, mitochondria-relevant genes and suggest that multiple aspects of mitochondrial function are impacted by this manipulation.

These alterations in mitochondrial state upon TFAM knockout were accompanied by significant changes in genes related to key microglial functions. Proinflammatory transcripts *Il1β* and *Tnfα* were significantly increased in *MG-TFAMKO* microglia while *Ifnβ1* was significantly decreased. For multiple microglial genes that show regional differences in expression (*Cd45, Cd68, P2ry12*), these regional differences were preserved in TFAM knockout mice, and were not significantly modulated by TFAM loss. Instead, *Tgfβ1* and *Cx3cr1*, which

**Fig. 7 | Loss of mitochondrial transcription factor A (TFAM) leads to altered morphology of both microglia and their mitochondria. A** Schematic of transgenic crosses used to achieve microglial-specific deletion of TFAM in microglia with or without simultaneous GFP labeling of microglial mitochondria. **B** qPCR-based analysis of TFAM expression (fold change relative to control, $2^{-\Delta\Delta CT}$) in FACS-isolated microglia from the midbrain (MB) and striatum (STR). Two-way ANOVA with Bonferroni correction for post hoc comparisons: main effect of genotype, $F_{(1,24)} = 115.410, P < 0.0001$, main effect of brain region $F_{(3,20)} = 1.654, P = 0.213$. $N = 7$ CTR mice (*filled circles*), $N = 5$ KO mice (*open circles*). **C** Representative high-magnification images of NAc microglia (IBA1) and mitochondria (GFP) from 5mo *MG-mitoGFP* (CTR) and *MG-mitoGFP-TFAMKO* (KO) mice. **D** Total mitochondrial volume within the field of view (FOV); Two-way ANOVA with Bonferroni correction for post hoc comparisons: main effect of genotype $F_{(3,14)} = 4.808, P = 0.046$, main effect of region $F_{(3,14)} = 4.716, P = 0.046$. **E** FOV mitochondrial number, Two-way ANOVA with Bonferroni correction for post hoc comparisons: main effect of genotype $F_{(3,14)} = 13.489, P = 0.002$, main effect of brain region $F_{(3,14)} = 15.864$, $P = 0.0013$. **F** Average FOV mitochondrial sphericity, Two-way ANOVA with Bonferroni correction for post hoc comparisons: main effect of genotype $F_{(3,14)} = 5.313$, $P = 0.037$, main effect of brain region $F_{(3,14)} = 1.636, P = 0.221$. **G** Microglial density, Two-way ANOVA with Bonferroni correction for post hoc comparisons: main effect of genotype $F_{(3,14)} = 1.666, P = 0.218$, main effect of brain region $F_{(3,14)} = 96.163$, $P < 0.0001$. **H** Microglial tissue coverage (FOV volume), Two-way ANOVA with Bonferroni correction for post hoc comparisons: main effect of genotype $F_{(3,14)} = 9.665, P = 0.007$, main effect of brain region $F_{(3,14)} = 43.274, P < 0.0001$. **I** Branching complexity of individual microglia (total number of Sholl intersections), NAc $N = 27$ CTR and $N = 35$ KO cells; VTA $N = 25$ CTR $N = 33$ KO cells, Two-way ANOVA with Bonferroni correction for post hoc comparisons: main effect of genotype $F_{(3,116)} = 10.314, P = 0.002$, main effect of brain region $F_{(3,116)} = 45.863$, $P < 0.0001$. **J** FOV mitochondrial number normalized to FOV microglial volume, Two-way ANOVA with Bonferroni correction for post hoc comparisons: main effect of genotype $F_{(3,14)} = 0.366, P = 0.554$, main effect of brain region $F_{(3,14)} = 2.295$, $P = 0.152$. **K** FOV mitochondrial mass (FOV mitochondrial volume normalized to FOV microglial volume), Two-way ANOVA with Bonferroni correction for post hoc comparisons: main effect of genotype $F_{(3,14)} = 0.119, P = 0.734$, main effect of brain region $F_{(3,14)} = 4.728, P = 0.047$. Data was plotted as mean +/- SEM. For panels **D–H, J** and **K**, $N =$ CTR 4 and 5 KO mice. For all data panels, CTR samples are represented by *filled circles* and KO samples are represented by *open circles*. Mice were treated with 4-hydroxytamoxifen and analyzed 2 months later as described in methods. * $P < 0.05$, ** $P < 0.01$,*** $P < 0.001$. Source data used to generate all graphs are provided as a Source Data file.

have both been implicated in modulation of overall microglial phenotypes, were significantly increased in microglia from *MG-TFAMKO* mice (Fig. 8C). Post hoc analysis revealed that many of the gene expression changes associated with TFAM KO occurred in a brain region-specific manner. Increases in *Tnfα* were primarily observed in MB TFAM knockout samples, while increases in *Tgfβ1* and *Cx3cr1* were only prominent in STR TFAM knockout microglia (Fig. 8C and Table S5). Leveraging PCA analysis to examine these gene expression changes in an integrated fashion revealed that *MG-TFAMKO* samples from STR and MB appeared to cluster separately from one another (Fig. 8C and Fig. S8B–C), providing further evidence that manipulating mitochondrial function exerts distinct effects on STR and MB microglia.

### Loss of TFAM causes subtle changes in LPS-induced mitochondrial remodeling and microglial responses to inflammatory challenge

In addition to defining how mitochondrial manipulation impacts basal microglial phenotypes, we sought to investigate the impact of TFAM knockout on microglial ability to respond to an inflammatory challenge. To this end, *MG-mitoGFP-TFAMKO* and *MG-mitoGFP* control mice were given i.p. injections of LPS, and microglial gene- and protein-expression were analyzed via FACS and downstream qPCR. To maximize ability to detect protein level alterations, as well as catch potential differences in time course of microglial transcriptional responses, samples were collected and analyzed at an intermediate time point (12 h) between those examined previously (4 h and 24 h, Figs. 4, 5). On the whole, TFAM loss did not appear to substantially impair remodeling of microglial mitochondria following LPS treatment. For example, both microglia from *MG-mitoGFP* control and *MG-mitoGFP-TFAMKO* mice showed increased mitoGFP at 12 h post LPS (Fig. 8D), consistent with previous observations of increased mitoGFP at 24 h post LPS (Fig. 4F). Moreover, the magnitude of the mitoGFP increase did not differ across genotypes. TMRM levels, which were seen to decrease at 24 h post LPS (Fig. 4F), had not yet declined at 12 h post LPS and did not differ across genotypes (Fig. S8D). Similarly, at the transcript level, LPS-induced changes in expression of mitochondria-relevant genes appeared to proceed irrespective of TFAM loss (Table S6). For example, *Sod2* (up-regulated at 4 h and 24 h compared to saline) and *Vdac1* (down-regulated at 4 h compared to saline) were still up- and down-regulated in TFAM KO microglia at 12 h post LPS (Fig. S8F, G). In cases where significant differences were observed across genotypes, many of these were already evident without LPS challenge. For example, *Cox4i* was

elevated in TFAM KO microglia relative to controls at baseline (Fig. 8B), and it remained elevated relative to controls after LPS (Fig. S8H).

The microglial response to LPS-challenge was also largely preserved at both the protein- and gene-expression, irrespective of TFAM loss. Microglia from both *MG-mitoGFP* control and *MG-mitoGFP-TFAMKO* mice showed increases in FSC (cell size) at 12 h post injection (Fig. 8D), consistent with observations at 24 h post LPS (Fig. 4F). TFAM KO microglia showed significantly higher FSC than control microglia, but this increased FSC was already evident in the absence of LPS treatment (Fig. 8A). Regional differences in expression of CD45 and CX3CR1 across STR and MB that were present at baseline in TFAM KO and control microglia (Fig. 4F, Fig. S4E, Fig. 8A) remained present at 12 h post LPS (Figs. 8D, and S8D). At the transcript level, the prominent increases in *Tnfα* and *Il1β* that are observed following LPS injection (Fig. 4B), were still observed in TFAM KO microglia (Fig. 8E and Fig. S8H) and did not differ significantly in magnitude from responses in control microglia. Similarly, LPS-induced downregulation of homeostatic transcripts *Tgfβ1* and *Cx3cr1* did not differ across genotypes (Fig. 8E and Fig. S8I). Expression levels of *Cd68* and *Itgam* did differ significantly across genotypes, with TFAM KO microglia showing more muted reductions in expression compared to control samples at 12 h post LPS injection (Fig. 8E).

Altogether, these subtle differences in response to LPS challenge were evident in PCA analysis comparing the gene expression patterns of saline-treated *MG-mitoGFP* samples (Fig. 4B), untreated *MG-TFAMKO* samples (Fig. 8B), and 12 h LPS-treated control and *MG-mitoGFP-TFAMKO* samples. This analysis revealed that LPS-treatment was the strongest driver of sample clustering (Fig. 8G) on PC1, with genotype effects being evident on PC2. Together, these data suggest that microglial mitochondrial remodeling and changes in microglial attributes in response to acute inflammatory challenge remain largely intact in *MG-mitoGFP-TFAMKO* mice. Future studies using repeated or chronic inflammatory challenge will be needed to determine whether the subtle differences in mitochondrial remodeling that were observed lead to more prominent downstream shifts in microglial function.

## Discussion

Changes in mitochondrial function have been associated with developmental delays, epilepsy, psychiatric disorders, aging, and neurodegeneration[77,78]. A key priority in uncovering how mitochondria are linked to CNS pathology is understanding how mitochondria impact the function of specific CNS cells. However, because

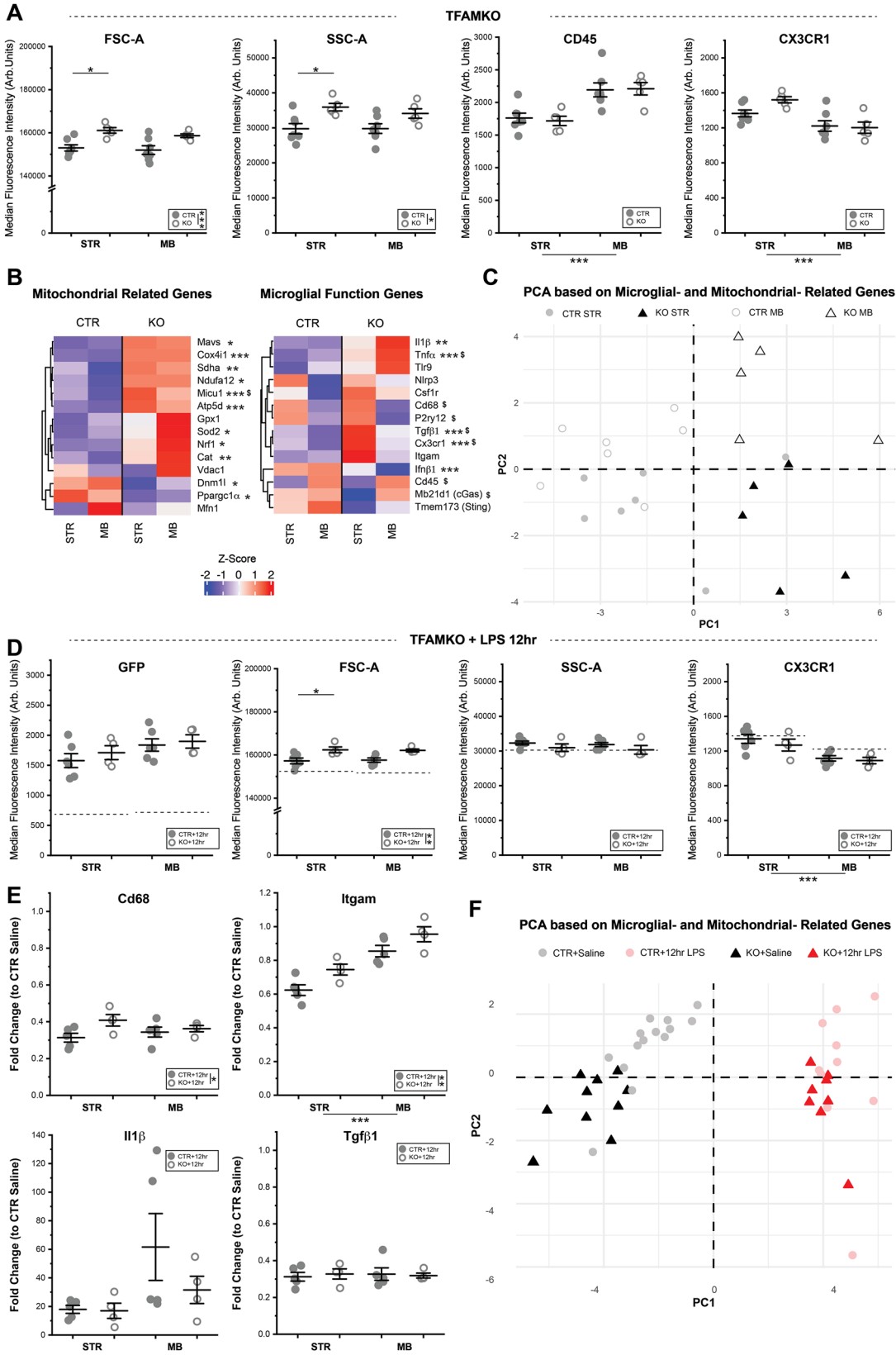

mitochondrial structure and molecular composition vary widely across distinct cell types[39], findings about the functional roles of these organelles cannot be generalized, and mitochondria need to be studied with cellular specificity. Our study provides detailed information about microglial mitochondria and supports the idea that mitochondrial function is distinct across CNS cells. We observed that

mitochondria occupy about 4-6% of microglial volume, and entire microglial branches sometimes have few, if any, mitochondria (Figs. 1, 2). This is similar to mature oligodendrocytes, where mitochondria make up about 4% of cell volume[79] but unlike astrocytes where mitochondria make up about 80% of astrocyte volume in young adult mice[80]. Qualitatively, our experiments using TMRM incubation in

**Fig. 8 | TFAM loss alters microglial size and baseline gene expression, but causes only subtle changes in microglial capacity to respond to acute inflammatory challenge. A** Median Fluorescence Intensity for forward scatter (FSC, cell size), side scatter (SSC, internal cellular complexity), CD45, and CX3CR1 in STR and MB microglia from $N = 7$ CTR mice (*filled circles*) and $N = 6$ TFAM KO mice (*open circles*). See Table S5 for F and P values from Two-way ANOVAs with Bonferroni correction for post hoc comparisons. **B** Heatmaps showing relative expression levels ($2^{-\Delta Ct}$) of key microglial function and mitochondrial function-related genes in each group. Z-scores represent average gene expression across genotypes and brain regions. See Table S5 for F and P values from Two-way ANOVA with Bonferroni correction for post hoc comparisons. * $P < 0.05$, ** $P < 0.01$, *** $P < 0.001$, significant effect of treatment, post hoc comparison with Bonferroni correction. $^{\$}$ $P < 0.05$, significant effect of brian region, post hoc comparison with Bonferroni correction. ($N = 7$ mice for CTR STR and CTR MB, $N = 5$ mice for TFAM KO STR and TFAM KO MB). **C** Principal Component Analysis of control (STR *filled gray circles*, MB *open gray circles*) and MG-TFAMKO (STR *filled black triangles*, MB *open black triangles*) samples based on microglial- and mitochondrial-function related genes. **D** Median Fluorescence Intensity of GFP, FSC, SSC, and CXC3CR1 signal in STR and MB microglia from $N = 6$ CTR mice (*filled circles*) and $N = 4$ TFAM KO mice (*open circles*) 12 h after LPS injection. The dotted line represents the average signal of microglia from saline-injected *MG-mitoGFP* mice (shown in Fig. 4F and Fig S4D). See Table S6 for F and P values from Two-way ANOVA with Bonferroni correction for post hoc comparisons. * $P < 0.05$, ** $P < 0.01$, *** $P < 0.001$, post hoc comparison with Bonferroni correction. **E** Fold change ($2^{-\Delta\Delta Ct}$) of key microglial genes *Cd68*, *Itgam*, *Il1β*, and *Tgfβ1* in $N = 5$ LPS-treated CTR mice (*filled circles*) and $N = 4$ LPS-treated KO mice (*open circles*) relative to microglia from saline-treated mice. See Table S7 for F and P values from Two-way ANOVA with Bonferroni correction for post hoc comparisons. * $P < 0.05$, ** $P < 0.01$, *** $P < 0.001$, post hoc comparison with Bonferroni correction. **F** Principal Component Analysis of microglia from saline- (*gray/black*) and LPS-treated (*pink/red*) CTR (*circles*) and TFAM KO (*triangles*) mice based on expression levels of microglial- and mitochondrial- function related genes. Fold-change analyses in *E* and PCA in *F* incorporate data from saline-treated *MG-mitoGFP* mice (Fig. 4 and Fig. S4) as well as untreated *MG-TFAMKO* mice (Fig. 8B). Mice for these experiments were injected with 4-hydroxytamoxifen and analyzed 2 months later as described in methods. Data in **A, D**, and **E** was plotted as mean ± SEM. Source data for all graphs are provided as a Source Data file.

acute brain sections also suggest that microglial mitochondria are less hyperpolarized than mitochondria of surrounding cells (Fig. S1H). Hence, our study underscores the need for cell-specific study of mitochondria in CNS cells and argues for caution in assuming that relationships between mitochondria and cell function in one cell type will also hold true for other cell types.

Previous studies have suggested that mitochondrial function in microglia changes in the context of disease, stress, and immune activation[31,32,52–54]. Almost nothing is known, however, about how these organelles are linked to the physiological functions of microglia. We found key relationships between mitochondrial abundance and total process length as well as branching complexity of microglia (Fig. 1). Hence, even if microglia have fewer mitochondria than other CNS cells, our data suggest that these organelles play a key role in shaping the general cellular architecture of microglia. This finding is further supported by our observation that targeted manipulation of microglial mitochondria (deletion of microglial TFAM) led to increased branching complexity of NAc and VTA microglia (Fig. 7I), as well as increased cell size assessed via FACS (Fig. 8A). Our qPCR-based analyses also indicated that expression of critical homeostatic microglial gene *P2ry12* was significantly correlated with expression of multiple mitochondrial genes under basal conditions. Moreover, targeted manipulation of microglial mitochondria (deletion of microglial TFAM) increased expression of the homeostatic microglial genes *Cx3cr1* and *Tgfβ1*, providing further evidence that the state of mitochondria impacts multiple molecules involved in basal microglial phenotypes. Finally, regional specialization is also a key feature of homeostatic microglia[6,9,10,30,42], and we found that regional differences in microglial cellular phenotypes were accompanied by regional differences in mitochondrial mass, mitochondrial number, and mitochondrial membrane potential in young adult mice (Figs. 1, 6). Altogether, these observations indicate that mitochondrial status is tightly linked to key microglial attributes under physiological conditions.

Additional studies will be required to fully elucidate the significance for physiological microglial function if a microglial cell exhibits higher mitochondrial mass, mitochondrial number, and/or distinct mitochondrial membrane potential. Increased mitochondrial mass, at the most basic level, means that a cell exhibits increased volume of mitochondria relative to the cell volume. While this could imply, simplistically, that a given cell has elevated energetic needs, the interpretation of increased mitochondrial mass is not so straightforward. Abundance of electron transport chain (ETC) components can vary across individual mitochondria[81,82], meaning that more mitochondrial volume does not automatically equate to enhanced capacity for oxidative phosphorylation and ATP production. Some mitochondria lack cristae and are specialized in the reductive synthesis of

macromolecules[81] needed to support growth and other cellular functions. The subcellular localization of mitochondria also matters. For example, mitochondria at pre-synaptic terminals in neurons play critical roles in calcium buffering[83]. Alternatively, mitochondria positioned together with ER membranes play critical roles in lipid synthesis and regulation of autophagy and inflammasome activation[84]. Thus, an elevated mitochondrial mass could have different functional implications depending on subcellular location.

In the present study, we did not observe obvious differences in subcellular distribution of mitochondria between NAc and VTA microglia (Fig. S2G–I) nor did we observe significant differences in the morphology/elongation of individual mitochondria (Fig. S2B–E) across these brain regions. We previously found via RNAseq of NAc and VTA microglia that the latter express higher levels of *Vdac1* and *Vdac2* (mitochondrial calcium channels) as well as *Sod2* (reactive oxygen species buffer). At the same time, VTA microglia express lower levels of *Atp5k* and *Atp5g2* (components of the ATP synthase). Taken together, these observations raise the possibility that elevated mitochondrial number and mitochondrial mass in VTA microglia may be related to non-ATP production functions of mitochondria. This idea would also be consistent with the observation of more depolarized mitochondrial membrane potential (lower TMRM intensity) observed in MB microglia (Fig. 6F), as mitochondrial membrane potential fuels ATP production. Future studies can employ genetically encoded mitochondrial calcium indicators and genetically encoded mitochondrial ROS indicators, as well as strategies to perturb mitochondrial calcium channels and ROS regulatory molecules in microglia, to further elucidate the functional significance of regional differences in microglial mitochondrial mass and number.

The idea that non-ATP-producing functions of mitochondria are important for microglia is also supported by our unexpected finding that microglial cell process motility did not show a positive correlation with the abundance of mitochondria (Fig. 2). We also analyzed the movement of the mitochondria themselves and observed the same forms of dynamic mitochondrial remodeling (fission, fusion, trafficking) described in other cell types. Yet, we failed to find any association between mitochondrial displacement and cell process motility, suggesting that mitochondrial trafficking to specific subcellular locations is not required for overall baseline microglial tissue surveillance. Connections between the abundance of mitochondria and cell process remodeling of fine branchlets may exist, however. Using in vivo multiphoton imaging, Pietramale and colleagues found that microglial branchlets with fewer mitochondria were more likely to be fully retracted[85]. Finally, we found that mitochondria are relatively stable within microglia as they rapidly extend cell processes toward focal laser lesions (Fig. 3). Similar responses were observed in vivo by

Pietramale and colleagues during early phases of microglial responses to a laser lesion. Intriguingly, at longer intervals post-lesion (6 h) they did observe significant mitochondrial remodeling in nearby microglia, suggesting roles for these organelles in microglial injury responses over hours-to-days[85]. Microglial tissue surveillance and rapid tissue injury responses are assumed to be energy-intensive processes. It may be, however, that the more rapid, albeit less efficient, process of glycolysis provides the primary energetic support for microglial tissue surveillance and injury-responsive motility. Future studies will need to investigate highly localized relationships between microglial filopodia and nearby mitochondria to examine whether mitochondrial membrane potential or other features of mitochondrial state (e.g., fission and fusion) are linked to movement of adjacent cell processes. Improvements in optical tools for the detection of ATP[86] and glycolytic function can also be used in conjunction with the assessment of mitochondrial abundance and localization to expand our understanding of how these organelles do or do not support the energetics of microglial motility.

Our LPS experiments indicate that remodeling of mitochondrial function is an early and prominent component of microglial responses to inflammatory challenge. Our finding of mitochondrial elongation following an LPS challenge differs from in vitro experiments that found fragmented mitochondria in primary microglia treated with LPS for 24 h[54]. Differences between in vitro versus in vivo microglia and direct versus systemic LPS treatment likely contribute to these distinct findings. In our experiments, rapid increases in microglial *Tnfα* and *Il1β* mRNA were accompanied by altered expression of multiple mitochondrial-relevant genes (Fig. 4). Moreover, expression levels of *Tnfα*, *Il1β*, and *Cd45* in individual samples at 4 h post-LPS were significantly correlated with expression levels of genes involved in mitochondrial fission and fusion, calcium handling, and ROS handling, in addition to genes involved in mitochondrial biogenesis and oxidative phosphorylation. These findings support the possibility that mitochondrial signaling *regulates* changes in microglial attributes rather than simply altering energy production to support a newly emerging cellular phenotype. The idea that mitochondrial state may regulate inflammatory profiles of microglia is further supported by the increased expression of *Tnfα* and *Il1β* in TFAM KO microglia, particularly in the midbrain (Fig. 8B). These qPCR results also suggest important future directions for investigation of the links between mitochondria and microglial inflammatory responses. For example, the surge in mitochondrial biogenesis transcription factor *Nrf1* at 4h post LPS (Fig. 4B) followed by significant increases in mitoGFP signal by 12h post LPS (Fig. 8D) suggests that targeted impairment of mitochondrial biogenesis capacity may be a fruitful strategy for dissecting how these organelles regulate microglial responses to inflammatory challenge.

Although TFAM deletion led to significant increases in microglial branching (Fig. 7I), overall cell size (Fig. 8A), and expression of inflammatory and homeostatic microglial genes (Fig. 8B), the impact of TFAM loss on acute microglial responses to LPS was relatively muted (Fig. 8D–F). A key consideration in interpreting these results is that TFAM loss did not substantially interfere with ability of microglia to increase mitochondrial biogenesis (increase mitoGFP) and alter expression of nuclear-encoded, mitochondria-relevant genes in response to LPS challenge. Hence, additional mitochondria-targeted manipulations will be needed to fully elucidate the relationship between these organelle and microglial responses to inflammatory challenge. Given that TFAM-KO microglia exhibit distinct morphology and expression of inflammatory and homeostatic genes at baseline, it will also be of interest to examine their responses to more chronic or repeated challenges, such as those encountered during normative aging or in the context of neurodegenerative disease. Although the impact of TFAM loss on acute inflammatory microglial responses is subtle, the cumulative effect of these subtle changes during a chronic challenge may be substantial.

Importantly, targeted manipulation of microglial mitochondria did not impact MB and STR microglia in the same way (Fig. 8C), suggesting that mitochondrial regulation of microglial function is not equivalent throughout the brain. Following TFAM loss, MB microglia showed substantial changes in proinflammatory genes (upregulation of *Tnfα* and *Il1β*, and downregulation of *Ifnβ1*). STR microglia, instead, exhibited prominent upregulation of homeostatic genes *Cx3cr1* and *Tgfβ1*. These findings align with some of our prior observations of regional differences in microglial responses to aging. Previously, we showed that VTA microglia exhibit early proliferative and inflammatory responses compared to NAc microglia during aging[43]. In the current study, we demonstrated similar trends in mitochondria, with MB microglia showing earlier and more prominent aging-induced changes to mitochondrial membrane potential compared to STR microglia (Fig. 6). When combined with our observation that targeted manipulation of microglial mitochondria leads to prominent changes in inflammatory factor expression specifically in MB microglia (Fig. 8), our findings support the idea that VTA/MB microglia are uniquely vulnerable to perturbations of mitochondrial function and that mitochondrial function is intimately linked to inflammatory profiles of these MB microglia.

Multiple mutations that increase risk for neurodegenerative disease are in mitochondrial-relevant genes[19,87,88] and microglia themselves are also highly implicated in shaping risk for and progression of neurodegeneration. Several recent studies have begun investigating links between microglial mitochondria and neurodegeneration, but more research is needed. Microglia deficient in mitochondrial complex I activity induced glial inflammation, synapse loss and premature death in juvenile mice[35] but, conversely, blocking complex I activity was neuroprotective in an EAE-model of multiple sclerosis[36]. Similarly, age- and disease model-dependent findings have emerged using models of Alzheimer's disease. In 5x FAD mice, microglial deletion of a subunit of complex III resulted in increased microglial coverage of amyloid beta plaques and decreased plaque number[37], while deletion of microglial mitochondrial translocator protein (TSPO) impaired microglial phagocytosis of amyloid beta in a different mouse model of Alzheimer's[38]. Future studies should delve into whether mitochondria-associated disease risk stems from brain region-specific impact of mitochondrial function on neuroprotective vs. neurotoxic phenotypes of microglia.

Several limitations of our study are important to note. First, we carried out our studies using light microscopy, and this limits the ability to determine if densely packed mitochondria remain distinct or are part of a larger, interconnected mitochondrial network. In microglial somas, this was particularly evident, and future studies using super-resolution approaches such as STED imaging will be needed to more clearly determine if there are changes in degree of mitochondrial interconnectedness in microglial somas across brain regions, during aging, or following pathological challenges. A second limitation of our study is that border-associated macrophages (BAMs) can also be labeled by Iba1 immunostaining and are expected to experience recombination and mitoGFP expression in *CX3CR1^{CreER/+};mitoGFP* mice. As BAMs are primarily concentrated in the meninges and vascular interfaces and are significantly less abundant than microglia, we are confident that our results are focused on parenchymal microglia[89]. Nonetheless, future studies employing markers of BAMs can explicitly analyze the mitochondria of BAMs and identify similarities and differences of BAM and microglial mitochondria and how these organelles regulate cell function.

Important future directions for this work and for the field also include deeper investigation of whether there are sex differences in microglial mitochondrial function. Both male and female mice were included in our study, and we did not observe clear sex differences in most analyses of mitochondrial number, elongation, and remodeling

following LPS challenge (Fig. S9). However, we did observe potential overall elevations of mitochondrial mass in microglia from female mice compared to microglia from male mice (Fig. S9A). Sex differences in microglial function are increasingly recognized in multiple contexts, including neurodevelopment and neurodegenerative disease[90] and this will be an important area to explore further. Finally, mitochondria are highly complex organelles with numerous functions beyond ATP production. Our study provides insight into how these organelles align with key physiological properties of microglia as well as their attributes during responses to CNS insults. To expand upon this work and continue to elucidate how mitochondria impact microglial attributes and functions, we need to develop more tools that allow cell-specific manipulation of discrete facets of mitochondrial function, such as calcium buffering, lipid metabolism, ROS handling, and interactions with other organelles.

## Methods

### Mouse lines

*C57Bl6 WT* mice were obtained from the NIA Aged Rodent Colony and housed in the UCLA vivarium for at least one month before experiments. *CX3CR1^CreER/+* mice were originally obtained from Jackson Labs (stock #025524) and subsequently maintained in our colony[41]. *Rosa26-lsl-mito-GFP* mice (*mitoGFP* mice) were obtained from Dr. Jeremy Nathans (Johns Hopkins University) and are available at Jackson Labs (stock #021429). These mice enable cre-dependent expression of GFP that is fused to an N-terminal signal derived from mouse cytochrome c oxidase, subunit VIIIa, enabling specific localization to mitochondria[40]. *Ai14 TdTomato* reporter mice were obtained from Jackson Labs (stock #007914) and subsequently maintained in our colony[46]. Mice used for experiments were heterozygous for these transgenes. *Tfam^fl/fl* floxed mice were obtained from Jackson Laboratories and have *loxP* sites flanking exons 6-7 of the transcription factor A, mitochondrial (*Tfam*) gene (stock #026123)[75]. These mice were crossed to the *CX3CR1^CreER* mice with a final breeding strategy to generate *CX3CR1^CreER;Tfam^fl/fl* and *CX3CR1^CreER;Tfam^+/+* littermate controls. In all experiments, balanced numbers of male and female mice were used. Mice were housed in a normal light/dark cycle and had ad libitum access to food and water. All experiments were performed in strict accordance to an animal use protocol (ARC−2018-103) approved by the Animal Care and Use Committees at UCLA.

### Tamoxifen injections

In general, experiments were carried out without treating mice with tamoxifen. For select experiments as noted, 4-Hydroxytamoxifen (H7904-25mg, 95% pure z-isomer) was dissolved at 20 mg/mL in ethanol. For injections preparation, 4-OHT was redissolved in ethanol and diluted in a 1:4 mixture of castor oil:sunflower seed oil (Sigma, Cat #s 259853 and S5007) to give a final concentration of 1 mg/kg. *CX3CR1^CreER/+;mitoGFP* mice at 3-4 mos received intraperitoneal injections of 4-OHT or vehicle (oil mixture) twice a day for 3 days at 1 mg/kg. Mice were euthanized 3 weeks later for immunohistochemistry and flow cytometry. *CX3CR1^CreER/+;mitoGFP;Tfam^fl/fl* mice and *CX3CR1^CreER/+;Tfam^fl/fl* at 3-4 mos received injections of 4-Hydroxytamoxifen twice a day for 3 days at 1 mg/kg. The mice were housed for 2 months after injections to allow for turnover of long-lived mitochondrial proteins[76,91]. Mice were then euthanized for immunohistochemistry or flow cytometry.

### Lipopolysaccharide injections

*CX3CR1^CreER/+;mitoGFP* and *CX3CR1^CreER/+;mitoGFP;Tfam^fl/fl* mice 3 to 7 months (mos) of age received single intraperitoneal injections of 1 mg/kg of Lipopolysaccharide (LPS) (Millipore Sigma, strain O55:B5, cat# L2880-25MG) or saline. All injections were performed around ~10am. For immunohistochemistry, mice were deeply anesthetized via isoflurane inhalation and transcardially perfused with

1 M phosphate-buffered saline (PBS; pH 7.4) followed by ice-cold 4% paraformaldehyde (PFA) in 1 M PBS. For flow cytometry, mice were deeply anesthetized via isoflurane inhalation and transcardially perfused with 10 mL of chilled 1 M PBS. Timepoints for experiments were 4 h, 12 h, or 24 h post-injection.

### Tissue collection and immunohistochemistry

Mice were deeply anesthetized via isoflurane inhalation and transcardially perfused with 1 M PBS followed by ice-cold 4% PFA in 1 M PBS. Brain tissue was isolated and postfixed in PFA for ~4 h at 4 °C and then stored in 1 M PBS with 0.1% sodium azide until sectioning. Coronal brain sections were prepared using a vibratome at 60 μm thickness. For immunostaining, free-floating brain sections were washed with 1 M PBS (5min) and then permeabilized and blocked in a solution of 2% bovine serum albumin (BSA), 3% normal donkey serum (NDS), and 0.01% Triton-X for 2 h with rotation at room temperature. Sections were then incubated overnight with primary antibodies prepared in 2% bovine serum albumin (BSA) and 3% normal donkey serum (NDS) at 4 °C with rotation. Sections from aged mice were incubated in True-Black lipofuscin autofluorescence quencher, (5%; Biotium cat: 23007) for 60 s followed by a 3×5 min rinse in 1 M PBS prior to incubation with primary antibodies. Following primary antibody incubation, sections were washed 3x with 1 M PBS and then incubated with secondary antibodies prepared in 2% bovine serum albumin (BSA) and 3% normal donkey serum (NDS) at room temperature for 2 h with rotation. This was followed by washes in 1 M PBS (3x) with a second wash containing 1:4000 dilution of DAPI in 1 M PBS. Sections were mounted using Aqua-Poly/Mount (Polysciences cat: 18606) and cover-slipped. Primary antibodies used include: rabbit anti- Iba1 (1:800; Wako, catalog #019-19741), rat anti-CD68 (1:200; clone FA11, AbD Serotec, catalog #MCA1957), chicken anti-TH (1:500; Aves, catalog #TYH). Secondary antibody combinations include rabbit AlexaFluor-647, chicken Alexa-Fluor-594, rat AlexaFluor-594, or chicken AlexaFluor-405 (used at 1:1000; all raised in donkey; Jackson ImmunoResearch Laboratories). For each set of analyses, three brain sections containing nucleus accumbens (NAc) and prefrontal cortex (PFC) and three brain sections containing ventral tegmental area (VTA) were chosen from each mouse. Brain sections were selected based on well-defined anatomical parameters and were matched for anterior-posterior location.

### Confocal image acquisition and analysis

Fixed tissue slides were imaged using a Zeiss LSM880 or Zeiss LSM700 confocal microscope. Within the NAc, images were acquired at the boundary between core and shell (identified anatomically) and include both subregions. In the VTA, images captured were medial to the medial lemniscus. For quantification of MitoGFP recombination within microglia, stacks of confocal images (z-stacks) with a z interval of 1 μm were taken using a 20x objective and imported into ImageJ software for analysis. Maximum intensity projections were counted manually for the % of microglia exhibiting MitoGFP expression. For 3D volumetric reconstruction of microglia and mitochondria, confocal z-stacks were acquired with a 63x objective and z interval of 0.3 μm and imported into Imaris (Bitplane) software for analysis. The surface reconstruction module was used for volumetric reconstruction of mitochondria and microglia within the entire field of view (FOV). Measures of the average volume of individual mitochondria, mitochondrial sphericity (degree of circularity, with 1 representing a perfect circle), and aspect ratio (longest axis of mitochondria / shortest axis of mitochondria) were generated within the Imaris surfaces module and exported for graphing and statistical testing. For FOV-based analyses, two or three images from separate brain sections were analyzed per mouse to obtain an average value for that mouse. 3 to 7 mice were analyzed per brain region, per age and/or per condition. In some cases, further analysis of individual microglia was carried out by selecting Imaris surfaces from an individual cell and the mitochondria

within that cell. In these cases, 4-6 cells were analyzed per mouse. To select individual microglia for full 3D morphological reconstruction in *MG-mitoGFP;Ai14* mice, the brain region of interest was located and centered using only TH staining and anatomical landmarks with 20x objectives. After switching to 63x objectives, TdTom+mitoGFP+ microglia within that field of view were imaged. Slight X-Y adjustments were made, if needed, to bring a TdTom+mitoGFP+ microglial cell away from the edge of the FOV to avoid truncating processes. Cells were excluded from reconstruction if their somas fell near the top or bottom edge of the brain slice rather than being situated near the middle of the z-stack. The Imaris filament tracer module was used for reconstruction and analysis of the branching morphology of individual cells, and surfaces were used to reconstruct mitochondria as described above. 1 or 2 cells were reconstructed per FOV per brain region, yielding 3-6 morphologically reconstructed cells per mouse. The Imaris filament tracer tool was also used for Sholl analysis, where each concentric circle was 1 μm apart. For analysis of mitoGFP signal intensity in *MG-mitoGFP-TFAMKO* and *MG-mitoGFP* control mice, mean fluorescence intensities for each individual, Imaris-reconstructed mitochondria in a field of view were extracted. Mean fluorescence intensity values for all reconstructed mitochondria from one mouse were then pooled ($n = 3$ FOVs per mouse per brain region) and a cumulative distribution plot for that mouse and brain region was generated. Cumulative distribution plots from all mice from one genotype and brain region were then averaged to generate an average cumulative distribution for that group.

### Acute brain slice preparation
*CX3CR1 $^{CreER/+}$; mitoGFP* or *CX3CR1 $^{CreER/+}$; mitoGFP; Ai14* mice at 1.5 to 2.5 months were anesthetized with isoflurane and perfused transcardially with 10 mL of oxygenated, ice-cold N-methyl-D-glucamine (NMDG)-based solution containing the following (in mM): 92 NMDG, 20 HEPES, 30 NaHCO$_3$, 1.2 NaH$_2$PO$_4$, 2.5 KCl, 5 sodium ascorbate, 3 sodium pyruvate, 2 thiourea, 10 MgSO$_4$, and 0.5 CaCl$_2$, 10 glucose, pH 7.4 (310 mOsm). Brains were then rapidly dissected free and horizontal midbrain sections (230 μm thick) and coronal forebrain sections (300 μm thick) were prepared using a vibratome in ice cold NMDG-based cutting solution bubbled continuously with 95% O$_2$ / 5% CO$_2$. Following sectioning, sections were transferred to a holding chamber containing NMDG solution at 34 °C for 5 min and then transferred to artificial cerebrospinal fluid (aCSF) solution at room temperature for 30 min recovery containing the following (in mM): 125 NaCl, 2.5 KCl, 1.25 NaH$_2$PO$_4$•2H$_2$O, 1 MgCl$_2$•6H$_2$O, 26 NaHCO$_3$, 11 Glucose and 2.4 CaCl$_2$•2H$_2$O and bubbled continuously with 95% O$_2$ / 5% CO$_2$.

### Acute brain slice multiphoton imaging
Acute brain slices from NAc and VTA were imaged using a Leica Stellaris 8 Dive, multiphoton microscope equipped with a Coherent Chameleon Ultra II laser and a Leica 25x/1.00 NA W motCORR objective. During video acquisition, acute brain slices were continuously perfused with aCSF bubbled with 95% O$_2$ / 5% CO$_2$. Videos were acquired at least 60 μm below the surface of the slice and video acquisition was carried out within 1–3 h after sectioning. For analysis of microglial mitochondria only in *CX3CR1 $^{CreER/+}$; mitoGFP* mice, an excitation wavelength of 930 nm was used and GFP was detected via a non-descanned detector (464−542 nm). For analysis of microglial mitochondria and microglial cell process motility in *CX3CR1 $^{CreER/+}$; mitoGFP; Ai14* mice, an excitation wavelength of 930 nm was used, and GFP and TdTom signals were detected simultaneously via two non descanned detectors (detector 1: 463−537 nm, detector 2: 558−668 nm). Stacks of images encompassing entire microglial cells (z-step interval 1 μm) were acquired at 0.02 Hz (for mitochondrial motility only) and 0.008 Hz (mitochondrial motility + microglial cell process motility). To induce a focal laser lesion, exposure to high power illumination of 750 nm for 10 sec was applied 60 μm below the surface with maximal zoom in the space between

4-5 GFP+TdTom+ microglial cells. Imaging of microglia and their mitochondria post-lesion was then carried out as described above.

### Analysis of microglial and mitochondrial motility
To analyze microglial and mitochondrial motility, 3Dmorph, a previously published automated analysis pipeline by York et al.[92], was adapted for 2D analysis. Movies were pre-processed in ImageJ. Pre-processing included: background subtraction, median filter, drift correction (MultiStackReg ImageJ plugin), and bleach correction (Histogram matching ImageJ plugin). Individual cells were highlighted by ROI of maximum projections at each time point and the signal outside of ROI was removed. The adapted 3D morph script was applied to the final 2D movie to determine added/subtracted pixels, total pixels changed, and the area of TdTomato (microglia) and GFP (mitochondria) channels. To calculate microglial and mitochondrial motility, the total number of pixels altered during a 10-min interval were normalized to the total number of pixels making up the cell (cell area). For quantification focused on individual microglial branches, preprocessed videos were cropped to isolate entire primary branches of the cell from the point at which the primary branch connects to the cell soma out to the terminal endings of branches emanating from that primary process. Analysis and quantification were then carried out in the same manner as for entire cells. Mitochondrial mass of whole microglia or primary branching processes was calculated as GFP area divided by TdTomato area in final frame. Morphological complexity of microglia was analyzed using Sholl analysis through the Simple Neurite Tracer plugin in ImageJ[42]. Each concentric circle was 1 μm apart, like the morphological analysis conducted in fixed tissue. Values from Sholl analysis of cell branching in the first and last frame of 10-min videos were averaged to obtain an overall assessment of cell complexity throughout the imaging period. To analyze mitochondrial motility in 3D, additional analyses were carried out within Imaris (Bitplane) Software. The surface reconstruction module was used to create surfaces for GFP+ mitochondria in microglial cell processes where individual mitochondria can be easily tracked; microglial somas contain large mitochondrial networks, complicating tracking of individual mitochondria. 4D movement tracks were then generated for each reconstructed mitochondrion. Some manual editing of tracks was performed to ensure accurate tracking of reconstructed mitochondria across time. 3D positional information at the start and end of 10 min videos was used to calculate displacement of individual mitochondria. Individual mitochondrial displacements were averaged per cell to obtain average mitochondrial displacement within individual microglia. The volume for individual mitochondria was averaged across the initial frame and the final frame and volume of all mitochondria in a cell was summed to obtain the mitochondrial volume for each microglial cell.

### Analysis of laser lesion response
Videos were preprocessed in ImageJ for background subtraction, median filter, maximum projection, and thresholding. Using an analysis strategy similar to that employed by others in the field[50,51], circular ROIs were used to create three rings within images at defined times post-lesion (0, 5, 10, and 20 mins), with the first ring enclosing the lesion, the second ring at 15 μm from the first, and the third ring at 30 μm from the first. ROI rings were used as masks to extract pixels within each ring at each time point and the measure function was used to extract ring area and percent of area covered by thresholded pixels. The percent change of pixels within a ring was calculated as initial pixels minus final pixels divided by initial pixels.

### Tissue dissociation and flow cytometry
2-4mo, 12-13mo, and 16-18mo WT, *CX3CR$^{CreER/+}$;mitoGFP* and *CX3CR1$^{CreER/+}$;Tfam$^{fl/fl}$* mice were deeply anesthetized with isoflurane and transcardially perfused with 10 mL of chilled 1 M PBS. The brain was rapidly

removed, cut coronally with a razor blade using anatomical landmarks and midbrain and striatum were dissected free. Samples were thoroughly minced with a razor blade on a chilled glass surface and transferred to 1.7 mL tubes containing chilled 1 mL Hibernate A Low-Fluorescence (BrainBits, Half100). Samples were then mechanically dissociated using gentle sequential trituration with fire-polished glass pipettes of decreasing diameter. Cell suspensions were pelleted via centrifugation and resuspended in 1 M PBS + Debris Removal Solution (Miltenyi, 130-109-398). Debris removal was carried out according to the manufacturer's instructions. Following debris removal, samples were resuspended in small volumes of 1 M PBS and incubated with primary antibodies on ice for 20 min. Tetramethylrhodamine (TMRM) (Biotium, 70017) was used in some experiments to assess mitochondrial membrane potential. For these experiments, samples were incubated with TMRM 50 nM for 20 min at room temperature, prior to antibody incubation. Following incubation, samples were washed with 1xPBS and filtered through a 35 $\mu$m filter (Falcon, 352235) prior to sorting. With the exception of TMRM incubation, throughout the experiment, samples were kept at 4 °C. Samples were sorted using a FACS Aria I cell sorter (BD Biosciences). The population of cells containing microglia could be readily identified based on forward scattering (FSC) and side scattering (SSC) properties. A gating strategy based on FSC and SSC width and height was used to select single cells and reduce dead cells and doublets. Microglial cells within this population were then identified and sorted based on combinations of microglial antibodies. Antibody combinations used included anti-CD45 (1:400; Biolegend, 147703), anti-P2RY12 (1:500; Biolegend, 848005), and anti-CX3CR1 (1:800; Biolegend, 149023). Microglia from CX3CR1[CreER/+];mitoGFP mice were further gated from microglial antibodies to sort GFP+ cells. Placement of gates was verified using unstained samples as well as single color controls and Fluorescence Minus One (FMOs) controls. Gated Microglia were FACS sorted into 1.7 mL tubes containing 50 μl Pico Pure RNA extraction buffer and, following the sort, samples were incubated at 42 °C for 30 min, and stored in RNase-free tubes at −80 °C until further processing. Total sample processing time from brain dissection to completion of FACS purification ranged from 2-4h depending on the sort order of individual samples. Samples were collected in batches with each day of sorting including a young adult, saline injected, or TFAM[+/+] control mouse. Sort order of samples from specific ages, brain regions, treatment conditions, or genotypes was interleaved to avoid introduction of bias related to tissue processing length. FACS-isolation itself was carried out at 4 °C. FACS-isolation of all samples for a given set of experiments was completed prior to downstream processing to allow simultaneous RNA isolation and cDNA preparation as described below.

## Flow cytometry analyses

FCS files from sorting experiments were imported into FlowJo software for additional analyses of flow cytometry data. After placement of gates as described above, median fluorescence intensity measures were exported for populations of interest. Graphing and statistical analyses of MFI values were performed in OriginPro. To examine TMRM values normalized for mitochondrial volume/abundance, TMRM signal for each individual cell in a given sample was normalized to mitoGFP signal for that cell. Median values for this TMRM/mitoGFP measure were then calculated across all cells for a given sample. In some experiments, fluorescence signals across the entire population of cells from one sample were examined via cumulative distribution plots. For these analyses, a cumulative distribution plot was generated for each sample. Cumulative distribution curves from all samples in a group were then averaged to generate an average plot for that group.

## RNA Extraction and qPCR

Total RNA was extracted from sorted cells using Pico Pure RNA isolation kit (Arcturus Bioscience) according to the manufacturer's

instructions. Single strand cDNAs were synthesized using Superscript III first-strand cDNA synthesis kit (Invitrogen, 18080051), according to the manufacturer's protocol. Samples were subjected to PreAmplification-RT-PCR. TaqMan PreAmp Master Mix Kit was used for cDNA preamplification (Cat# 4391128; Applied Biosystems, Life Technologies), using pooled primer mixes of 20x dilution of TaqMan Gene Expression Assay and 80 nM of custom-designed primer sets. cDNAs were pre-amplified in an Veriti 96-Well Fast Thermal Cycler (ThermoFisher Scientific, 4375305) using the program: 95 °C hold for 10 min, 14 cycles of 90 °C denaturation for 15 s, and 60 °C annealing and extension for 4 min. Pre-amplification PCR products were immediately diluted five times with molecular biology grade water and stored at −20 °C, or immediately processed for RT-PCR. For qPCR analysis of gene expression, single plex qPCR assays were performed on technical duplicates using TaqMan probes for each gene target or for endogenous control gene (Gapdh for LPS experiments, Gusb for MG-TFAMKO/MG-mitoGFP-TFAMKO experiments). TaqMan Advanced Fast PCR Master Mix was used (catalog #4444963; Invitrogen). To avoid amplification of any genomic DNA contamination, primers and probes that amplify across target gene exon–exon junctions were selected when possible. qPCRs reactions were run in a QuantStudio5 using the program: 95 °C hold for 20 s, followed by 40 cycles of 95 °C denaturation for 3 s, and 60 °C annealing and extension for 30 s. Calculations of relative expression and expression fold change from Ct data were conducted according to User Bulletin #2 for ABI Prism 7900 Sequence Detection System. For each target gene, the average Ct value for the endogenous control gene (Gapdh/Gusb) was subtracted from the average Ct value for the target gene, to obtain ΔCt. The relative expression was then plotted as $2^{-\Delta\Delta Ct}$. ΔΔCt was obtained by subtracting the average ΔCt of the control group from the average ΔCt of a given sample. Expression fold change was then plotted as $2^{-\Delta\Delta Ct}$. The average ΔCt of the control group was calculated separately for STR and MB samples. All TaqMan Assays and custom primers/probes that were used are detailed in Table S8.

## Statistical analyses

Statistical analysis was performed in OriginPro, Matlab, and R. Heatmaps were generated in R using the ComplexHeatmap package by Gu Z et al[93]. PCAs were generated in R using the ggplot2 package by Wickham H[94]. Robust regression with False Discovery Rate control analyses were performed in R or Matlab. Cumulative distribution plots were analyzed for significant differences using Kolmogorov-Smirnov tests in OriginPro or Matlab. All graphed values are shown as mean ± SEM. Statistical details of experiments (statistical tests used, exact value of N, what N represents) can be found in results and the figure legends. In general, statistical significance was assessed using one- or two-way ANOVA. Post hoc comparisons were conducted using Student's t-test with Bonferroni correction for multiple comparisons. Data are sufficiently normal and variance within groups is sufficiently similar for use of parametric tests. 2 tailed paired t-tests were used to compare trends across two different brain regions within individual mice. Statistical significance was considered if $P < 0.05$. Asterisks were used to delineate statistical significance.

## Reporting summary

Further information on research design is available in the Nature Portfolio Reporting Summary linked to this article.

# Data availability

Source data are provided with this paper. Source data used to prepare all figures in this study have been deposited in FigShare database (https://doi.org/10.6084/m9.figshare.30152542). Raw data files are available to researchers wishing to carry out additional analyses or validation analyses for non-commercial purposes. Access to raw data

files can be obtained by contacting Dr. Lindsay De Biase, ldebiase@mednet.ucla.edu.

## Code availability

Custom MatLab code used for analysis of microglial and mitochondrial motility is available on CodeOcean (https://doi.org/10.24433/CO.1668583.v1).

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

## Acknowledgements

This work was supported by the Diana Jacobs Kalman Scholarship (American Federation on Aging Research scholarship to A.W.S), the Jennifer S. Buchwald Graduate Fellowship (UCLA Department of Physiology fellowship to K.E.), a University of California Office of the President Pre-Professoriate Fellowship (to K.E.), UCLA CARE Summer Undergraduate Fellowship (to C.C.), UCLA CARE Summer Undergraduate Fellowship (to K.M.), David Geffen School of Medicine at UCLA start-up funds (to L.M.D.) and a Stanley Fahn Junior Faculty Award (PF-SF-JFA-836586; Parkinson's Foundation Grant to L.M.D.). We thank the UCLA Eli and Edythe Broad Center of Regenerative Medicine and Stem Cell Research Flow Cytometry Core and the UCLA Jonsson Comprehensive Cancer Center and Center for AIDS Research Flow Cytometry Core for assistance with FACS-based experiments. We thank the UCLA Eli and Edythe Broad Center of Regenerative Medicine and Stem Cell Research Microscopy Core for access to and use of confocal microscopes and Imaris workstations. We thank Dr. Elaine Hsiao for the use of the Hsiao lab QuantStudio5 for qPCR experiments. We thank Dr. Fanny Etienne for assistance with training for multiphoton imaging experiments.

## Author contributions

K.E., A.W.S., and L.M.D. designed research; K.E., A.W.S., A.S., A.E., K.M., M.Y., and C.C. performed research; K.E., A.W.S., D.T.G., A.S., K.M., M.Y., and C.C. analyzed data; K.E., A.W.S., and L.M.D. edited the paper; K.E. and A.W.S. wrote the first draft of the paper; K.E. and A.W.S. wrote the paper; L.M.D. supervised research.

## Competing interests

The authors declare no competing interests.
