## [Transparent Peer Review file · Nature Communications]

Dynamic changes in mitochondria support phenotypic flexibility of microglia

Corresponding Author: Dr Lindsay De Biase

Version 0:

Reviewer comments:

Reviewer #1

(Remarks to the Author)

In this research paper, the authors studied the role of microglial mitochondria in surveillance, response to injury, inflammatory stimulus or aging. They identify region-specific differences in the distribution of microglial mitochondria, together with a correlation of mitochondrial content with microglial morphological complexity. They also show that mitochondrial distribution is stable during microglial reaction to focal lesion, and interestingly, that there is no, or even negative correlation between mitochondrial mass and microglial motility. They also show mitochondrial remodeling at the early stages of microglial response to inflammatory challenge, and morphological/functional changes during aging. Finally, they show that genetic manipulation of microglial mitochondria leads to changes of mitochondrial morphology and gene expression.

Overall, the topic of the paper is important and timely. The authors used largely appropriate experimental models and control measurements. While the main observations made by using microglial MG-mitoGFP and MG-TFAMKO mice concerning the role of mitochondria in microglial phenotype changes and responses are novel, some additional functional studies would be required to provide deeper mechanistic insight into morphological and gene expression changes related to mitochondrial function and TFAM in microglia.

Specific points:

- The authors studied the nucleus accumbens (NAc) and ventral tegmental area (VTA), two brain regions where microglia display distinct cellular phenotypes. While this is a fair comparison to make, most studies assessing microglial morphology and function were performed in the cerebral cortex. The authors should not be requested to replicate all data sets in the cortex, but the statement "these observations indicate that regional differences in microglial cellular phenotype are accompanied by regional differences in mitochondrial metrics" would be highly supported by morphometry comparisons of NAc and VTA microglia with microglia in the cerebral cortex. The same applies to mitochondria numbers that appear to correlate with microglial morphological complexity primarily in the VTA.
- One limitation of this work is that imaging studies were performed in brain slices ex vivo and slice preparation markedly impacts on microglial morphology and responses. The authors are aware of this issue, but the fact that microglial motility (total process remodeling, normalized to cell area) not correlated with time elapsed since preparation of brain sections or number of Sholl intersections, does not resolve the expected changes in cellular complexity of microglia upon slice preparation. Do the authors have data to show how microglial morphologies and mitochondrial networks related to microglial branching structure change in brain slices in different areas compared to the in vivo situation?
- As the authors claim that mitochondrial content is correlated with microglial morphological complexity, it would be interesting to see whether microglial 3D morphology would be altered after deletion of mitochondrial transcription factor A (TFAM). Also, it would be good to test the possible influence of TFAM downregulation on microglial process motility and microglial reaction to focal injury. The results of these experiments would greatly enhance our understanding of the role of microglial mitochondria.

- The authors show that changes in microglial function genes and those for some inflammatory mediators are region specific after TFAM deletion. While these data sets are interesting, in functional terms it would be more important to show how TFAM deletion influences mitochondrial genes and inflammation after LPS injection and how these changes correlate with microglial morphological states.

Reviewer #2

(Remarks to the Author)

Espinoza and colleagues use a similar approach with mitochondria-targeted GFP to profile mitochondrial dynamics in microglia from multiple brain regions in mouse acute brain slices in the steady state, in the course of laser lesions, normal aging, in response to LPS, and following prolonged knockout of the mitochondrial transcription factor TFAM. The authors reveal many interesting insights, including regions variability in mitochondrial density and morphology, a lack of mitochondrial dynamics during process extension, and increased mitochondrial number in aging. While these findings are valuable to the field, it should be noted that they are descriptive, and causal links between microglial function and mitochondrial dynamics cannot be made from these data. However, the authors are generally careful to acknowledge this point. Additionally, transcriptomic experiments are fraught with potential issues in experimental design and execution, and would be significantly improved with the suggestions laid out below.

A major concern is lack of proper RNA seq from TFAM null microglia. TFAM is known to control mtDNA-cGAS-Sting (see work of West and Shadel). Any insight into whether perturbing TFAM is affecting this pathway is necessary and important for the field.

• Major comments

o The choice to forgo tamoxifen in many experiments with the Cx3cr1-CreER is questionable. While leaky expression is indeed observed with this driver, and this can, in theory, lead to mosaic expression, it is very unclear if this yields an unbiased representation of microglia. Reconstruction of these cells from animals treated with tamoxifen also should not be prohibitive due to crowding. As autofluorescence in green channels is a genuine concern in aged mice or in cases of neurodegeneration, the value of this method comes into question if tamoxifen truly cannot be used. Figure S1 should include: 1) a representative gating strategy for tamoxifen-treated mice using the same voltages and instrument as in Fig. S1D (where higher GFP expression may be observed), and 2) quantification of GFP+ microglia from tamoxifen treated and untreated animals to show even representation. Alternatively, these cells could be FACS sorted from treated and untreated mice to show that there is no bias toward different microglial states in the untreated mice.

o The authors should acknowledge that the Cx3cr1-CreER driver as well as IBA1 staining also targets border-associated macrophages which are not easily visually distinguished from microglia.

o Correlations which are nonsignificant ($P > 0.05$), such as in S2H-K should be clearly described as being nonsignificant. In these cases, and in S3D-F, p-value correction, such as FDR correction, should also be applied to account for multiple comparisons. Likely none of these associations will still be significant after correction.

o Figure 4 is largely unconvincing, perhaps as a result of incomplete microglia purification. Is the entire FACS gating strategy for microglia shown in Fig. 4A? If so, ignoring FSC and SSC has almost certainly introduced a great deal of myelin debris into the final lysate. Additionally, far too many cells (the vast majority) appear to be CD45+, which is extremely unlikely unless pre-enrichment was performed (not mentioned in methods). The authors should show the complete gating strategy in a supplemental figure and should include representative gating for unstained controls and fluorescence-minus-one (FMO) controls to ensure that these antibodies worked correctly and voltages were set to reasonable values. Furthermore, were voltages set correctly, microglia should be CD45 intermediate, not CD45 high. The P2RY12+, CD45int population seen outside the gate may actually be true microglia. This does not appear to be the same scheme shown in Fig. S1D, which is far more rigorous.

As a result, upregulation of a number of pro-inflammatory genes, including the pro-inflammatory cytokine genes Il1b and Tnf is very modest, just one standard deviation above the mean at 4hrs post-LPS, and indistinguishable from saline controls at 24hrs. With the exception of Cd45 and Itgam, all gene comparisons are <2 SD. Relative expression should be shown as fold-increases as is the standard for qPCR, which will likely highlight these modest changes, and all samples should be shown. Curiously this is performed in Fig. 8 (albeit strangely), so clearly the authors are aware of this convention. Normalization to Gapdh may have been an issue here, given the potential metabolic shifts observed in these samples. Finally, improper statistics were performed. ANOVAs were performed without multiple testing correction, and no post-hoc analysis was performed.

PCA and correlation plots are largely uninformative given the small number of genes sampled

All of these issues would likely be remedied with a well-designed RNA-seq experiment on purified microglia using the sorting scheme shown in Fig. S1D.

o While the motivation to ablate TFAM in microglia is not clear from Fig. 4, it is still a very interesting experiment for the field, as a whole. The finding of increased coverage of microglia in Fig. 7I is very interesting and somewhat unexpected but would be bolstered significantly by counting individual microglia using either the existing images or flow cytometry to determine if these is proliferation or increases in size or process extension.

o For Fig. 8 to be considered reliable, Tfam must be shown to demonstrate effective knockout and also microglia purity. Unlike in Fig. 4, $\Delta\Delta Ct$ values are shown, rather than Z-scores, but again fold-change would be much more interpretable. Here again the use of Gapdh as a normalization control is questionable given the direct targeting of metabolism with Tfam knockout. Again, this experiment would be significantly improved with a well-designed RNA-seq experiment on purified microglia using the sorting scheme shown in Fig. S1D.

• Minor comments

o If the authors have set alpha to 0.05 for these experiments, $P = 0.05$ should not be marked as significant (see Fig. 1L)

- o Figure callouts frequently occur out of order. Panels should appear in the order they are mentioned in the text. Similarly, some panels appear to never be referenced and should be either mentioned or removed, e.g. S2H, S2J-K.
- o The use of multiple Y-axis scales (as in S2F-G) is not ideal data visualization and should be avoided. Faceting or simply presenting a scatter plot is far more interpretable.
- o All heatmaps should show all samples, rather than just the group average.
- o Flow cytometry data should be displayed as contour plots with outliers shown. At present, with scatter plots, it appears that many populations are split arbitrarily. While this may not be the case, this is important for showing the true distribution.
- o Data presentation in Figs. 6F and 7J is nonstandard and makes use of pseudoreplication by treating each cell or mitochondrion as an independent measurement. Instead, comparisons of median fluorescence intensity for each mouse / region should be displayed (ideally as boxplots) and compared statistically. These findings are potentially very interesting and proper analysis would better highlight them.

Reviewer #3

(Remarks to the Author)

Reviewer #4

(Remarks to the Author)

The manuscript presents valuable findings about the location of mitochondria in microglia and will, therefore, have some implications for the microglia community. The results align with previous *in vivo* microglia data analysis (Maes et al. 2023, cited by the authors) from a different mitochondria labeling mouse model using fixed and live *in vivo* imaging in the nucleus accumbens (NAc) and the ventral tegmental area (VTA). The data presented is solid and broadly supports the claims. The most compelling data is that dynamic microglia processes often lack mitochondria, which is counterintuitive to likely energy-demanding microglial processes and the concept of mitochondria as an energy source. The manuscript contains a lot of experiments yet does not go beyond this observation. The manuscript remains mainly descriptive of the microglia and mitochondria phenotypes. The rationale for exploring different environmental alterations and their impact on the microglia-mitochondria network is not straightforwardly defined, leaving the impression of a phenotype search.

Introduction:

The introduction needs to provide more confidence and perspective on the goal of the manuscript. Although the first paragraph introduces microglia and their ability to shift between functional states, the concept of a microglia state must be defined. The second paragraph introduces the expression of different cell surface receptors, finishing with a question of how microglia interpret different signals. From the third paragraph, it is not intuitive how mitochondria as “cell attributes” are involved in the previously introduced perspective of functional states and interpreting different signals. What impact would the mitochondrial network remodeling have on the microglia? Furthermore, if the authors consider “cell attributes” as intrinsic properties of microglia to define their functional state, then this should be clearly outlined and justified. Lysosomes, for example, would be a more intuitive “attribute” to distinguish the different microglial states in the conditions tested within this manuscript compared to mitochondria.

In addition, the introduction across all three paragraphs consists mainly of attribute lists not followed up in the manuscript, such as high-fat diet, pollution exposure, and TLR4 and TGFBR2. Also, the list of mitochondrial parameters needs to be more focused since the study does not evaluate buffering calcium, metabolites, or interactions with other organelles.

Overall, the introduction only loosely connects mitochondria distribution and potential microglia state and does not provide a clear rationale for justifying the study.

Results:

Fig 1/S1-2:

i) The authors performed a detailed verification of their mouse model to ensure that mitochondria were labeled. This data set looks convincing. However, without ultrastructural analysis, it cannot be ensured that MG-mitoGFP mice “accurately renew the *in vivo* distribution of microglia mitochondria.” If the authors seek this accuracy as they write, they must perform ultrastructural analysis.

ii) Since the main text states that mitoGFP recombination was substantial without tamoxifen injection, is the data in Fig. S1 without tamoxifen injection, while Fig. 1 with tamoxifen? For clarification, the authors should annotate in the figure legend if they have administered tamoxifen. If Fig. S1 is without tamoxifen, the authors should validate the model again with tamoxifen since tamoxifen could potentially affect the mitoGFP distribution.

Furthermore, for Fig. 1F, the analysis has been done without tamoxifen for rigorous analysis. This is surprising since microglia coverage across tissue is not overlapping, and confocal images should be sufficient to resolve the mitochondria signature for each microglia.

iii) How relevant is the correlation between mitochondrial number and microglia complexity/number of intersections? Are these just larger cells with more mitochondria? Based on the data in Fig 1E, the data point variation for each region is small. Furthermore, how have the microglia for reconstruction been selected?

What is the rationale for choosing the “maximum number of Sholl intersections”? Which radius size has been used? This

can severely impact the readout. Furthermore, was this maximum always at a similar distance from the soma, or has there been some variability? This could explain potential variability in the data. The same question applies to Fig. S3E. The value used for sholl analysis in Fig. S3D is also unclear.

iv) The choice for looking at the NAc and VTA needs better clarification, also in perspective for the later analysis with LPS and aging. Are those brain regions particularly vulnerable?

v) Based on the figure legend description in Fig 1. It is understood that microglial volume is calculated per field of view (tissue coverage)- and mitochondrial number, volume, and mass are then calculated based on this FOV microglial volume. However, the text states, “greater number of mitochondria relative to cell volume, as well as a greater total mitochondrial volume relative to cell volume (mitochondrial mass)”. If the quantification is relative to total microglial volume per FOV, and not on a per-cell basis, stating ‘cell volume’ is confusing. For example, Fig. 1D-E, is the relative ‘microglia volume’ in 1D the same denominator as ‘cell volume’ in 1E? In addition, does the number of mitochondria within a FOV bring us important information? Since many microglial processes not connected to a cell body are present within a brain volume, these averages do not indicate information on a per-cell basis. The authors should identify precise nomenclature and analysis methods, describe this in detail in the methods, and use the terminology consistently throughout the manuscript.

vi) The mitochondrial field focuses more on measuring network connectivity, i.e., fragmented or connected networks. This metric is not shown throughout the manuscript but would be an essential feature to describe when considering mitochondrial structural network changes.

Fig 2:

i) The conclusion of Fig. 2C needs to be clarified. If VTA and NAc show the same motility, how can the conclusion be made that the VTA is surveying less based on their morphology?

ii) Do the motile processes contain mitochondria? Or are motile mitochondria only found in primary processes that are more stable? (also related to Fig. 3).

Fig 3:

i) This data is very interesting and raises exciting new questions; however, no quantification supports the concluding statement.

ii) At this resolution, it seems difficult to quantify and confirm that the mitochondrial networks remain stable after focal injury. Can the authors image at the resolution seen in Fig 2 and provide data analysis of mitochondrial network motility before and after injury? Do all mitochondria stop moving, or only a few travel within the extending process?

Fig 4/5:

i) If mitochondria are not linked to microglial tissue surveillance and injury, as the authors conclude in Fig. 2J and Fig. 3, respectively, how do they rationalize using LPS? Based on live imaging, mitochondria are not linked in a “direct fashion to baseline surveillance” or injury-induced damage. Why would it suddenly be related to an inflammatory response induced by LPS?

ii) qPCR of Dnml1 is used to indicate fission- however, this data should also be reflected in the image analysis in Fig 5. Could the authors indicate changes in the median volume of organelles (per cell) to compare alongside the mitochondrial number and aspect ratio? If the mitochondria are indeed elongated as indicated by an aspect ratio measure, the median volume of organelles per cell should increase (aligning with a mass increase) since the number of mitochondria did not change between saline and LPS for both brain regions.

iii) Do the authors have the details of the post-hoc analysis of 5B-D?

iv) Technical aspect related to sampling for flow cytometry. The method section must clarify whether the samples were processed sequentially or in batches. Furthermore, information related to the total time from tissue removal to sorting must be included. It has been shown that the microglia transcriptional profile adapts within 30 minutes of incubation on ice (Marsh 2022. DOI: 10.1038/s41593-022-01022-8). This can result in misleading results. Therefore, the authors should add the animal grouping strategy and total processing time to the methods and, if necessary, repeat the experiments when the timing is longer than 30 min.

v) How was the correlation analysis performed in Fig. 4D? Cherry-picked genes were connected to putative microglial states, an aspect that is also not well-defined, introducing a bias. Also, the explanation of the “few CD68 correlations with other genes” does not seem solid (Fig. S4B) since the heatmap shows much more correlation than any of the other plots in Fig. 4.

Fig 6:

Microglia are known to show a sex-specific response also in adulthood and, especially, during aging. Therefore, the authors must precisely annotate the data plots, indicating which data points come from male or female animals.

Fig 7:

i) Related to the mitochondrial quantification in 7D: Is this data represented as an average volume of all organelles within the cell? If yes, it would be better to use a median, as small mitochondria can skew the average value and misrepresent the

data.

ii) Mitochondrial analysis: In Fig. 7, the analysis focused on volume, number, and sphericity, whereas earlier figures describing the mitochondria in fixed tissue focused on mass, number, aspect ratio, volume, sphericity. To allow comparison between the datasets, the authors should be consistent and include all metrics for all figures. Furthermore, the methods section needs to include a detailed description of the analysis method for each parameter.

Fig 8:

How do the microglia change after TFAM KO? A two-month waiting period may allow time for compensatory mechanisms to circumvent initial damage.

Discussion:

In line with the loose introduction, the discussion is also only superficial. It mostly repeats the results without putting the research data into the context of the literature. For example, a reported result is increased mitochondrial mass. What does increased mitochondrial mass imply? The manuscript would enormously benefit from a complete revision of the introduction and discussion to make it coherent.

Overstatements:

The manuscript suffers from several overstatements and preliminary conclusions, which must be fixed. A few examples are listed below:

i) "First time" in the introduction.

ii) The authors regularly write "in vivo" within their manuscript, yet it becomes quickly apparent that the majority of the experiments are performed in a fixed tissue environment. This should be clarified in the text to avoid misunderstandings.

Reviewer #5

(Remarks to the Author)

Version 1:

Reviewer comments:

Reviewer #1

(Remarks to the Author)

No further questions.

Reviewer #2

(Remarks to the Author)

It is much improved.

They should cite and discuss their findings with other previous studies on mitochondria and microglia.

PMID: 36787364

PMID: 39048800

PMID: 38480879

PMID: 39048801

Reviewer #3

(Remarks to the Author)

Reviewer #4

(Remarks to the Author)

Overall, the revised manuscript is significantly improved in clarity, making it accessible to a broader readership. It offers insights into the role of mitochondria in microglia across a range of physiological, pathological, and aging conditions, providing an orientation for future studies in this direction.

Our specific concerns have been adequately addressed, with only a few minor points remaining.

Our comments for each argument are uploaded as .pdf in green.

[Editorial Note: See end of file]

Reviewer #5

(Remarks to the Author)

REVIEWER COMMENTS

Reviewer #1 (Remarks to the Author):

In this research paper, the authors studied the role of microglial mitochondria in surveillance, response to injury, inflammatory stimulus or aging. They identify region-specific differences in the distribution of microglial mitochondria, together with a correlation of mitochondrial content with microglial morphological complexity. They also show that mitochondrial distribution is stable during microglial reaction to focal lesion, and interestingly, that there is no, or even negative correlation between mitochondrial mass and microglial motility. They also show mitochondrial remodeling at the early stages of microglial response to inflammatory challenge, and morphological/functional changes during aging. Finally, they show that genetic manipulation of microglial mitochondria leads to changes of mitochondrial morphology and gene expression.

Overall, the topic of the paper is important and timely. The authors used largely appropriate experimental models and control measurements. While the main observations made by using microglial MG-mitoGFP and MG-TFAMKO mice concerning the role of mitochondria in microglial phenotype changes and responses are novel, some additional functional studies would be required to provide deeper mechanistic insight into morphological and gene expression changes related to mitochondrial function and TFAM in microglia.

We thank the reviewer for recognizing the timely contributions of our study to the field and for agreeing that experiments are well designed. We also fully agree that additional functional studies that probe mechanistic links between discrete mitochondrial properties and specific microglial attributes are a critical future direction for our work and the field more broadly. We hope the reviewer will agree that, with our additional experiments and analyses, this study provides an essential foundation for that future work.

Specific points:

- The authors studied the nucleus accumbens (NAc) and ventral tegmental area (VTA), two brain regions where microglia display distinct cellular phenotypes. While this is a fair comparison to make, most studies assessing microglial morphology and function were performed in the cerebral cortex. The authors should not be requested to replicate all data sets in the cortex, but the statement “these observations indicate that regional differences in microglial cellular phenotype are accompanied by regional differences in mitochondrial metrics” would be highly supported by morphometry comparisons of NAc and VTA microglia with microglia in the cerebral cortex. The same applies to mitochondria numbers that appear to correlate with microglial morphological complexity primarily in the VTA.

The reviewer rightly points to the utility of analyzing additional brain regions to better place our findings from the VTA and NAc in context.

→ To respond to this concern, we used triple transgenic *MG-mitoGFP;Ai14* mice to analyze microglia and their mitochondria in prefrontal cortex (Fig. 1G-J, and Fig. S2G,J). These triple

transgenic mice allow for rigorous reconstruction of the branching morphology of individual microglial cells, together with reconstruction of that cell's mitochondria. This analysis revealed that morphological complexity of PFC microglia (total process length and total branch points) was similar to that of VTA microglia. Morphological complexity of NAc microglia was elevated compared to both of these other regions. In terms of the relationship between morphology and mitochondria, the abundance of mitochondria in PFC microglia peaked at the same distance from the soma as the max number of Sholl intersections (**Fig. S2G**), similar to what was observed for VTA and NAc microglia (**Fig. S2H,I**). The correlation between mitochondrial and cell morphological parameters continued to be most evident in VTA microglia, with one significant association ($P = 0.02$) and three nearly significant associations ($P = 0.05$, $P = 0.07$, $P = 0.08$, **Fig. S2J**). The correlation between mitochondrial and cell morphological parameters was not as prominent in PFC or NAc microglia. PFC microglia showed one significant association ($P = 0.02$) and one nearly significant ($P = 0.05$). NAc microglia showed one significant association ($P = 0.03$), and two nearly significant associations ($P = 0.08$, $P = 0.08$, **Fig. S2J**). All together, these data suggest that there are links between microglial mitochondria and baseline morphological features of these cells, with some regional variation in the strength of this association.

We would also like to refer the reviewer to a study by Pietramale et. al (BioRxiv 2024.10.15.618505) that was co-submitted with our study. In this work, the authors carry out a number of similar analyses within somatosensory cortex, providing additional contextual information for whether what we observe in NAc and VTA is similar or distinct from what is observed in cortex.

- One limitation of this work is that imaging studies were performed in brain slices *ex vivo* and slice preparation markedly impacts on microglial morphology and responses. The authors are aware of this issue, but the fact that microglial motility (total process remodeling, normalized to cell area) not correlated with time elapsed since preparation of brain sections or number of Sholl intersections, does not resolve the expected changes in cellular complexity of microglia upon slice preparation. Do the authors have data to show how microglial morphologies and mitochondrial networks related to microglial branching structure change in brain slices in different areas compared to the *in vivo* situation?

We are glad that our awareness of the limitations of acute brain slice preparations is evident to the reviewer. The reviewer is correct that microglia in acute brain sections exhibit simplified morphology compared to that seen *in vivo*. This is widely known in the field and is evident in direct comparisons of Sholl intersections analyzed in fixed tissue compared to in acute brain sections (**Reviewer Figure 1**, see also PMID 28689984 with fixed tissue reconstructions of VTA/SNc/SNr microglia in Fig. 2C and dye filled VTA/SNc/SNr microglia from acute brain sections in Fig. S2I). Despite this reduced complexity, core features of microglial cell process motility are preserved in acute brain sections. This includes ability to respond to repeated changes in neuronal activity (PMID 25100586), chemotaxis toward point sources of ATP (PMID 25100587, PMID 29290552), and impact of potassium on ramification and tissue surveillance (PMID 29290552). In the present study, our objective is to gain initial insight into whether features of microglial mitochondria are clearly linked to motility of these cells. Given that correlations between microglial branching and mitochondrial volume are still evident in acute

brain sections (Fig. S3D) and that microglial and mitochondrial motility are not significantly correlated with time since brain slice preparation (Fig. S3E,F), we feel that this preparation is appropriate for initial investigations of the relationship between microglial motility and their mitochondria.

→ To address the reviewer's concern, we modified the wording of the results to say "Although microglia show more simplified branching in acute brain sections, total mitochondrial volume and total number of microglial Sholl intersections were positively correlated (Fig. S3D), indicating that alignment of mitochondria with morphological architecture of the cells was preserved in acute brain sections."

- As the authors claim that mitochondrial content is correlated with microglial morphological complexity, it would be interesting to see whether microglial 3D morphology would be altered after deletion of mitochondrial transcription factor A (TFAM). Also, it would be good to test the possible influence of TFAM downregulation on microglial process motility and microglial reaction to focal injury. The results of these experiments would greatly enhance our understanding of the role of microglial mitochondria.

We completely agree that assessment of microglial cell process motility and responses to laser lesion in the context of microglial TFAM knockout would be extremely useful analyses to carry out. At present, we do not have transgenic lines that enable microglial TFAM knockout *and* live imaging of microglia. This would require generation of $CX3CR1^{CreER/+};TFAM^{fl/fl};Ai14$ mice or similar, making these experiments a key future direction but beyond the scope of the present study.

→ To address the reviewer's first point, we carried out additional analysis of microglial attributes following TFAM knockout in fixed tissue. Density of microglia in NAc and VTA was not significantly altered by loss of TFAM (Fig. 7G). We also carried out Sholl analysis of individual microglia (Fig. 7I) and found a significant increase in total number of intersections in TFAM KO compared to control mice, particularly in the NAc. These findings support the conclusion that a change in mitochondrial status is accompanied by a change in morphological attributes of microglia. Moreover, this *increase* in morphological complexity upon TFAM loss argues against the idea that this manipulation simply makes the microglia "sick" or "reactive."

- The authors show that changes in microglial function genes and those for some inflammatory mediators are region specific after TFAM deletion. While these data sets are interesting, in functional terms it would be more important to show how TFAM deletion influences

mitochondrial genes and inflammation after LPS injection and how these changes correlate with microglial morphological states.

Understanding how mitochondrial state regulates key microglial functions, including their responses to pathological challenges, is absolutely an important future direction for this work and for the field.

→ As the reviewer suggested, we have treated MG–TFAM-KO mice and controls with LPS and used a combination of flow cytometry and qPCR to analyze the microglial response to this inflammatory challenge. In our initial FACS analyses of microglial responses to LPS injection, we noted a substantial increase in intensity of the mitoGFP signal in both striatum (STR) and midbrain (MB) microglia (Fig. 4F). This increase in mitoGFP was still observed in TFAM KO mice at 12hrs post-LPS injection (Fig. 8D), suggesting that loss of TFAM does not prevent microglia from mounting a substantial mitochondrial biogenesis response to LPS challenge. In our initial qPCR analysis of responses to LPS, 10/14 mitochondria-relevant genes showed significant changes in expression level following LPS (Fig. 4B). The majority of these mitochondria-relevant genes behaved similarly in this new set of LPS experiments irrespective of genotype, suggesting, again, that microglial capacity to engage in mitochondrial remodeling in response to LPS challenge is largely intact in TFAM KO mice. Only 5 mitochondria-relevant genes (*Cox4i1*, *Vdac*, *Micu*, *Mfn1*, and *Mavs*) showed significant differences across genotype in the context of LPS challenge. In most cases TFAM KO microglia showed the same *type* of change in expression as control microglia following LPS treatment, and simply differed in the *magnitude* of up- or down-regulated expression compared controls.

Perhaps somewhat predictably, as capacity to remodel mitochondria is largely intact, changes in microglial attributes in response to LPS were also largely similar across MG-TFAMKO mice and controls. Inflammatory factors such as *Tnf α* and *Il1 β* were elevated following LPS across brain regions and genotypes. Decreases in *Cd68* were slightly more muted in KO mice, as were decreases in *Itgam* (CD11b). When taking expression of all analyzed genes into account, PCA analysis shows that LPS is the strongest driver of cell phenotype (driving variance on PC1) and genotype is a secondary determinant of variance on PC2. These observations suggest that the impact of TFAM loss on microglial responses to inflammatory challenge are subtle and might be more apparent in the context of repeated or chronic inflammatory challenges.

It is worth remembering that TFAM KO alters mitochondrial state via impacting the mitochondrial DNA. Out of 96 genes that code for components of the electron transport chain (ETC), 13 reside in the mtDNA rather than nuclear DNA. Altered ETC function as a result of TFAM KO can impact mitochondrial respiration, mitochondrial calcium buffering, and reactive oxygen species production, among other mitochondrial attributes. This broad impact on multiple aspects of mitochondrial state is precisely what made this manipulation attractive as a tool for probing links between mitochondria and microglial phenotypic attributes. However, TFAM KO will not impact all aspects of mitochondrial state and may exert only mild effects on mobilization of nuclear-encoded mitochondrial genes, particularly during an acute pathology response. As a field, we ideally need to generate synergistic information about how distinct manipulations of mitochondrial function regulate microglial attributes and functions in physiological and pathological contexts.

Reviewer #2 (Remarks to the Author):

Espinoza and colleagues use a similar approach with mitochondria-targeted GFP to profile mitochondrial dynamics in microglia from multiple brain regions in mouse acute brain slices in the steady state, in the course of laser lesions, normal aging, in response to LPS, and following prolonged knockout of the mitochondrial transcription factor TFAM. The authors reveal many interesting insights, including regions variability in mitochondrial density and morphology, a lack of mitochondrial dynamics during process extension, and increased mitochondrial number in aging. While these findings are valuable to the field, it should be noted that they are descriptive, and causal links between microglial function and mitochondrial dynamics cannot be made from these data. However, the authors are generally careful to acknowledge this point. Additionally, transcriptomic experiments are fraught with potential issues in experimental design and execution, and would be significantly improved with the suggestions laid out below.

We thank the reviewer for the recognition that our work provides valuable insights for the field. We have carried out additional experiments, additional analyses, and made changes to the figures to try and address concerns about qPCR-based analyses.

A major concern is lack of proper RNA seq from TFAM null microglia. TFAM is known to control mtDNA-cGAS-Sting (see work of West and Shadel). Any insight into whether perturbing TFAM is affecting this pathway is necessary and important for the field.

This is a valuable suggestion, and we have added qPCR of cGAS (*Mb21d1*) and Sting (*Tmem173*) to the panel of genes being analyzed following TFAM KO (Fig. 8B). cGas differed significantly across brain regions (higher in midbrain, $P = 0.03$, Table S5) but did not differ across TFAM KO vs control samples. Sting showed trends toward downregulation in KO microglia from both brain regions, but these did not reach significance ($P = 0.416$, Table S5).

- Major comments

- o The choice to forgo tamoxifen in many experiments with the Cx3cr1-CreER is questionable. While leaky expression is indeed observed with this driver, and this can, in theory, lead to mosaic expression, it is very unclear if this yields an unbiased representation of microglia. Reconstruction of these cells from animals treated with tamoxifen also should not be prohibitive due to crowding. As autofluorescence in green channels is a genuine concern in aged mice or in cases of neurodegeneration, the value of this method comes into question if tamoxifen truly cannot be used. Figure S1 should include: 1) a representative gating strategy for tamoxifen-treated mice using the same voltages and instrument as in Fig. S1D (where higher GFP expression may be observed), and 2) quantification of GFP+ microglia from tamoxifen treated and untreated animals to show even representation. Alternatively, these cells could be FACS sorted from treated and untreated mice to show that there is no bias toward different microglial states in the untreated mice.

We appreciate the reviewer's desire to ensure that analyzed cells are accurately representative for the overall microglial cell population. Given that our initial histology and FACS-based quantification of mitoGFP positive microglia indicated that greater than 80% of the cells were expressing mitoGFP in the absence of tamoxifen injection, we felt confident that we were not analyzing a limited subset of microglia and that results should be representative of the microglial population as a whole.

→ At the reviewer's suggestion, we have carried out additional experiments to treat mitoGFP mice with 4-hydroxytamoxifen (4HT) and compare microglial properties from 4HT- and oil-injected mice. FACS-based analyses indicate that 4HT injection increased the % of recombined cells from ~85% to ~94% (Fig. S1D-G). Reviewer Figure 2 also confirms that FSC (cell size), SSC (internal complexity), mitoGFP, TMRM, CX3CR1, CD45, and P2RY12 median fluorescence intensities (MFI) are not different between microglia from 4HT-injected and oil-injected mice. Based on these additional analyses, we feel that analysis of microglia and their mitochondria in untreated mice does not yield a biased assessment of these cells or their organelles.

→ To address the reviewer's comment about autofluorescence in aging mice (which is indeed a concern for accurate analysis of microglial properties and functions), we added representative images to Fig. S6A. These images illustrate that our tissue processing with True Black eliminates the majority of autofluorescent signals from tissue of aging mice. Residual dim autofluorescence in the somas of some microglia colocalize with CD68, and mitoGFP signals are clearly distinguished above these residual signals.

o The authors should acknowledge that the Cx3cr1-CreER driver as well as IBA1 staining also targets border-associated macrophages which are not easily visually distinguished from microglia. **We have added the following statement to the discussion:**

“Border associated macrophages (BAMs) can also be labeled by Iba1 immunostaining and are expected to experience recombination and mitoGFP expression in *CX3CR1^{CreER/+};mitoGFP* mice. As BAMs are primarily concentrated in the meninges and vascular interfaces and are significantly less abundant than microglia, we are confident that our results are focused on parenchymal microglia. Nonetheless, future studies employing markers of BAMs can explicitly analyze the mitochondria of BAMs and identify similarities and differences of BAM and microglial mitochondria and how these organelles regulate cell function.”

o Correlations which are nonsignificant ($P > 0.05$), such as in S2H-K should be clearly described as being nonsignificant. In these cases, and in S3D-F, p-value correction, such as FDR correction, should also be applied to account for multiple comparisons. Likely none of these associations will still be significant after correction. **We thank the reviewer for helping to ensure rigor of our analyses. We have followed the reviewer’s request to add “n.s.”** to more clearly indicate linear regression analyses where no significance was present. We also re-analyzed all linear regressions in the manuscript using robust regression with false discovery rate control and now report statistics based on these analyses.

o Figure 4 is largely unconvincing, perhaps as a result of incomplete microglia purification. Is the entire FACS gating strategy for microglia shown in Fig. 4A? If so, ignoring FSC and SSC has almost certainly introduced a great deal of myelin debris into the final lysate. Additionally, far too many cells (the vast majority) appear to be CD45⁺, which is extremely unlikely unless pre-enrichment was performed (not mentioned in methods). The authors should show the complete gating strategy in a supplemental figure and should include representative gating for unstained controls and fluorescence-minus-one (FMO) controls to ensure that these antibodies worked correctly and voltages were set to reasonable values. Furthermore, were voltages set correctly, microglia should be CD45 intermediate, not CD45 high. The P2RY12⁺, CD45^{int} population seen outside the gate may actually be true microglia. This does not appear to be the same scheme shown in Fig. S1D, which is far more rigorous.

We thank the reviewer for pointing out aspects of this figure that could be confusing or misleading. As described in the methods of the original submission, our tissue dissociation workflow includes use of Miltenyi debris removal solution, which substantially reduces debris (including myelin debris) in tissue suspensions. As described in the methods, FSC and SSC gates are then used to isolate the population likely to contain cells. This is further refined using gates for FSC-H and FSC-W as well as SSC-H and SSC-W to reduce dead cells and doublets. The Fig. 4A panel in the original submission was only meant to be a schematic to indicate that FACS-purification of microglia was employed. However - as the reviewer indicates - this could be misinterpreted as being meant to illustrate the full gating strategy used.

→ To respond to this concern, we have replaced the gating “schematic” in Fig. 4A with a more general cartoon of FACS sorting.

→ We have added text throughout the results to clarify that the gating strategy shown in Figure S1 is being used. For example, the description of FACS experiments in Fig. 4 now reads “Young adult (3-4mo) *MG-MitoGFP* mice were given i.p. injections of saline or LPS and microglial gene- and protein- expression were analyzed via FACS, using the previously described

gating strategy to identify and isolate microglia (Fig. S1D), followed by downstream qPCR (Fig. 4A).”

→ We have added the following statement to the methods “Placement of gates was verified using unstained samples as well as single color controls and Fluorescence Minus One (FMOs) controls.”

As we are already stretched for space in supplementary figures, we feel it should be sufficient to describe these standard approaches and controls in the methods. We also include a representative example in Reviewer Figure 3.

Regarding observed CD45 expression levels - we perfuse animals prior to tissue dissociation and observe very few CD45 high macrophages using the gating strategy described above. We routinely use this and similar FACS gating strategies in the lab and have carried out RNAseq analysis on isolated cells for multiple projects. Mining data from one of those projects (PMID 28689984) illustrates that cells isolated using our workflows express high levels of microglial markers and low levels of macrophage markers (Reviewer Figure 4).

Reviewer Figure 3. Representative examples of FMO controls used to set microglial gating strategy and analyze mitochondrial mitoGFP intensity and mitochondrial membrane potential (TMRM). These controls are included and used in every FACS experiment.

Reviewer Figure 4. Microglial identity of purified cells. RNAseq of microglia FACS purified from cortex (Ctx), nucleus accumbens (NAc), ventral tegmental area (VTA), and substantia nigra (SN), from 2mo *CX3CR1^{EGFP/+}* mice. We use these mice to validate all FACS antibodies and gate placement used to purify microglia. Cells purified using these workflows express high levels of microglial specific genes and most macrophage specific genes are well below the 1 RPKM threshold considered to indicate reliable transcript detection in this dataset.

§ As a result, upregulation of a number of pro-inflammatory genes, including the pro-inflammatory cytokine genes *Il1b* and *Tnf* is very modest, just one standard deviation above the mean at 4hrs post-LPS, and indistinguishable from saline controls at 24hrs. With the exception of *Cd45* and *Itgam*, all gene comparisons are <2 SD. Relative expression should be shown as fold-increases as is the standard for qPCR, which will likely highlight these modest changes, and

all samples should be shown. Curiously this is performed in Fig. 8 (albeit strangely), so clearly the authors are aware of this convention. Normalization to Gapdh may have been an issue here, given the potential metabolic shifts observed in these samples. Finally, improper statistics were performed. ANOVAs were performed without multiple testing correction, and no post-hoc analysis was performed.

We thank the reviewer for the detailed assessment of our qPCR data. First, as the reviewer can hopefully appreciate, we are attempting to probe the relationship between microglia (as cells - represented by expression of genes and proteins associated with key microglial functions) and their mitochondria (dynamic organelles - represented by expression of genes and proteins associated with diverse mitochondrial functions), in the context of an evolving response to inflammatory challenge. There are numerous valid ways to carry out and present these analyses. In specific sets of experiments, we have chosen analysis and display strategies that we feel most effectively highlight the central findings of the data set. For example, by using heat maps and PCA plots in **Figure 4B** and **C**, we aim to highlight the overall temporal dynamics of gene expression changes during LPS response, rather than focusing in depth on individual genes. Presenting the qPCR values as relative expression ($2^{-\Delta Ct}$) rather than as a fold change relative to saline ($2^{-\Delta\Delta Ct}$) allows the reader to more fully appreciate overall temporal evolution of a panel of genes that represent microglial function, as well as a panel of genes that represent mitochondrial function. We feel this allows the reader to appreciate whether changes in *microglial* gene expression occur first, and mitochondrial gene expression changes lag behind, or whether changes to expression of mitochondrial genes are observed as early as changes to microglial genes. In our opinion, the heatmap approach is the best way to capture this. Use of PCA analysis based only on mitochondrial genes further highlights that overall cellular state in response to LPS treatment can be fully captured by looking only at the mitochondrial relevant genes of that cell (24hr samples cluster clearly away from others and 4hr samples show trends of moving toward the 24hr samples).

Use of relative expression ($2^{-\Delta Ct}$) to quantify and display qPCR data is explained in the Applied Biosystems User Bulletin #2, which describes in detail multiple approaches for quantification of qPCR data. Relative expression ($2^{-\Delta Ct}$) is widely used in the field (A few examples - PMID: 23603814 Nature Medicine; PMID: 38104136 Molecular Neurodegeneration; PMID: 36581670 Scientific Reports; PMID: 34266459 Molecular Neurodegeneration).

Regarding the magnitude and time course of the inflammatory response to LPS treatment, these responses can vary widely and are impacted by the specific dose used, route of administration (i.p. versus s.c. versus intracranial), LPS type (O111:B4 vs. O127:B8 vs O55:B5 vs O128:B12), age of the mice, background strain of mice, and the vivarium in which the animals live (PMID: 11723144; PMID: 38183948, PMID: 31075861). The increases in pro-inflammatory factor mRNA that we report are within this wide range of magnitudes/time courses that have been previously observed. Moreover, it's worth noting that our z-score heatmaps emphasize changes in expression of individual genes across time and do not emphasize the relative magnitude of gene expression changes across individual genes. The increases we observed in Tnf α and Il1 β at 4hrs were highly significant (see **Supplementary Table 2**) and, if converted into

a fold change relative to saline ($2^{-\Delta\Delta Ct}$), are, by far, the largest fold changes we observed among all interrogated genes (Reviewer Figure 5).

Regarding testing for statistical significance - as stated in our methods in the original submission,

Bonferroni correction for multiple comparisons were included for all post-hoc tests following ANOVA or two-way ANOVAs.

→ To address the reviewer's concern and ensure that our statistical approaches are clear, we have added statements to relevant figure legends that correction for multiple comparisons was employed.

Regarding the choice of endogenous control gene for qPCR, the reviewer brings up the excellent point that one needs to be cautious regarding *Gapdh* in some contexts.

→ To address this concern, we repeated most of the qPCR experiments from MG-TFAM KO and control mice using *Gusb* as a housekeeping gene. We also used *Gusb* for the new qPCR experiments involving LPS challenge of MG-TFAM KO and control mice (Fig. 8 and Fig. S8). PCA plots using the gene expression results from *Gusb* (Fig. 8C) and from *Gapdh* (Fig. S8C) both indicate that TFAM KO is a key driver of observed variance, accounting for microglial distribution on PC1 or PC2, and both PCAs show that midbrain and striatum TFAM KO microglia exhibit distinct responses to this manipulation.

§ PCA and correlation plots are largely uninformative given the small number of genes sampled. We respectfully disagree with the reviewer on this point. PCA helps probe and illustrate the main "overall take home message" of any dataset that has a "large" number of variables. We have interrogated 23 different genes in samples taken from two different brain regions (MB and STR) and three different treatment conditions (saline, LPS 4hr, LPS 24hr). What is the overall main driver of microglial state in this situation? Is it the brain region from which the cells were isolated? Is it the treatment group that the animal was part of? Is it the state of inflammatory factor transcription? Carrying out dimensionality reduction via principle component analysis can help sort this out. We feel that the PCA in Fig. 4C helps convey the message that mitochondrial genes alone are enough to identify which samples are in different phases of responding to LPS challenge.

One does not need to sample 1000s of genes for PCA to be useful. Essentially anytime you have 3 or more continuous variables, dimensionality reduction can be useful. We have 23 variables when including all genes analyzed and 14 variables when only including mitochondria-relevant genes. Other important conditions for using PCA include having a sufficient number of samples

per variable. SPSS recommends at least 5-10 samples per variable. We have 41 samples total, 13-14 samples per treatment group, and 20-21 samples per brain region. Examples of PCA being used for qPCR data (rather than RNAseq or other -omics datasets) are abundant in the literature. A few examples - PMID: 20142997 (lung cancer) PMID: 35692040 (plant biology) PMID: 35401509 (atlantic salmon) PMID: 29926971(achilles tendon) PMID: 36499629 (mesenchymal stem cells) PMID: 36992300 (avian influenza).

§ All of these issues would likely be remedied with a well-designed RNA-seq experiment on purified microglia using the sorting scheme shown in Fig. S1D.

We fully agree that more comprehensive analysis of the microglial transcriptome in the context of targeted manipulation of microglial mitochondria is of great interest. (RNAseq of microglia during responses to LPS have already been published, e.g. PMID 26884166, PMID 30477578). We have carried out RNAseq of microglia in TFAM KO and control animals and analysis of this dataset is underway. However, this dataset is part of a follow-up project that investigates the contribution of microglial mitochondria to risk for Parkinson's Disease. We feel that inclusion of RNAseq data is well beyond the scope of this current (foundational) manuscript which reveals the microglial mitochondrial landscape in multiple contexts in which these cells exhibit phenotypic remodeling.

o While the motivation to ablate TFAM in microglia is not clear from Fig. 4, it is still a very interesting experiment for the field, as a whole. The finding of increased coverage of microglia in Fig. 7I is very interesting and somewhat unexpected but would be bolstered significantly by counting individual microglia using either the existing images or flow cytometry to determine if these is proliferation or increases in size or process extension. We apologize that the rationale for targeting TFAM was not clearly described. The text of the manuscript originally said "TFAM deletion in diverse cell types leads to reduced expression of electron transport chain subunits encoded by mitochondrial DNA, as well as changes in mitochondrial membrane potential and mitochondrial morphology"

→ **To address the reviewer's concern we have expanded our explanation of the rationale for the TFAM manipulation with the following text** "Results presented thus far indicate that there are correlations between region-specific microglial morphology and mitochondrial mass/number. They also reveal correlations between mitochondrial mass, mitochondrial elongation, mitochondrial membrane potential and changes in microglial morphology and inflammatory profiles during responses to LPS challenge and aging. To further probe whether there are causal links between mitochondria and these microglial phenotypic attributes, we crossed *CX3CR1^{CreER/+}* mice and *CX3CR1^{CreER/+};mitoGFP* mice with *Tfam^{fl/fl}* mice to achieve targeted deletion of mitochondrial transcription factor A (TFAM) in microglia (MG-TFAMKO or MG-mitoGFP-TFAMKO)(**Fig. 7A**). TFAM deletion in diverse cell types leads to reduced expression of electron transport chain subunits encoded by mitochondrial DNA, and gives rise to changes in mitochondrial membrane potential and mitochondrial morphology (62,63), making this an ideal genetic manipulation to further elucidate the relationship between mitochondria and microglial phenotype."

→ To expand on our observation of increased tissue coverage in TFAM KO microglia we have analyzed microglial density and process branching of microglia in TFAM KO mice. See responses to Reviewer #1, point 3.

o For Fig. 8 to be considered reliable, *Tfam* must be shown to demonstrate effective knockout and also microglia purity. Unlike in Fig. 4, $\Delta\Delta Ct$ values are shown, rather than Z-scores, but again fold-change would be much more interpretable. Here again the use of *Gapdh* as a normalization control is questionable given the direct targeting of metabolism with *Tfam* knockout. Again, this experiment would be significantly improved with a well-designed RNA-seq experiment on purified microglia using the sorting scheme shown in Fig. S1D.

We thank the reviewer for pointing out that we neglected to include qPCR data indicating purity of the collected cells. We carry out analysis of cell purity at the start of any qPCR experiment, and had intended to include these data in the manuscript. **These data have now been added to Figure S4A** and indicate that microglia purified with the indicated FACS gating workflow show high expression of microglial markers *Cx3cr1* and *P2ry12*, and negligible expression of markers for other cell types (*Olig1*, *Olig2*, *Aldh1l1*, *Slc1a2* [GLT1], *Ppp1r1b* [DARPP-32], *Slc32a1* [VIAAT]). Demonstration of TFAM knockout was shown by qPCR in **Figure 7B** in the initial submission. It is now also shown in **Figure S8A**.

The qPCR data shown in **Fig. 8** in the original submission were displaying fold change of TFAM KO microglia relative to control microglia from the same brain region, calculated using $2^{-\Delta\Delta Ct}$. We thank the reviewer for pointing out that y-axis labels of “ $\Delta\Delta Ct$ ” could cause confusion in this regard. **In the updated Fig. 8, y-axes are labeled “fold change”** in any instance where $2^{-\Delta\Delta Ct}$ has been used to compute displayed values.

→ To address the reviewer’s concern regarding choice of endogenous housekeeping gene, we repeated most of the qPCR experiments from MG-TFAM KO and control mice using *Gusb* as a housekeeping gene. We also used *Gusb* for the new qPCR experiments involving LPS challenge of MG-TFAM KO and control mice (**Fig. 8 and Fig. S8**).

- Minor comments

o If the authors have set alpha to 0.05 for these experiments, $P = 0.05$ should not be marked as significant (see Fig. 1L) We thank the reviewer for highlighting this important detail. We have re-checked all figures to make sure that $P = 0.05$ was not marked as significant.

o Figure callouts frequently occur out of order. Panels should appear in the order they are mentioned in the text. Similarly, some panels appear to never be referenced and should be either mentioned or removed, e.g. S2H, S2J-K. We thank the reviewer for pointing this out and have adjusted figure organization and results text to address this issue.

o The use of multiple Y-axis scales (as in S2F-G) is not ideal data visualization and should be avoided. Faceting or simply presenting a scatter plot is far more interpretable. We now plot data on separate histograms to address this concern (**Figure S2G-I**).

o All heatmaps should show all samples, rather than just the group average.

While we agree with the reviewer in principle, given the that we aim to reveal and emphasize underlying gene expression patterns across two brain regions, three treatment groups (or two genotypes), and two major categories of genes (mitochondria-function relevant genes and microglial function relevant genes), we feel that inclusion of heat maps with all samples is not the most effective method for visually highlighting overall gene expression patterns. **However, to address the reviewer's concern, we have added heatmaps that include all samples to Figure S4 and Figure S8.**

o Flow cytometry data should be displayed as contour plots with outliers shown. At present, with scatter plots, it appears that many populations are split arbitrarily. While this may not be the case, this is important for showing the true distribution.

We have updated all FACS data to be displayed as contour plots. In most cases, we are not attempting to split populations for quantification. Rather, quadrant boundaries have been placed to allow shifts in the overall population to be detected. For example, in **Fig. 4E**, all microglia are TMRM+ and the boundary for GFP intensity was placed just above the population center in saline samples. This highlights increases in mitoGFP intensity that emerge at 4hrs and 24hrs following LPS treatment.

o Data presentation in Figs. 6F and 7J is nonstandard and makes use of pseudoreplication by treating each cell or mitochondrion as an independent measurement. Instead, comparisons of median fluorescence intensity for each mouse / region should be displayed (ideally as boxplots) and compared statistically. These findings are potentially very interesting and proper analysis would better highlight them.

We respectfully disagree that cumulative distributions are a non-standard data presentation. These distributions are generated by first plotting a cumulative distribution for each mouse (in essence, the *shape* of the distribution for one mouse represents one datapoint).

Distribution curves from all mice in a group are then averaged to generate the average distribution curve for a given age/ brain region / genotype, which is what is being compared statistically. We feel this largely mitigates the concern about pseudo replication.

→ **To address the reviewer's concern, we have included a new section in the methods "Flow cytometry analyses" that includes the following description of how cumulative distributions for Figure 6 were generated:**

"In some experiments, fluorescence signals across the entire population of cells from one sample were examined via cumulative distribution plots. For these analyses, a cumulative distribution plot was generated for each sample. Cumulative distribution curves from all samples in a group were then averaged to generate an average plot for that group."

→ **We also added the following description to the methods section describing "Confocal image acquisition and analysis":** "For analysis of mitoGFP signal intensity in *MG-mitoGFP-TFAMKO* and *MG-mitoGFP* control mice, mean fluorescence intensities for each individual, Imaris-reconstructed mitochondria in a field of view were extracted. Mean fluorescence intensity values for all reconstructed mitochondria from one mouse were then pooled (n=3 FOVs per mouse per brain region) and a cumulative distribution plot for that mouse and brain

region was generated. Cumulative distribution plots from all mice from one genotype and brain region were then averaged to generate an average cumulative distribution for that group.”

Reviewer #3 (Remarks to the Author):

Reviewer #4 (Remarks to the Author):

The manuscript presents valuable findings about the location of mitochondria in microglia and will, therefore, have some implications for the microglia community. The results align with previous in vivo microglia data analysis (Maes et al. 2023, cited by the authors) from a different mitochondria labeling mouse model using fixed and live in vivo imaging in the nucleus accumbens (NAc) and the ventral tegmental area (VTA). The data presented is solid and broadly supports the claims. The most compelling data is that dynamic microglia processes often lack mitochondria, which is counterintuitive to likely energy-demanding microglial processes and the concept of mitochondria as an energy source. The manuscript contains a lot of experiments yet does not go beyond this observation. The manuscript remains mainly descriptive of the microglia and mitochondria phenotypes. The rationale for exploring different environmental alterations and their impact on the microglia-mitochondria network is not straightforwardly defined, leaving the impression of a phenotype search.

We thank the reviewer for recognizing that our data make valuable contributions to the field and that the presented data are solid. We are sorry to hear that the rationale for analyzing mitochondrial properties and how they relate to key microglial attributes was not easily appreciated and understood. Our launching point for this study was precisely the emerging recognition that mitochondria do more than simply produce energy. In fact, for some cell types such as macrophages and other immune cells, this organelle plays essential roles in regulating dynamic changes in the overall functional state of the cell. Given that very little is known about microglial mitochondria, to explore whether there are important relationships between this organelle and overall microglial functional state, we aimed to analyze key properties of microglia and their mitochondria in instances where the functional state of these cells changes. We specifically chose examples of microglial phenotypic remodeling that operate at distinct temporal scales (laser lesion - minutes; LPS - hours to days; aging - months to years) and degrees of physiological vs pathological (baseline regional differences in microglial phenotype - physiological; baseline microglial tissue surveillance - physiological; normative aging - arguably physiological; laser lesion - pathological; LPS - pathological). We also chose pathological challenges that are distinct in modality (tissue injury vs. immunological/ inflammatory insult vs. lifespan tissue maintenance) Also by design, we wished to interrogate the relationship between microglial mitochondria and microglial attributes in paradigms where abundant data regarding microglial phenotype are available (from our own prior studies and the broader field). It is

disappointing to hear that our study design leaves the impression of a phenotype search, but this means that we need to do a better job of articulating rationale behind study design in the abstract/introduction/results.

→ **To address this concern, we have made changes to wording of the abstract, introduction, and beginning of each section in the results. We hope that the reviewers find that this highlights and justifies study design rationale more effectively.**

We also acknowledge that much of our study is descriptive, but we would argue that it is precisely these types of studies that are needed to enable follow-up mechanistic studies in the field. Mitochondria can impact cellular function and phenotype through energy/ATP production, but they also impact cellular function and phenotype via calcium buffering (which impacts intracellular signaling pathways), mitochondrial-nucleus signaling (which regulates gene expression), production of reactive oxygen species (which have important intracellular signaling roles), production of metabolites used for protein posttranslational modifications, and lipid metabolism (which can shape cell membrane properties and dynamics). These are observations about mitochondrial function built up from many fields and disparate cell types. Without foundational guidepost observations regarding *which* microglial attributes are most tightly linked to *which* aspects of mitochondrial status, we can't, as a field, make focused hypotheses or design optimal mechanistic interventions that test causation. Given the state of knowledge in the field regarding microglial mitochondria, it would be a very high bar to request both the foundational mapping and the mechanistic manipulations in the same study.

Introduction:

The introduction needs to provide more confidence and perspective on the goal of the manuscript. Although the first paragraph introduces microglia and their ability to shift between functional states, the concept of a microglia state must be defined. The second paragraph introduces the expression of different cell surface receptors, finishing with a question of how microglia interpret different signals. From the third paragraph, it is not intuitive how mitochondria as “cell attributes” are involved in the previously introduced perspective of functional states and interpreting different signals. What impact would the mitochondrial network remodeling have on the microglia? Furthermore, if the authors consider “cell attributes” as intrinsic properties of microglia to define their functional state, then this should be clearly outlined and justified. Lysosomes, for example, would be a more intuitive “attribute” to distinguish the different microglial states in the conditions tested within this manuscript compared to mitochondria.

In addition, the introduction across all three paragraphs consists mainly of attribute lists not followed up in the manuscript, such as high-fat diet, pollution exposure, and TLR4 and TGFBR2. Also, the list of mitochondrial parameters needs to be more focused since the study does not evaluate buffering calcium, metabolites, or interactions with other organelles.

Overall, the introduction only loosely connects mitochondria distribution and potential microglia state and does not provide a clear rationale for justifying the study.

We are sorry to hear that the reviewer did not feel that the introduction clearly and cohesively set up the rationale for the study.

→ We have edited and expanded the introduction to try and address many of the points raised by the reviewer. Some of the information (e.g. description of multiple functional roles for mitochondria as organelles) is necessary background information for readers who may not be familiar with the fact that mitochondria do more than simply produce ATP.

Results:

Fig 1/S1-2:

i) The authors performed a detailed verification of their mouse model to ensure that mitochondria were labeled. This data set looks convincing. However, without ultrastructural analysis, it cannot be ensured that MG-mitoGFP mice “accurately reveal the in vivo distribution of microglia mitochondria.” If the authors seek this accuracy as they write, they must perform ultrastructural analysis.

Use of this tool to accurately label mitochondria has been published previously (PMID 28132831). We would also like to refer the reviewer to studies by Pietramale et. al (BioRxiv 2024.10.15.618505) and Maes et. al (PMID 37731609) that also use a cre-dependent mitochondrial label (PhAM; JAX #018385) in microglia and find very similar distribution and morphological features of microglial mitochondria as what we report using mitoGFP (JAX #021429). Pietramale et. al also carry out analysis of human cortical EM datasets and found very similar somatic vs cell process distribution of microglial mitochondria as what is revealed by both cre-dependent fluorescent reporters.

Together with our acute brain slice experiments using TMRM to evaluate colocalization with microglial mitoGFP signal (Fig. S1J,K), we feel that we are justified in stating that this label accurately reveals the distribution of microglial mitochondria. In addition, we have carried out experiments in which bone marrow derived macrophages (BDBMs) were generated from $CX3CR1^{CreER/+};mitoGFP$ mice. Visualization of all mitochondria in these *in vitro* preps is very clear and indicates that all TMRE labeled mitochondria in the BMDMs are also labeled by GFP. Moreover, all GFP+ structures are TMRE+ (i.e. GFP is not labeling other organelles) and the full extent of the TMRE labeled mitochondria is also captured by GFP (Reviewer Figure 6). If the

reviewer feels strongly that these data are necessary to demonstrate fidelity of our mitochondrial labeling, we can incorporate these data into supplementary figures.

ii) Since the main text states that mitoGFP recombination was substantial without tamoxifen injection, is the data in Fig. S1 without tamoxifen injection, while Fig. 1 with tamoxifen? For clarification, the authors should annotate in the figure legend if they have administered tamoxifen. If Fig. S1 is without tamoxifen, the authors should validate the model again with tamoxifen since tamoxifen could potentially affect the mitoGFP distribution. Furthermore, for Fig. 1F, the analysis has been done without tamoxifen for rigorous analysis. This is surprising since microglia coverage across tissue is not overlapping, and confocal images should be sufficient to resolve the mitochondria signature for each microglia.

All experiments were carried out without tamoxifen injection except experiments to induce TFAM knockout from microglia.

→ **We have added clarification to this effect in the results and methods sections. We have also added clarification to all figure legends about whether tamoxifen was used for that specific set of analyses.** The decision to forgo tamoxifen injection in MG-MitoGFP mice was motivated by primarily by the observation that 80-90% of microglia showed recombination and mitoGFP expression in the absence of tamoxifen injection. With such a high percentage of cells expressing the mitoGFP tag, we did not feel that there was substantial risk that we were only sampling a subset of cells.

→ **We have also carried out additional experiments to treat mice with tamoxifen and compare the properties of the mitoGFP+ population of microglia with and without tamoxifen treatment (Figure S1D-H and Reviewer Figure 2).** Tamoxifen injection resulted in a slightly higher percentage of microglia expressing mitoGFP but did not result in any significant changes in GFP intensity, FSC, SSC, CX3CR1, CD45, or P2RY12, confirming that forgoing tamoxifen treatment does not result in biased sampling of only a subset of microglia.

With regard to experiments for Figure 1F using the triple transgenic *MG-MitoGFP;Ai14* animals, the presence of two cre-dependent transgenes results in sparser recombination and lower densities of mitoGFP+ only, TdTom+ only, and mitoGFP+TdTom+ cells. Although we agree with the reviewer that reasonable reconstruction of microglia can be carried out when the microglia are labeled with Iba1 staining or a universally expressed fluorophore (e.g. *CX3CR1-EGFP* mice), it can be tricky to unequivocally distinguish which terminal processes belong to which cell, particularly in brain regions where microglia present at higher density and show greater morphological complexity, such as the nucleus accumbens (NAc). Hence, we maintain that 3D reconstruction of individual cells is more rigorous when the cells are sparsely labeled.

iii) How relevant is the correlation between mitochondrial number and microglia complexity/number of intersections? Are these just larger cells with more mitochondria? Based on the data in Fig 1E, the data point variation for each region is small. Furthermore, how have the microglia for reconstruction been selected?

What is the rationale for choosing the “maximum number of Sholl intersections”? Which radius size has been used? This can severely impact the readout. Furthermore, was this maximum always at a similar distance from the soma, or has there been some variability? This could

explain potential variability in the data. The same question applies to Fig. S3E. The value used for sholl analysis in Fig. S3D is also unclear.

→ **To clarify how cells were selected for 3D reconstruction the following text was added to the methods:**

“To select microglia for 3D reconstruction, the brain region of interest was located and centered using only TH staining and anatomical landmarks with 20x objectives. After switching to 60x objectives, TdTom+mitoGFP+ microglia within that field of view were imaged. Slight X-Y adjustments were made, if needed, to bring a TdTom+mitoGFP+ microglial cell away from the edge of the FOV to avoid truncating processes. Cells were excluded from reconstruction if their somas fell near the top or bottom edge of the brain slice rather than being situated near the middle of the z-stack.”

Regarding maximum number of Sholl intersections - in the original submission we chose to highlight the relationship between this morphological metric and mitochondria because this nicely captures the largest degree of branching complexity that a cell displays, irrespective of whether that branching complexity is close to the soma or further away at distal processes. As the reviewer points out, each individual cell will likely achieve its maximum number of intersections at a different radius from the soma. On its own, this metric doesn't offer any special advantage over other morphological metrics, which is why we also analyze total process length and sum of branch points (obtained directly from Imaris reconstructions).

→ **In the revised manuscript, we highlight total process length and sum of branch points in the main figure (Fig. 1G-J) and also present data from max # of Sholl intersections in the table in Fig. S2J.** Data from PFC microglia are now also presented in addition to NAc and VTA microglia.

Regarding data presented in **Fig. S3D**, total number of intersections is the sum of all intersections at all radius lengths. This metric is meant to capture the total branching complexity of the cell. → **To clarify this point, the text of the figure legend has been changed to “Robust regression plot showing relationship between total mitochondrial volume per cell and microglial morphological complexity (total Sholl intersections across all radii).”**

Regarding the biological significance or relevance of the correlations between microglial mitochondria and morphological metrics - Nothing is known about whether energy production or any other facet of mitochondrial function plays a critical role in determining the type of morphology that a microglial cell can adopt. If the mitochondria do play some role in regulating cell morphology, one might expect to find a correlation between mitochondrial number/volume and morphological complexity. If, instead, mitochondria were *only* important for regulating microglial production and secretion of trophic/inflammatory factors, there might be no correlation between the morphological complexity of an individual cell and the abundance of mitochondria in that cell. In **Reviewer Figure 6**, above, one can appreciate that a fairly amorphous Bone Marrow Derived Macrophage (BMDM) has tons of mitochondria. But it would be difficult to relate mitochondrial abundance to morphological *complexity* of BMDMs in any fashion. In this regard, the relationship between mitochondria and microglial morphological complexity may be more akin to what is observed in neurons, where mitochondria are linked to dendritic spine morphology and remodeling (PMID 32714603).

→ Elaboration of some of these points has been added to the discussion.

iv) The choice for looking at the NAc and VTA needs better clarification, also in perspective for the later analysis with LPS and aging. Are those brain regions particularly vulnerable?

→ We have added text throughout the results section to explain in greater depth our rationale for focusing on these two brain regions. These additions include:

“In fixed tissue from young adult (2mo) double transgenic *CX3CR1^{CreER/+};mitoGFP* mice (*MG-mitoGFP* mice hereafter), numerous GFP+ structures could be observed throughout the somas and cell processes of microglia in the nucleus accumbens (NAc) and ventral tegmental area (VTA), two interconnected brain regions where we have extensively mapped microglial phenotypes across the lifespan (Fig. 1B and Fig. S1A,B)”

“Microglia show prominent differences in cell density, branching complexity, and gene expression across the NAc and VTA³⁰. To determine if these regional differences in baseline microglial attributes were accompanied by regional differences in microglial mitochondria, we carried out high resolution confocal imaging and 3D reconstruction of both microglia and their mitochondria in the NAc and VTA of young adult mice (Fig. 1B and Fig. S2A).”

“We also showed previously that microglial responses to aging vary across brain region, with VTA microglia displaying proliferative and inflammatory changes months before microglia in the NAc. Whether there are links between mitochondrial state and early VTA microglial aging has also not been investigated.”

v) Based on the figure legend description in Fig 1. It is understood that microglial volume is calculated per field of view (tissue coverage)- and mitochondrial number, volume, and mass are then calculated based on this FOV microglial volume. However, the text states, “greater number of mitochondria relative to cell volume, as well as a greater total mitochondrial volume relative to cell volume (mitochondrial mass)”. If the quantification is relative to total microglial volume per FOV, and not on a per-cell basis, stating ‘cell volume’ is confusing. For example, Fig. 1D-E, is the relative ‘microglia volume’ in 1D the same denominator as ‘cell volume’ in 1E? In addition, does the number of mitochondria within a FOV bring us important information? Since many microglial processes not connected to a cell body are present within a brain volume, these averages do not indicate information on a per-cell basis. The authors should identify precise nomenclature and analysis methods, describe this in detail in the methods, and use the terminology consistently throughout the manuscript.

We performed analyses using a FOV approach to obtain information about microglia as a cell population in that particular brain region. We feel that carrying out FOV analyses AND individual cell focused analyses is more rigorous than either approach alone. FOV analyses can smooth out the effects of cell-to-cell variation and remove concerns about any unintended bias in selecting individual cells for analysis.

→ We have updated descriptions in the methods, figure legends and text to ensure consistency and avoid confusion about the distinction between FOV-based versus individual cell-based analyses.

vi) The mitochondrial field focuses more on measuring network connectivity, i.e., fragmented or connected networks. This metric is not shown throughout the manuscript but would be an essential feature to describe when considering mitochondrial structural network changes. The reviewer is correct that measures of “fragmentation” vs interconnected “networks” are commonly seen in the mitochondrial field. The vast majority of these studies are carried out in cultured cells where genetically-encoded mitochondrial reporters and/or mitochondrial dyes can easily be used with fantastic signal-to-noise, allowing automated reconstruction and analysis of mitochondria using machine learning platforms like Avia. In fixed tissue, however, even with optimal confocal imaging, distinguishing whether a cluster of mitochondria are fully, physically interconnected or separate is much more challenging. This is particularly true in the cell soma, where mitochondria are often very closely packed. For this reason, we feel it is more accurate to analyze mitochondrial elongation, mitochondrial sphericity, mitochondrial number, and mitochondrial mass which are also widely used in the mitochondrial field and to avoid classifications like “networked” or “fragmented,” which carry functional implications.

→ To address the reviewer’s point, we have included new material in the discussion, commenting on the limitations of light microscopy for resolving whether mitochondria are truly networked or discrete, and additional future approaches that can be used to determine whether small mitochondria are have high ROS / low TMRM and are likely to be undergoing mitophagy (which is often assumed when employing the term fragmented) versus they have undergone fission to facilitate trafficking.

Fig 2:

i) The conclusion of Fig. 2C needs to be clarified. If VTA and NAc show the same motility, how can the conclusion be made that the VTA is surveying less based on their morphology?

We originally advanced this interpretation because, *in vivo*, VTA microglia are less dense and less branched than NAc microglia (**Fig 1C,G,I** and PMID: 28689984 and PMID: 35396327).

Hence, if microglia in both brain regions show similar motility, it will take VTA microglia longer to execute a complete surveillance of the surrounding tissue than the more abundant, more highly branched NAc microglia.

→ However, given that our observations of motility are made in an acute brain section preparation, we removed this statement to address any concerns that it is too speculative.

ii) Do the motile processes contain mitochondria? Or are motile mitochondria only found in primary processes that are more stable? (also related to Fig. 3). The reviewer brings up the very interesting question of how subcellular location relates to the motility of individual microglial cell processes and the motility of the mitochondria themselves.

→ To address this point, we have carried out additional analyses focusing on individual microglial cell processes (Fig. 2J). Similar to our findings at the level of the whole microglial cell, this analysis revealed that there was not a significant correlation between the abundance of mitochondria in a primary cell branch and the motility of that branch. There was also no significant correlation between the motility of mitochondria in a branch and the motility of the branch itself (or vice versa). Hence, it is not the case that microglial process branches that have more mitochondria exhibit any heightened capacity for cell process remodeling, at least over the time frame we are analyzing. We draw the reviewer’s attention to the study by Pietramale

et. al (BioRxiv 2024.10.15.618505) co-submitted with our study that suggest that this result may differ if focused on even smaller branchlets. By focusing on smaller branchlets (rather than entire arbors branching out from one primary process), they found that microglial branchlets with fewer mitochondria were more motile. There could also be important brain region differences in this mitochondria-branch motility relationship. **This will be an important area for the field to investigate further, and we elaborate on some of these considerations further in the discussion.**

Fig 3:

i) This data is very interesting and raises exciting new questions; however, no quantification supports the concluding statement.

We are glad that the reviewer finds these data intriguing.

→ **In this resubmission, we have included quantification of microglial process motility and mitochondrial motility following laser lesion (Fig. 3F-I).** These analyses revealed that abundance of microglial processes close to the lesion increases over time, whereas abundance of microglial mitochondria close to the lesion does not increase, confirming our qualitative observations from the initial manuscript submission.

ii) At this resolution, it seems difficult to quantify and confirm that the mitochondrial networks remain stable after focal injury. Can the authors image at the resolution seen in Fig 2 and provide data analysis of mitochondrial network motility before and after injury? Do all mitochondria stop moving, or only a few travel within the extending process?

In this set of experiments, we elected to image a larger field of view in order to sample a larger number of microglia to determine how they and their mitochondrial networks respond to laser lesion. While repeat experiments to image smaller fields of view at higher resolution could be carried out, we felt this would best be accomplished via *in vivo* multiphoton imaging that would also permit mitochondria to be monitored over a much longer time frame. Using such *in vivo* multiphoton approaches, Pietramale et. al (BioRxiv 2024.10.15.618505) found similar responses of microglia and their mitochondria in cortex, with negligible mitochondrial remodeling within 40min of the laser lesion. Intriguingly, they *did* observe mitochondrial remodeling at 6hrs following lesion.

Fig 4/5:

i) If mitochondria are not linked to microglial tissue surveillance and injury, as the authors conclude in Fig. 2J and Fig. 3, respectively, how do they rationalize using LPS? Based on live imaging, mitochondria are not linked in a “direct fashion to baseline surveillance” or injury-induced damage. Why would it suddenly be related to an inflammatory response induced by LPS? We are sorry to hear that the rationale for utilizing LPS to further interrogate the relationship between mitochondria and microglial responses to tissue challenge was not clearly communicated. Laser lesion is a focal tissue injury that elicits a rapid (within minutes) response from microglia. Purinergic signaling from the injured tissue to microglial P2RY12 receptors plays a critical role in this rapid injury response. Our data indicate that rapid extension of microglial processes is not accompanied by rapid remodeling of mitochondrial networks (at least on a morphological level). Given this result, to continue testing whether mitochondria play an

important role in regulating microglial phenotypic remodeling in response to pathological challenges, it was desirable to examine microglial responses that occur over a longer time scale (hours to days) as well as microglial responses to a distinct modality of challenge (inflammatory challenge rather than tissue damage).

→ As described above, we have made changes to wording of the abstract, introduction, and beginning of each section in the results to better explain rationale for study design.

ii) QPCR of *Dnml1* is used to indicate fission- however, this data should also be reflected in the image analysis in Fig 5. Could the authors indicate changes in the median volume of organelles (per cell) to compare alongside the mitochondrial number and aspect ratio? If the mitochondria are indeed elongated as indicated by an aspect ratio measure, the median volume of organelles per cell should increase (aligning with a mass increase) since the number of mitochondria did not change between saline and LPS for both brain regions. The reviewer is correct that dynamin-related protein 1 (DRP1; encoded by the gene *Dnml1*) plays a key role in promoting mitochondrial fission. In both our FACS-based analyses in Fig. 4F and Fig. 8D, the prominent increase in GFP median fluorescence intensity suggests that mitochondrial biogenesis accompanies microglial responses to LPS challenge.

The increase in mitochondrial mass detected via histology (Fig. 5B), further indicates an overall increase in mitochondria within microglia following LPS challenge. The reviewer is correct that, in general, mean volume of individual mitochondria trended toward increases at 24hrs following LPS (Reviewer Figure 7), particularly in the NAc, so the FACS-based increase in GFP and the increase in mitochondrial mass via histology is likely driven by larger, longer mitochondria.

These findings don't necessarily represent an inconsistency with the increase in *Dnml1* transcript found by qPCR. Indeed, *Dnml1* may be getting upregulated precisely because there are larger, longer mitochondria and increased fission will be needed to return the mitochondrial network to its pre-LPS state.

iii) Do the authors have the details of the post-hoc analysis of 5B-D?

As described in the methods in the original submission, all one-way and two-way ANOVAs were performed with Bonferroni correction for multiple comparisons being applied to all post-hoc comparisons.

→ We have added information about post-hoc corrections to the figure legends.

iv) Technical aspect related to sampling for flow cytometry. The method section must clarify whether the samples were processed sequentially or in batches. Furthermore, information related to the total time from tissue removal to sorting must be included. It has been shown that the microglia transcriptional profile adapts within 30 minutes of incubation on ice (Marsh

2022. DOI: 10.1038/s41593-022-01022-8). This can result in misleading results. Therefore, the authors should add the animal grouping strategy and total processing time to the methods and, if necessary, repeat the experiments when the timing is longer than 30 min.

The authors are correct that microglia are extremely sensitive and adaptable cells that will respond to tissue dissociation. Hence, care needs to be taken in the tissue processing workflow to avoid artifacts. We are very familiar with the Marsh et. al paper to which the reviewer refers. In this study, the key finding is that exposing microglia to elevated temperature (37°C) to carry out enzymatic digestion causes them to exhibit altered transcriptional profiles. See Figure 1C of the paper. This panel shows that you can prevent aberrant microglial transcriptional signatures by keeping the entire workflow at 4°C (i.e. on ice) OR by including transcriptional inhibitors. For all experiments in our study, the entire workflow was carried out at 4 °C as described in the methods. In our prior studies (PMID: 28689984), RNAseq confirms that this workflow does not elicit upregulation of inflammatory factors typically associated with microglial reactivity. Moreover, this workflow does not induce increased expression of the specific genes highlighted in the Marsh et. al study (**Reviewer Figure 8**).

Of note, none of the workflows in Marsh et al are completed in under 30min. All of their tissue processing workflows take 70-130min as indicated by their workflow diagrams in Figures 1 and 2. 30 minutes refers to the portion of their workflow during which microglia were exposed to elevated temperature for enzymatic dissociation OR maintained at 4 degrees, with the latter

treatment preserving *in vivo* gene expression signatures. Moreover, their stated workflow time does not include time for sorting, but rather indicates time prior to when sorting begins.

→ **As requested by the reviewer, we have added the following information to the methods regarding our sample processing:**

“Total sample processing time from brain dissection to completion of FACS purification ranged from 2-4hrs depending on sort order of individual samples. Samples were collected in batches with each day of sorting including a young adult, saline injected, or TFAM-wt control mouse. Sort order of samples from specific ages, brain regions, treatment conditions, or genotypes was interleaved to avoid introduction of bias related to tissue processing length. FACS-isolation itself was carried out at 4°C. FACS-isolation of all samples for a given set of experiments was completed prior to downstream processing to allow simultaneous RNA isolation and cDNA preparation as described below.”

v) How was the correlation analysis performed in Fig. 4D? Cherry-picked genes were connected to putative microglial states, an aspect that is also not well-defined, introducing a bias. Also, the explanation of the “few CD68 correlations with other genes” does not seem solid (Fig. S4B) since the heatmap shows much more correlation than any of the other plots in Fig. 4.

To test for potential correlations between microglial genes reflective of overall microglial functional state and our entire panel of 14 mitochondria-relevant genes, we carried out linear regression analyses for each microglial gene and all of the mitochondrial genes. For considerations of space, we included results for 6 / 9 microglial genes in the main figure and an additional microglial gene in Figure S4. We respectfully disagree that showing data from 7/9 microglial genes constitutes “cherry-picking” genes.

It also seems that we have not done an optimal job of explaining the significance of the *Cd68* result. Expression levels of *Cd68* are indeed correlated with expression levels of multiple other microglial genes (*Tgfb1*, *Cx3cr1*, *P2ry12*, **Fig. S4D**). However, the important point here is that this degree of correlation was not modulated by LPS challenge (i.e. these significant correlations were present at baseline and remained present at both 4hrs and 24hrs post-LPS). Also importantly, expression levels of *Tnfa* and *Il1b* were correlated with expression levels of multiple mitochondrial genes, particularly at 4hrs post LPS. Yet, no correlation was observed between *Cd68* expression levels and those of *Tnfa* or *Il1b*. Together, these findings suggest a more robust relationship between the state of mitochondria (as organelles) and acute microglial responses to LPS compared to the relationship between lysosomes and acute microglial responses to LPS. (**Please note that in response to Reviewer 2’s comment, we re-executed our linear regression analyses using Robust regression with false discovery rate control; using this more rigorous approach, some of the correlations that were found to be significant in the original submission are no longer significant in this revised analysis*).

→ **To address the reviewer’s comment, we have modified the text of the results:**

“Moreover, examination of *Cd68*, a key microglial lysosome gene, hints that mitochondria, as organelles, may be uniquely positioned to regulate microglial functional state during inflammatory challenge. While LPS treatment increased the correlation between most mitochondria-relevant genes and key microglial function genes (*Cd45*, *Cx3cr1*, *P2ry12*,

Tgfb1)(**Fig. 4D**), LPS treatment did not modulate correlations between *Cd68* and *Cx3cr1*, *P2ry12*, *Tgfb1* (correlations between *Cd68* and *Cd45* increased at 24hrs post LPS)(**Fig. S4D** and **Table S4**). Moreover, while multiple mitochondria-relevant genes were positively correlated with expression of the microglial pro-inflammatory factors *Tnfa* and *Il1b* at 4hrs post-LPS, *Cd68* expression was not correlated with *Tnfa* and *Il1b* in any of the conditions examined (**Fig. S4D**)."

Fig 6:

Microglia are known to show a sex-specific response also in adulthood and, especially, during aging. Therefore, the authors must precisely annotate the data plots, indicating which data points come from male or female animals.

We agree that probing for sex-differences is important for the microglial field and we included both male and female mice in all our analyses.

→ To address the reviewer's comment, we have added a supplementary figure that replicates key findings from each of our analyses with the annotation of datapoints that come from male and female mice (**Fig. S9**) We have also added commentary to the discussion that future research should continue to investigate possible sex differences in microglial mitochondria.

Fig 7:

i) Related to the mitochondrial quantification in 7D: Is this data represented as an average volume of all organelles within the cell? If yes, it would be better to use a median, as small mitochondria can skew the average value and misrepresent the data. **Figure 7D** shows the sum total volume of all the mitochondria in a field of view (FOV). This value was obtained for 3 FOVs for each mouse to compute a mouse average, which is displayed in the graph.

→ We have adjusted the label of the y axis and the description of this panel in the figure legend to clearly specify that these are FOV measurements. We have also added FOV median mitochondrial volume and FOV maximum volume of an individual mitochondrion to **Figure S7F**. This highlights that increases in the total number of mitochondria within the FOV are driving increases in total FOV mitochondrial volume, rather than increases in the size of individual mitochondria.

ii) Mitochondrial analysis: In Fig. 7, the analysis focused on volume, number, and sphericity, whereas earlier figures describing the mitochondria in fixed tissue focused on mass, number, aspect ratio, volume, sphericity. To allow comparison between the datasets, the authors should be consistent and include all metrics for all figures. Furthermore, the methods section needs to include a detailed description of the analysis method for each parameter.

→ We have added analysis of mitochondrial mass to this figure (**Fig. 7K**). We have expanded the description in the methods of how these metrics were generated within Imaris analysis software.

Fig 8:

How do the microglia change after TFAM KO? A two-month waiting period may allow time for compensatory mechanisms to circumvent initial damage. **Understanding how TFAM KO impacts microglia is indeed the primary goal of these experiments.**

→ In addition to the histology in the original submission (previous Fig 7I, now Fig 7H) which showed that microglial tissue coverage within the field of view (FOV) increases after microglial TFAM KO, we now show that morphological complexity of individual microglia (analyzed via Sholl analysis) increases in MG-TFAM KO mice (Fig. 7I).
→ We also show that microglial density is unchanged (Fig. 7G).
→ In addition to the qPCR in the original submission that showed multiple changes in microglial gene expression in MG-TFAM KO mice (previous Fig. 8C, now Fig. 8B), we now also show that protein level expression of CD45 and CX3CR1 (MFI during FACS) are unchanged in MG-TFAM KO microglia. Cell size (FSC) and internal complexity (SSC) are increased in microglia from MG-TFAM KO mice (Fig. 8A).
→ Finally, in this resubmission, we challenge MG-TFAM KO mice with LPS and utilize FACS and qPCR to assess the capacity of the cells to mount a mitochondrial and cellular response to this challenge (see detailed description in response to reviewer #1, point 4).

As described in the results, we elected to wait for 2 months following tamoxifen injection for the following reasons. 1) This will allow for any peripheral monocytes/macrophages impacted by this manipulation to cycle out and be replaced. 2) Mitochondrial proteins are among the most long-lived proteins in the CNS (PMID: 30315172). Hence, it is estimated that approximately 1 month would be needed for existing mitochondrial proteins to cycle out. Moreover, TFAM exerts its broad effects on the mitochondria by altering stability of the mitochondrial DNA. This impacts capacity of the mitochondria to produce key components of the electron transport chain, with additional downstream consequences for other aspects of mitochondrial function. In line with this, virtually all studies that have used cell-specific TFAM KO have either used constitutive Cre lines (PMID 31269437, PMID 17227870, PMID 9916807, PMID 25005176, PMID 23168219, PMID 18945718), or waited at least a month (PMID 35165165, PMID 40210116). The reviewer is correct that some compensatory responses may be elicited and some of the changes we observe in expression of nuclear-encoded mitochondrial components may represent attempts at compensation. Nonetheless, we feel that our choice of 2 months is well justified, and it will fall to future studies to tease apart direct vs compensatory effects on the organelle.

Discussion:

In line with the loose introduction, the discussion is also only superficial. It mostly repeats the results without putting the research data into the context of the literature. For example, a reported result is increased mitochondrial mass. What does increased mitochondrial mass imply? The manuscript would enormously benefit from a complete revision of the introduction and discussion to make it coherent.

We are sorry to hear that the reviewer did not find the discussion informative.

→ **We have updated the discussion to include some potential implications of key observations based on what is known in the literature from mitochondrial behavior in other cell types.** Multiple follow-up studies will be required to fully elucidate what a particular mitochondrial attribute means for microglial function in a given context. Our aim with this study was to inspire those types of follow-up studies by revealing how closely linked the general state of the organelle is with the overall state of the cell.

Overstatements:

The manuscript suffers from several overstatements and preliminary conclusions, which must be fixed. A few examples are listed below:

- i) "First time" in the introduction. We have removed this type of wording from the manuscript.
- ii) The authors regularly write "in vivo" within their manuscript, yet it becomes quickly apparent that the majority of the experiments are performed in a fixed tissue environment. This should be clarified in the text to avoid misunderstandings. We have adjusted wording in the manuscript to avoid confusion as the reviewer suggests.

Reviewer #1 (Remarks to the Author):

No further questions.

We thank the reviewer for their time and feedback.

Reviewer #2 (Remarks to the Author):

It is much improved.

They should cite and discuss their findings with other previous studies on mitochondria and microglia.

PMID: 36787364

PMID: 39048800

PMID: 38480879

PMID: 39048801

We thank the reviewer for their time and feedback, and for pointing out additional citations that needed to be included. We have revised our discussion and introduction to incorporate the suggested studies listed above.

Reviewer #3 (Remarks to the Author):

Reviewer #4 (Remarks to the Author):

The manuscript presents valuable findings about the location of mitochondria in microglia and will, therefore, have some implications for the microglia community. The results align with previous in vivo microglia data analysis (Maes et al. 2023, cited by the authors) from a different mitochondria labeling mouse model using fixed and live in vivo imaging in the nucleus accumbens (NAc) and the ventral tegmental area (VTA). The data presented is solid and broadly supports the claims. The most compelling data is that dynamic microglia processes often lack mitochondria, which is counterintuitive to likely energy-demanding microglial processes and the concept of mitochondria as an energy source. The manuscript contains a lot of experiments yet does not go beyond this observation. The manuscript remains mainly descriptive of the microglia and mitochondria phenotypes. The rationale for exploring different environmental alterations and their impact on the microglia- mitochondria network is not straightforwardly defined, leaving the impression of a phenotype search.

We thank the reviewer for recognizing that our data make valuable contributions to the field and that the presented data are solid. We are sorry to hear that the rationale for analyzing mitochondrial properties and how they relate to key microglial attributes was not easily appreciated and understood. Our launching point for this study was precisely the emerging

recognition that mitochondria do more than simply produce energy. In fact, for some cell types such as macrophages and other immune cells, this organelle plays essential roles in regulating dynamic changes in the overall functional state of the cell. Given that very little is known about microglial mitochondria, to explore whether there are important relationships between this organelle and overall microglial functional state, we aimed to analyze key properties of microglia and their mitochondria in instances where the functional state of these cells changes. We specifically chose examples of microglial phenotypic remodeling that operate at distinct temporal scales (laser lesion - minutes; LPS - hours to days; aging - months to years) and degrees of physiological vs pathological (baseline regional differences in microglial phenotype - physiological; baseline microglial tissue surveillance - physiological; normative aging - arguably physiological; laser lesion - pathological; LPS - pathological). We also chose pathological challenges that are distinct in modality (tissue injury vs. immunological/ inflammatory insult vs. lifespan tissue maintenance) Also by design, we wished to interrogate the relationship between microglial mitochondria and microglial attributes in paradigms where abundant data regarding microglial phenotype are available (from our own prior studies and the broader field). It is disappointing to hear that our study design leaves the impression of a phenotype search, but this means that we need to do a better job of articulating rationale behind study design in the abstract/introduction/results.

→ To address this concern, we have made changes to wording of the abstract, introduction, and beginning of each section in the results. We hope that the reviewers find that this highlights and justifies study design rationale more effectively.

We also acknowledge that much of our study is descriptive, but we would argue that it is precisely these types of studies that are needed to enable follow-up mechanistic studies in the field. Mitochondria can impact cellular function and phenotype through energy/ATP production, but they also impact cellular function and phenotype via calcium buffering (which impacts intracellular signaling pathways), mitochondrial-nucleus signaling (which regulates gene expression), production of reactive oxygen species (which have important intracellular signaling roles), production of metabolites used for protein posttranslational modifications, and lipid metabolism (which can shape cell membrane properties and dynamics). These are observations about mitochondrial function built up from many fields and disparate cell types. Without foundational guidepost observations regarding *which* microglial attributes are most tightly linked to *which* aspects of mitochondrial status, we can't, as a field, make focused hypotheses or design optimal mechanistic interventions that test causation. Given the state of knowledge in the field regarding microglial mitochondria, it would be a very high bar to request both the foundational mapping and the mechanistic manipulations in the same study.

Introduction:

The introduction needs to provide more confidence and perspective on the goal of the manuscript. Although the first paragraph introduces microglia and their ability to shift between functional states, the concept of a microglia state must be defined. The second paragraph introduces the expression of different cell surface receptors, finishing with a question of how microglia interpret different signals. From the third paragraph, it is not intuitive how mitochondria as "cell attributes" are involved in the previously introduced perspective of

functional states and interpreting different signals. What impact would the mitochondrial network remodeling have on the microglia? Furthermore, if the authors consider “cell attributes” as intrinsic properties of microglia to define their functional state, then this should be clearly outlined and justified. Lysosomes, for example, would be a more intuitive “attribute” to distinguish the different microglial states in the conditions tested within this manuscript compared to mitochondria.

In addition, the introduction across all three paragraphs consists mainly of attribute lists not followed up in the manuscript, such as high-fat diet, pollution exposure, and TLR4 and TGFBR2. Also, the list of mitochondrial parameters needs to be more focused since the study does not evaluate buffering calcium, metabolites, or interactions with other organelles.

Overall, the introduction only loosely connects mitochondria distribution and potential microglia state and does not provide a clear rationale for justifying the study.

We are sorry to hear that the reviewer did not feel that the introduction clearly and cohesively set up the rationale for the study.

→ **We have edited and expanded the introduction to try and address many of the points raised by the reviewer.** Some of the information (e.g. description of multiple functional roles for mitochondria as organelles) is necessary background information for readers who may not be familiar with the fact that mitochondria do more than simply produce ATP.

We acknowledge the adjustment of the introduction to cover the points. We did not question the necessity of explaining the mitochondria as organelles; yet, the initial draft suffered from being insufficiently concise listed facts that are not relevant for the manuscript.

We are glad to hear the reviewer feels we have adequately addressed concerns about introduction content and organization.

Results:

Fig 1/S1-2:

i) The authors performed a detailed verification of their mouse model to ensure that mitochondria were labeled. This data set looks convincing. However, without ultrastructural analysis, it cannot be ensured that MG-mitoGFP mice “accurately reveal the in vivo distribution of microglia mitochondria.” If the authors seek this accuracy as they write, they must perform ultrastructural analysis.

Use of this tool to accurately label mitochondria has been published previously (PMID 28132831). We would also like to refer the reviewer to studies by Pietramale et. al (BioRxiv 2024.10.15.618505) and Maes et. al (PMID 37731609) that also use a cre-dependent mitochondrial label (PhAM; JAX #018385) in microglia and find very similar distribution and morphological features of microglial mitochondria as what we report using mitoGFP (JAX #021429). Pietramale et. al also carry out analysis of human cortical EM datasets and found very similar somatic vs cell process distribution of microglial mitochondria as what is revealed by both cre-dependent fluorescent reporters.

Together with our acute brain slice experiments using TMRM to evaluate colocalization with microglial mitoGFP signal (Fig. S1J,K), we feel that we are justified in stating that this label accurately reveals the distribution of microglial mitochondria. In addition, we have carried out experiments in which bone marrow derived macrophages (BDBMs) were generated from *CX3CR1^{CreER/+};mitoGFP* mice. Visualization of all mitochondria in these *in vitro* preps is very clear and indicates that all TMRE labeled mitochondria in the BDBMs are also labeled by GFP. Moreover, all GFP+ structures are TMRE+ (i.e. GFP is not labeling other organelles) and the full extent of the TMRE labeled mitochondria is also captured by GFP (Reviewer Figure 6). If the reviewer feels strongly that these data are necessary to demonstrate fidelity of our mitochondrial labeling, we can incorporate these data into supplementary figures.

We agree that the accuracy of the cytochrome c oxidase Subunit VIIIa -based fluorescence in the mitochondria mouse model is valid. We also agree that the distribution of mitochondrial GFP tag co-localizes with the mitochondrial dye TMRM in both Fig S1J,K as well as the reviewer supplied figure 6. We appreciate the extra effort to confirm this observation.

Our initial comment was primarily directed at the text in the original version, which stated: "Together these results indicate that *MG-mitoGFP* mice accurately reveal the *in vivo* distribution of microglial mitochondria." (End of results paragraph 1). The authors have revised this to conclude: "Together these results indicate that *MG-mitoGFP* mice accurately label microglial mitochondria." to which we agree completely, and appreciate the text amendment to reflect the data.

We thank the reviewer for acknowledging our additional efforts to validate the accuracy of the mito-GFP tag and are glad they agree with our amended text.

ii) Since the main text states that mitoGFP recombination was substantial without tamoxifen injection, is the data in Fig. S1 without tamoxifen injection, while Fig. 1 with tamoxifen? For clarification, the authors should annotate in the figure legend if they have administered tamoxifen. If Fig. S1 is without tamoxifen, the authors should validate the model again with tamoxifen since tamoxifen could potentially affect the mitoGFP distribution.

Furthermore, for Fig. 1F, the analysis has been done without tamoxifen for rigorous analysis. This is surprising since microglia coverage across tissue is not overlapping, and confocal images should be sufficient to resolve the mitochondria signature for each microglia.

All experiments were carried out without tamoxifen injection except experiments to induce TFAM knockout from microglia.

→ We have added clarification to this effect in the results and methods sections. We have also added clarification to all figure legends about whether tamoxifen was used for that specific set of analyses. The decision to forgo tamoxifen injection in MG-MitoGFP mice was motivated by primarily by the observation that 80-90% of microglia showed recombination and mitoGFP expression in the absence of tamoxifen injection. With such a high percentage of cells expressing the mitoGFP tag, we did not feel that there was substantial risk that we were only sampling a subset of cells.

→ We have also carried out additional experiments to treat mice with tamoxifen and compare the properties of the mitoGFP+ population of microglia with and without tamoxifen treatment (Figure S1D-H and Reviewer Figure 2). Tamoxifen injection resulted in a slightly higher percentage of microglia expressing mitoGFP but did not result in any significant changes in GFP intensity, FSC, SSC, CX3CR1, CD45, or P2RY12, confirming that forgoing tamoxifen treatment does not result in biased sampling of only a subset of microglia.

With regard to experiments for Figure 1F using the triple transgenic *MG-MitoGFP;Ai14* animals, the presence of two cre-dependent transgenes results in sparser recombination and lower densities of mitoGFP+ only, TdTom+ only, and mitoGFP+TdTom+ cells. Although we agree with the reviewer that reasonable reconstruction of microglia can be carried out when the microglia are labeled with Iba1 staining or a universally expressed fluorophore (e.g. *CX3CR1-EGFP* mice), it can be tricky to unequivocally distinguish which terminal processes belong to which cell, particularly in brain regions where microglia present at higher density and show greater morphological complexity, such as the nucleus accumbens (NAc). Hence, we maintain that 3D reconstruction of individual cells is more rigorous when the cells are sparsely labeled.

We appreciate the clarification of tamoxifen administration in figure legends, as it makes it straightforward to follow the presented conditions. Given the additional experiments suggested by Reviewer 1 and presented in Reviewer Fig. 2, the authors have satisfied our initial comments.

We are glad that our revisions have satisfied this specific reviewer concern.

iii) How relevant is the correlation between mitochondrial number and microglia complexity/number of intersections? Are these just larger cells with more mitochondria? Based on the data in Fig 1E, the data point variation for each region is small. Furthermore, how have the microglia for reconstruction been selected?

What is the rationale for choosing the “maximum number of Sholl intersections”? Which radius size has been used? This can severely impact the readout. Furthermore, was this maximum always at a similar distance from the soma, or has there been some variability? This could explain potential variability in the data. The same question applies to Fig. S3E. The value used for sholl analysis in Fig. S3D is also unclear.

→ To clarify how cells were selected for 3D reconstruction the following text was added to the methods:

“To select microglia for 3D reconstruction, the brain region of interest was located and centered using only TH staining and anatomical landmarks with 20x objectives. After switching to 60x objectives, TdTom+mitoGFP+ microglia within that field of view were imaged. Slight X-Y adjustments were made, if needed, to bring a TdTom+mitoGFP+ microglial cell away from the edge of the FOV to avoid truncating processes. Cells were excluded from reconstruction if their somas fell near the top or bottom edge of the brain slice rather than being situated near the middle of the z-stack.”

Regarding maximum number of Sholl intersections - in the original submission we chose to highlight the relationship between this morphological metric and mitochondria because this

nicely captures the largest degree of branching complexity that a cell displays, irrespective of whether that branching complexity is close to the soma or further away at distal processes. As the reviewer points out, each individual cell will likely achieve its maximum number of intersections at a different radius from the soma. On its own, this metric doesn't offer any special advantage over other morphological metrics, which is why we also analyze total process length and sum of branch points (obtained directly from Imaris reconstructions).

→ In the revised manuscript, we highlight total process length and sum of branch points in the main figure (Fig. 1G-J) and also present data from max # of Sholl intersections in the table in Fig. S2J. Data from PFC microglia are now also presented in addition to NAc and VTA microglia.

Regarding data presented in Fig. S3D, total number of intersections is the sum of all intersections at all radius lengths. This metric is meant to capture the total branching complexity of the cell. → To clarify this point, the text of the figure legend has been changed to “Robust regression plot showing relationship between total mitochondrial volume per cell and microglial morphological complexity (total Sholl intersections across all radii).”

Regarding the biological significance or relevance of the correlations between microglial mitochondria and morphological metrics - Nothing is known about whether energy production or any other facet of mitochondrial function plays a critical role in determining the type of morphology that a microglial cell can adopt. If the mitochondria do play some role in regulating cell morphology, one might expect to find a correlation between mitochondrial number/volume and morphological complexity. If, instead, mitochondria were *only* important for regulating microglial production and secretion of trophic/inflammatory factors, there might be no correlation between the morphological complexity of an individual cell and the abundance of mitochondria in that cell. In Reviewer Figure 6, above, one can appreciate that a fairly amorphous Bone Marrow Derived Macrophage (BMDM) has tons of mitochondria. But it would be difficult to relate mitochondrial abundance to morphological *complexity* of BMDMs in any fashion. In this regard, the relationship between mitochondria and microglial morphological complexity

may be more akin to what is observed in neurons, where mitochondria are linked to dendritic spine morphology and remodeling (PMID 32714603).

→ Elaboration of some of these points has been added to the discussion.

The addition of methodological details, microglial metrics including branch points and process length, and the further discussion of these aspects satisfy our initial comments.

Minor comment: Please clarify in the method section whether for the Sholl radius 1 um has been used.

We have clarified in the methods section that for Sholl analysis “each concentric circle was 1 um apart”.

iv) The choice for looking at the NAc and VTA needs better clarification, also in perspective for the later analysis with LPS and aging. Are those brain regions particularly vulnerable? → We

have added text throughout the results section to explain in greater depth our rationale for focusing on these two brain regions. These additions include:

“In fixed tissue from young adult (2mo) double transgenic *CX3CR1^{CreER/+};mitoGFP* mice (*MG-mitoGFP* mice hereafter), numerous GFP+ structures could be observed throughout the somas and cell processes of microglia in the nucleus accumbens (NAc) and ventral tegmental area (VTA), two interconnected brain regions where we have extensively mapped microglial phenotypes across the lifespan (**Fig. 1B** and **Fig. S1A,B**)”

“Microglia show prominent differences in cell density, branching complexity, and gene expression across the NAc and VTA³⁰. To determine if these regional differences in baseline microglial attributes were accompanied by regional differences in microglial mitochondria, we carried out high resolution confocal imaging and 3D reconstruction of both microglia and their mitochondria in the NAc and VTA of young adult mice (**Fig. 1B** and **Fig. S2A**).”

“We also showed previously that microglial responses to aging vary across brain region, with VTA microglia displaying proliferative and inflammatory changes months before microglia in the NAc. Whether there are links between mitochondrial state and early VTA microglial aging has also not been investigated.”

We appreciate the changes that improved the clarity of the manuscript.

We are pleased to hear that the reviewer feels that clarity was improved.

v) Based on the figure legend description in Fig 1. It is understood that microglial volume is calculated per field of view (tissue coverage)- and mitochondrial number, volume, and mass are then calculated based on this FOV microglial volume. However, the text states, “greater number of mitochondria relative to cell volume, as well as a greater total mitochondrial volume relative to cell volume (mitochondrial mass)”. If the quantification is relative to total microglial volume per FOV, and not on a per-cell basis, stating ‘cell volume’ is confusing. For example, Fig. 1D-E, is the relative ‘microglia volume’ in 1D the same denominator as ‘cell volume’ in 1E? In addition, does the number of mitochondria within a FOV bring us important information? Since many microglial processes not connected to a cell body are present within a brain volume, these averages do not indicate information on a per-cell basis. The authors should identify precise nomenclature and analysis methods, describe this in detail in the methods, and use the terminology consistently throughout the manuscript. We performed analyses using a FOV approach to obtain information about microglia as a

cell population in that particular brain region. We feel that carrying out FOV analyses AND individual cell focused analyses is more rigorous than either approach alone. FOV analyses can smooth out the effects of cell-to-cell variation and remove concerns about any unintended bias in selecting individual cells for analysis.

→ We have updated descriptions in the methods, figure legends and text to ensure consistency and avoid confusion about the distinction between FOV-based versus individual cell-based analyses.

These additions satisfy our initial comments.

We are glad that our revisions have satisfied this specific reviewer concern.

vi) The mitochondrial field focuses more on measuring network connectivity, i.e., fragmented or connected networks. This metric is not shown throughout the manuscript but would be an essential feature to describe when considering mitochondrial structural network changes. The reviewer is correct that measures of “fragmentation” vs interconnected “networks” are commonly seen in the mitochondrial field. The vast majority of these studies are carried out in cultured cells where genetically-encoded mitochondrial reporters and/or mitochondrial dyes can easily be used with fantastic signal-to-noise, allowing automated reconstruction and analysis of mitochondria using machine learning platforms like Avia. In fixed tissue, however, even with optimal confocal imaging, distinguishing whether a cluster of mitochondria are fully, physically interconnected or separate is much more challenging. This is particularly true in the cell soma, where mitochondria are often very closely packed. For this reason, we feel it is more accurate to analyze mitochondrial elongation, mitochondrial sphericity, mitochondrial number, and mitochondrial mass which are also widely used in the mitochondrial field and to avoid classifications like “networked” or “fragmented,” which carry functional implications.

→ To address the reviewer’s point, we have included new material in the discussion, commenting on the limitations of light microscopy for resolving whether mitochondria are truly networked or discrete, and additional future approaches that can be used to determine whether small mitochondria have high ROS / low TMRM and are likely to be undergoing mitophagy (which is often assumed when employing the term fragmented) versus they have undergone fission to facilitate trafficking.

These additions satisfy our initial comments.

We are glad that our revisions have satisfied this specific reviewer concern.

Fig 2:

i) The conclusion of Fig. 2C needs to be clarified. If VTA and NAc show the same motility, how can the conclusion be made that the VTA is surveying less based on their morphology? We originally advanced this interpretation because, *in vivo*, VTA microglia are less dense and less branched than NAc microglia (Fig 1C,G,I and PMID: 28689984 and PMID: 35396327). Hence, if microglia in both brain regions show similar motility, it will take VTA microglia longer to execute a complete surveillance of the surrounding tissue than the more abundant, more highly branched NAc microglia.

→ However, given that our observations of motility are made in an acute brain section preparation, we removed this statement to address any concerns that it is too speculative.

This amendment satisfies our initial comments.

We are glad that our revisions have satisfied this specific reviewer concern.

ii) Do the motile processes contain mitochondria? Or are motile mitochondria only found in primary processes that are more stable? (also related to Fig. 3). The reviewer brings up the very interesting question of how subcellular location relates to the motility of individual microglial cell processes and the motility of the mitochondria themselves.

→ To address this point, we have carried out additional analyses focusing on individual microglial cell processes (Fig. 2J). Similar to our findings at the level of the whole microglial cell, this analysis revealed that there was not a significant correlation between the abundance of mitochondria in a primary cell branch and the motility of that branch. There was also no significant correlation between the motility of mitochondria in a branch and the motility of the branch itself (or vice versa). Hence, it is not the case that microglial process branches that have more mitochondria exhibit any heightened capacity for cell process remodeling, at least over the time frame we are analyzing. We draw the reviewer's attention to the study by Pietramale et. al (BioRxiv 2024.10.15.618505) co-submitted with our study that suggest that this result may differ if focused on even smaller branchlets. By focusing on smaller branchlets (rather than entire arbors branching out from one primary process), they found that microglial branchlets with fewer mitochondria were more motile. There could also be important brain region differences in this mitochondria-branch motility relationship. **This will be an important area for the field to investigate further, and we elaborate on some of these considerations further in the discussion.**

We appreciate the consideration.

Minor comment: The outlined description fits into a discussion and the authors should cite the co-submitted study/biorxiv Pietramale, which investigates the question in more depth.

We have added descriptions of the findings and implications of the Pietramale et. al study to the discussion.

Fig 3:

i) This data is very interesting and raises exciting new questions; however, no quantification supports the concluding statement.

We are glad that the reviewer finds these data intriguing.

→ In this resubmission, we have included quantification of microglial process motility and mitochondrial motility following laser lesion (Fig. 3F-I). These analyses revealed that abundance of microglial processes close to the lesion increases over time, whereas abundance of microglial mitochondria close to the lesion does not increase, confirming our qualitative observations from the initial manuscript submission.

We appreciate the addition of quantification as it significantly strengthens the figure.

We are glad that our revisions have satisfied this specific reviewer concern.

ii) At this resolution, it seems difficult to quantify and confirm that the mitochondrial networks remain stable after focal injury. Can the authors image at the resolution seen in Fig 2 and provide data analysis of mitochondrial network motility before and after injury? Do all mitochondria stop moving, or only a few travel within the extending process?

In this set of experiments, we elected to image a larger field of view in order to sample a larger number of microglia to determine how they and their mitochondrial networks

respond to laser lesion. While repeat experiments to image smaller fields of view at higher resolution could be carried out, we felt this would best be accomplished via *in vivo* multiphoton

imaging that would also permit mitochondria to be monitored over a much longer time frame. Using such *in vivo* multiphoton approaches, Pietramale et. al (BioRxiv 2024.10.15.618505) found similar responses of microglia and their mitochondria in cortex, with negligible mitochondrial remodeling within 40min of the laser lesion. Intriguingly, they *did* observe mitochondrial remodeling at 6hrs following lesion.

We acknowledge the shortcomings, which should be outlined in the discussion and referenced to the Pietramale study.

We have added descriptions of the findings and implications of the Pietramale et. al study to the discussion.

Fig 4/5:

i) If mitochondria are not linked to microglial tissue surveillance and injury, as the authors conclude in Fig. 2J and Fig. 3, respectively, how do they rationalize using LPS? Based on live imaging, mitochondria are not linked in a “direct fashion to baseline surveillance” or injury-induced damage. Why would it suddenly be related to an inflammatory response induced by LPS? We are sorry to hear that the rationale for utilizing LPS to further interrogate the relationship between mitochondria and microglial responses to tissue challenge was not clearly communicated. Laser lesion is a focal tissue injury that elicits a rapid (within minutes) response from microglia. Purinergic signaling from the injured tissue to microglial P2RY12 receptors plays a critical role in this rapid injury response. Our data indicate that rapid extension of microglial processes is not accompanied by rapid remodeling of mitochondrial networks (at least on a morphological level). Given this result, to continue testing whether mitochondria play an important role in regulating microglial phenotypic remodeling in response to pathological challenges, it was desirable to examine microglial responses that occur over a longer time scale (hours to days) as well as microglial responses to a distinct modality of challenge (inflammatory challenge rather than tissue damage).

→ As described above, we have made changes to wording of the abstract, introduction, and beginning of each section in the results to better explain rationale for study design.

We appreciate the changes that improved the clarity of the manuscript.

We are pleased to hear that the reviewer feels that clarity was improved.

ii) QPCR of *Dnm1* is used to indicate fission- however, this data should also be reflected in the image analysis in Fig 5. Could the authors indicate changes in the median volume of organelles (per cell) to compare alongside the mitochondrial number and aspect ratio? If the mitochondria are indeed elongated as indicated by an aspect ratio measure, the median volume of organelles per cell should increase (aligning with a mass increase) since the number of mitochondria did not change between saline and LPS for both brain regions.

The reviewer is correct that dynamin-related protein 1 (DRP1; encoded by the gene *Dnm1*) plays a key role in promoting mitochondrial fission. In both our FACS-based analyses in Fig. 4F and Fig. 8D, the prominent increase in GFP median fluorescence intensity suggests that mitochondrial biogenesis accompanies microglial responses to LPS challenge.

The increase in mitochondrial mass detected via histology (**Fig. 5B**), further indicates an overall increase in mitochondria within microglia following LPS challenge. The reviewer is correct that, in general, mean volume of individual mitochondria trended toward increases

at 24hrs following LPS (**Reviewer Figure 7**), particularly in the NAc, so the FACS-based increase in GFP and the increase in mitochondrial mass via histology is likely driven by larger, longer mitochondria.

These findings don't necessarily represent an inconsistency with the increase in *Dnml1* transcript found by qPCR. Indeed, *Dnml1* may be getting upregulated precisely because there are larger, longer mitochondria and increased fission will be needed to return the mitochondrial network to its pre-LPS state.

The image analysis data support the concept that there is an increase in mitochondrial mass, accompanied by more elongated mitochondria. The support of the reviewer figure 7 helps clarify this point. The finding that mitochondria are elongated after LPS challenge is in contrast to the reported effect of LPS on microglial mitochondria in the *in vitro* setting PMID: 30637805, which is an interesting point to highlight in the discussion.

We have included a statement in our discussion highlighting this difference between *in vitro* results and our findings and suggesting potential reasons for this discrepancy.

iii) Do the authors have the details of the post-hoc analysis of 5B-D?

As described in the methods in the original submission, all one-way and two-way ANOVAs were performed with Bonferroni correction for multiple comparisons being applied to all post-hoc comparisons.

→ **We have added information about post-hoc corrections to the figure legends.**

We appreciate the addition.

We are glad that our revisions have satisfied this specific reviewer concern.

iv) Technical aspect related to sampling for flow cytometry. The method section must clarify whether the samples were processed sequentially or in batches. Furthermore, information related to the total time from tissue removal to sorting must be included. It has been shown that the microglia transcriptional profile adapts within 30 minutes of incubation on ice (Marsh 2022. DOI: 10.1038/s41593-022-01022-8). This can result in misleading results. Therefore, the authors should add the animal grouping strategy and total processing time to the methods and, if necessary, repeat the experiments when the timing is longer than 30 min.

The authors are correct that microglia are extremely sensitive and adaptable cells that will respond to tissue dissociation. Hence, care needs to be taken in the tissue processing workflow to avoid artifacts. We are very familiar with the Marsh et. al paper to which the reviewer refers. In this study, the key finding is that exposing microglia to elevated temperature (37°C) to carry out enzymatic digestion causes them to exhibit altered transcriptional profiles. See Figure 1C of the paper. This panel shows that you can prevent aberrant microglial transcriptional signatures by keeping the entire workflow at 4°C (i.e. on ice) OR by including transcriptional inhibitors. For

all experiments in our study, the entire workflow was carried out at 4 °C as described in the methods. In our prior studies (PMID: 28689984), RNAseq confirms that this workflow does not elicit upregulation of inflammatory factors typically associated with microglial reactivity. Moreover, this workflow does not induce increased expression of the specific genes highlighted in the Marsh et. al study (**Reviewer Figure 8**).

Of note, none of the workflows in Marsh et al are completed in under 30min. All of their tissue processing workflows take 70-130min as indicated by their workflow diagrams in Figures 1 and 2. 30 minutes refers to the portion of their workflow during which microglia were exposed to elevated temperature for enzymatic dissociation *OR* maintained at 4 degrees, with the latter treatment preserving *in vivo* gene expression signatures. Moreover, their stated workflow time does not include time for sorting, but rather indicates time prior to when sorting begins.

→ **As requested by the reviewer, we have added the following information to the methods regarding our sample processing:**

“Total sample processing time from brain dissection to completion of FACS purification ranged from 2-4hrs depending on sort order of individual samples. Samples were collected in batches with each day of sorting including a young adult, saline injected, or TFAM-wt control mouse. Sort order of samples from specific ages, brain regions, treatment conditions, or genotypes was interleaved to avoid introduction of bias related to tissue processing length. FACS-isolation itself was carried out at 4°C. FACS-isolation of all samples for a given set of experiments was completed prior to downstream processing to allow simultaneous RNA isolation and cDNA preparation as described below.”

We appreciate the addition.

We are glad that our revisions have satisfied this specific reviewer concern.

v) How was the correlation analysis performed in Fig. 4D? Cherry-picked genes were connected to putative microglial states, an aspect that is also not well-defined, introducing a bias. Also, the explanation of the “few CD68 correlations with other genes” does not seem solid (Fig. S4B) since the heatmap shows much more correlation than any of the other plots in Fig. 4.

To test for potential correlations between microglial genes reflective of overall microglial functional state and our entire panel of 14 mitochondria-relevant genes, we carried out linear regression analyses for each microglial gene and all of the mitochondrial genes. For considerations of space, we included results for 6 / 9 microglial genes in the main figure and an additional microglial gene in Figure S4. We respectfully disagree that showing data from 7/9 microglial genes constitutes “cherry-picking” genes.

It also seems that we have not done an optimal job of explaining the significance of the *Cd68* result. Expression levels of *Cd68* are indeed correlated with expression levels of multiple other microglial genes (*Tgfβ1*, *Cx3cr1*, *P2ry12*, **Fig. S4D**). However, the important point here is that this degree of correlation was not modulated by LPS challenge (i.e. these significant correlations were present at baseline and remained present at both 4hrs and 24hrs post-LPS). Also importantly, expression levels of *Tnfα* and *Il1β* were correlated with expression levels of multiple mitochondrial genes, particularly at 4hrs post LPS. Yet, no correlation was observed

between *Cd68* expression levels and those of *Tnfa* or *Il1b*. Together, these findings suggest a more robust relationship between the state of mitochondria (as organelles) and acute microglial responses to LPS compared to the relationship between lysosomes and acute microglial responses to LPS. (**Please note that in response to Reviewer 2's comment, we re-executed our linear regression analyses using Robust regression with false discovery rate control; using this more rigorous approach, some*

of the correlations that were found to be significant in the original submission are no longer significant in this revised analysis).

→ To address the reviewer's comment, we have modified the text of the results:

“Moreover, examination of *Cd68*, a key microglial lysosome gene, hints that mitochondria, as organelles, may be uniquely positioned to regulate microglial functional state during inflammatory challenge. While LPS treatment increased the correlation between most mitochondria-relevant genes and key microglial function genes (*Cd45*, *Cx3cr1*, *P2ry12*, *Tgfβ1*)(**Fig. 4D**), LPS treatment did not modulate correlations between *Cd68* and *Cx3cr1*, *P2ry12*, *Tgfβ1* (correlations between *Cd68* and *Cd45* increased at 24hrs post LPS)(**Fig. S4D** and **Table S4**). Moreover, while multiple mitochondria-relevant genes were positively correlated with expression of the microglial pro-inflammatory factors *Tnfa* and *Il1b* at 4hrs post-LPS, *Cd68* expression was not correlated with *Tnfa* and *Il1b* in any of the conditions examined (**Fig. S4D**).”

We appreciate the addition.

We are glad that our revisions have satisfied this specific reviewer concern.

Fig 6:

Microglia are known to show a sex-specific response also in adulthood and, especially, during aging. Therefore, the authors must precisely annotate the data plots, indicating which data points come from male or female animals.

We agree that probing for sex-differences is important for the microglial field and we included both male and female mice in all our analyses.

→ To address the reviewer's comment, we have added a supplementary figure that replicates key findings from each of our analyses with the annotation of datapoints that come from male and female mice (Fig. S9) We have also added commentary to the discussion that future research should continue to investigate possible sex differences in microglial mitochondria.

We appreciate the addition, which sets an essential criterion for the transparency of the research data and will enable the community to acknowledge sex-specific differences to be follow up in the future, especially in neurodevelopmental and neurodegenerative diseases.

Minor point: For sampling with at least 3males/females, it would be appreciated to see which metrics did carry significant sex-specific differences.

We agree that this is an important topic for future investigation and we anticipate delving into this in greater detail in future work.

Fig 7:

i) Related to the mitochondrial quantification in 7D: Is this data represented as an average volume of all organelles within the cell? If yes, it would be better to use a median, as small mitochondria can skew the average value and misrepresent the data. **Figure 7D shows the sum total volume of all the mitochondria in a field of view (FOV). This value was obtained for 3 FOVs for each mouse to compute a mouse average, which is displayed in the graph. → We have adjusted the label of the y axis and the description of this panel in the figure legend to clearly specify that these are FOV measurements. We have also added FOV median mitochondrial volume and FOV maximum volume of an individual mitochondrion to Figure S7F.** This highlights that increases in the total number of mitochondria within the

FOV are driving increases in total FOV mitochondrial volume, rather than increases in the size of individual mitochondria.

ii) Mitochondrial analysis: In Fig. 7, the analysis focused on volume, number, and sphericity, whereas earlier figures describing the mitochondria in fixed tissue focused on mass, number, aspect ratio, volume, sphericity. To allow comparison between the datasets, the authors should be consistent and include all metrics for all figures. Furthermore, the methods section needs to include a detailed description of the analysis method for each parameter.

→ **We have added analysis of mitochondrial mass to this figure (Fig. 7K). We have expanded the description in the methods of how these metrics were generated within Imaris analysis software.**

This amendment satisfies our initial comments.

We are glad that our revisions have satisfied this specific reviewer concern.

Fig 8:

How do the microglia change after TFAM KO? A two-month waiting period may allow time for compensatory mechanisms to circumvent initial damage. **Understanding how TFAM KO impacts microglia is indeed the primary goal of these experiments. → In addition to the histology in the original submission (previous Fig 7I, now Fig 7H) which showed that microglial tissue coverage within the field of view (FOV) increases after microglial TFAM KO, we now show that morphological complexity of individual microglia (analyzed via Sholl analysis) increases in MG-TFAM KO mice (Fig. 7I).**

→ **We also show that microglial density is unchanged (Fig. 7G).**

→ **In addition to the qPCR in the original submission that showed multiple changes in microglial gene expression in MG-TFAM KO mice (previous Fig. 8C, now Fig. 8B), we now also show that protein level expression of CD45 and CX3CR1 (MFI during FACS) are unchanged in MG-TFAM KO microglia. Cell size (FSC) and internal complexity (SSC) are increased in microglia from MG-TFAM KO mice (Fig. 8A).**

→ **Finally, in this resubmission, we challenge MG-TFAM KO mice with LPS and utilize FACS and qPCR to assess the capacity of the cells to mount a mitochondrial and cellular response to this challenge (see detailed description in response to reviewer #1, point 4).**

As described in the results, we elected to wait for 2 months following tamoxifen injection for the following reasons. 1) This will allow for any peripheral monocytes/macrophages impacted by this manipulation to cycle out and be replaced. 2) Mitochondrial proteins are among the most long-lived proteins in the CNS (PMID: 30315172). Hence, it is estimated that approximately 1 month would be needed for existing mitochondrial proteins to cycle out. Moreover, TFAM exerts its broad effects on the mitochondria by altering stability of the mitochondrial DNA. This impacts capacity of the mitochondria to produce key components of the electron transport chain, with additional downstream consequences for other aspects of mitochondrial function. In line with this, virtually all studies that have used cell-specific TFAM KO have either used constitutive Cre lines (PMID 31269437, PMID 17227870, PMID 9916807, PMID 25005176, PMID 23168219, PMID 18945718), or waited at least a month (PMID 35165165, PMID 40210116) . The reviewer is correct that some compensatory

responses may be elicited and some of the changes we observe in expression of nuclear-encoded mitochondrial components may represent attempts at compensation. Nonetheless, we feel that our choice of 2 months is well justified, and it will fall to future studies to tease apart direct vs compensatory effects on the organelle.

Discussion:

In line with the loose introduction, the discussion is also only superficial. It mostly repeats the results without putting the research data into the context of the literature. For example, a reported result is increased mitochondrial mass. What does increased mitochondrial mass imply? The manuscript would enormously benefit from a complete revision of the introduction and discussion to make it coherent.

We are sorry to hear that the reviewer did not find the discussion informative.

→ **We have updated the discussion to include some potential implications of key observations based on what is known in the literature from mitochondrial behavior in other cell types.** Multiple follow-up studies will be required to fully elucidate what a particular mitochondrial attribute means for microglial function in a given context. Our aim with this study was to inspire those types of follow-up studies by revealing how closely linked the general state of the organelle is with the overall state of the cell.

Overstatements:

The manuscript suffers from several overstatements and preliminary conclusions, which must be fixed. A few examples are listed below:

i) “First time” in the introduction. **We have removed this type of wording from the manuscript.**

ii) The authors regularly write “in vivo” within their manuscript, yet it becomes quickly apparent that the majority of the experiments are performed in a fixed tissue environment. This should be clarified in the text to avoid misunderstandings. **We have adjusted wording in the manuscript to avoid confusion as the reviewer suggests.**

Overall, the revised manuscript is significantly improved in clarity, making it accessible to a broader readership. It offers insights into mitochondria in microglia across a range of physiological, pathological, and aging conditions, providing the foundation for many future

studies. Our specific concerns have been adequately addressed, with only a few minor points remaining.

We are pleased to hear that the reviewer feels that clarity was improved.

Reviewer #5 (Remarks to the Author):

We thank the reviewer for their time and feedback.

Responses to our section:

Reviewer #4 (Remarks to the Author):

The manuscript presents valuable findings about the location of mitochondria in microglia and will, therefore, have some implications for the microglia community. The results align with previous in vivo microglia data analysis (Maes et al. 2023, cited by the authors) from a different mitochondria labeling mouse model using fixed and live in vivo imaging in the nucleus accumbens (NAc) and the ventral tegmental area (VTA). The data presented is solid and broadly supports the claims. The most compelling data is that dynamic microglia processes often lack mitochondria, which is counterintuitive to likely energy-demanding microglial processes and the concept of mitochondria as an energy source. The manuscript contains a lot of experiments yet does not go beyond this observation. The manuscript remains mainly descriptive of the microglia and mitochondria phenotypes. The rationale for exploring different environmental alterations and their impact on the microglia-mitochondria network is not straightforwardly defined, leaving the impression of a phenotype search.

We thank the reviewer for recognizing that our data make valuable contributions to the field and that the presented data are solid. We are sorry to hear that the rationale for analyzing mitochondrial properties and how they relate to key microglial attributes was not easily appreciated and understood. Our launching point for this study was precisely the emerging recognition that mitochondria do more than simply produce energy. In fact, for some cell types such as macrophages and other immune cells, this organelle plays essential roles in regulating dynamic changes in the overall functional state of the cell. Given that very little is known about microglial mitochondria, to explore whether there are important relationships between this organelle and overall microglial functional state, we aimed to analyze key properties of microglia and their mitochondria in instances where the functional state of these cells changes. We specifically chose examples of microglial phenotypic remodeling that operate at distinct temporal scales (laser lesion - minutes; LPS - hours to days; aging - months to years) and degrees of physiological vs pathological (baseline regional differences in microglial phenotype - physiological; baseline microglial tissue surveillance - physiological; normative aging - arguably physiological; laser lesion - pathological; LPS - pathological). We also chose pathological challenges that are distinct in modality (tissue injury vs. immunological/ inflammatory insult vs. lifespan tissue maintenance) Also by design, we wished to interrogate the relationship between microglial mitochondria and microglial attributes in paradigms where abundant data regarding microglial phenotype are available (from our own prior studies and the broader field). It is disappointing to hear that our study design leaves the impression of a phenotype search, but this means that we need to do a better job of articulating rationale behind study design in the abstract/introduction/results.

→ To address this concern, we have made changes to wording of the abstract, introduction, and beginning of each section in the results. We hope that the reviewers find that this highlights and justifies study design rationale more effectively.

We also acknowledge that much of our study is descriptive, but we would argue that it is precisely these types of studies that are needed to enable follow-up mechanistic studies in the field. Mitochondria can impact cellular function and phenotype through energy/ATP production, but they also impact cellular function and phenotype via calcium buffering (which impacts intracellular signaling pathways), mitochondrial-nucleus signaling (which regulates gene expression), production of reactive oxygen species (which have important intracellular signaling roles), production of metabolites used for protein posttranslational modifications, and lipid metabolism (which can shape cell membrane properties and dynamics). These are observations about mitochondrial function built up from many fields and disparate cell types. Without foundational guidepost observations regarding *which* microglial attributes are most tightly linked to *which* aspects of mitochondrial status, we can't, as a field, make focused hypotheses or design optimal mechanistic interventions that test causation. Given the state of knowledge in the field regarding microglial mitochondria, it would be a very high bar to request both the foundational mapping and the mechanistic manipulations in the same study.

Introduction:

The introduction needs to provide more confidence and perspective on the goal of the manuscript. Although the first paragraph introduces microglia and their ability to shift between functional states, the concept of a microglia state must be defined. The second paragraph introduces the expression of different cell surface receptors, finishing with a question of how microglia interpret different signals. From the third paragraph, it is not intuitive how mitochondria as “cell attributes” are involved in the previously introduced perspective of functional states and interpreting different signals. What impact would the mitochondrial network remodeling have on the microglia? Furthermore, if the authors consider “cell attributes” as intrinsic properties of microglia to define their functional state, then this should be clearly outlined and justified. Lysosomes, for example, would be a more intuitive “attribute” to distinguish the different microglial states in the conditions tested within this manuscript compared to mitochondria.

In addition, the introduction across all three paragraphs consists mainly of attribute lists not followed up in the manuscript, such as high-fat diet, pollution exposure, and TLR4 and TGFBR2. Also, the list of mitochondrial parameters needs to be more focused since the study does not evaluate buffering calcium, metabolites, or interactions with other organelles.

Overall, the introduction only loosely connects mitochondria distribution and potential microglia state and does not provide a clear rationale for justifying the study.

We are sorry to hear that the reviewer did not feel that the introduction clearly and cohesively set up the rationale for the study.

→ **We have edited and expanded the introduction to try and address many of the points raised by the reviewer.** Some of the information (e.g. description of multiple functional roles for mitochondria as organelles) is necessary background information for readers who may not be familiar with the fact that mitochondria do more than simply produce ATP.

We acknowledge the adjustment of the introduction to cover the points. We did not question the necessity of explaining the mitochondria as organelles; yet, the initial draft

suffered from being insufficiently concise listed facts that are not relevant for the manuscript.

Results:

Fig 1/S1-2:

i) The authors performed a detailed verification of their mouse model to ensure that mitochondria were labeled. This data set looks convincing. However, without ultrastructural analysis, it cannot be ensured that MG-mitoGFP mice “accurately reveal the *in vivo* distribution of microglial mitochondria.” If the authors seek this accuracy as they write, they must perform ultrastructural analysis.

Use of this tool to accurately label mitochondria has been published previously (PMID 28132831). We would also like to refer the reviewer to studies by Pietramale et. al (BioRxiv 2024.10.15.618505) and Maes et. al (PMID 37731609) that also use a cre-dependent mitochondrial label (PhAM; JAX #018385) in microglia and find very similar distribution and morphological features of microglial mitochondria as what we report using mitoGFP (JAX #021429). Pietramale et. al also carry out analysis of human cortical EM datasets and found very similar somatic vs cell process distribution of microglial mitochondria as what is revealed by both cre-dependent fluorescent reporters.

Together with our acute brain slice experiments using TMRE to evaluate colocalization with microglial mitoGFP signal (**Fig. S1J,K**), we feel that we are justified in stating that this label accurately reveals the distribution of microglial mitochondria. In addition, we have carried out experiments in which bone marrow derived macrophages (BDBMs) were generated from *CX3CR1^{CreER/+};mitoGFP* mice. Visualization of all mitochondria in these *in vitro* preps is very clear and indicates that all TMRE labeled mitochondria in the BMDMs are also labeled by GFP. Moreover, all GFP+ structures are TMRE+ (i.e. GFP is not labeling other organelles) and the full extent of the TMRE labeled mitochondria is also captured by GFP (**Reviewer Figure 6**). If the reviewer feels strongly that these data are necessary to demonstrate fidelity of our mitochondrial labeling, we can incorporate these data into supplementary figures.

We agree that the accuracy of the cytochrome c oxidase Subunit VIIIa -based fluorescence in the mitochondria mouse model is valid. We also agree that the distribution of mitochondrial GFP tag co-localizes with the mitochondrial dye TMRE in both Fig S1J,K as well as the reviewer supplied figure 6. We appreciate the extra effort to confirm this observation.

Our initial comment was primarily directed at the text in the original version, which stated: “Together these results indicate that *MG-mitoGFP* mice accurately reveal the *in vivo* distribution of microglial mitochondria.” (End of results paragraph 1). The authors have revised this to conclude: “Together these results indicate that *MG-mitoGFP* mice accurately label microglial mitochondria.” to which we agree completely, and appreciate the text amendment to reflect the data.

ii) Since the main text states that mitoGFP recombination was substantial without tamoxifen injection, is the data in Fig. S1 without tamoxifen injection, while Fig. 1 with tamoxifen? For clarification, the authors should annotate in the figure legend if they have administered tamoxifen. If Fig. S1 is without tamoxifen, the authors should validate the model again with tamoxifen since tamoxifen could potentially affect the mitoGFP distribution.

Furthermore, for Fig. 1F, the analysis has been done without tamoxifen for rigorous analysis. This is surprising since microglia coverage across tissue is not overlapping, and confocal images should be sufficient to resolve the mitochondria signature for each microglia. All experiments were carried out without tamoxifen injection except experiments to induce TFAM knockout from microglia.

→ We have added clarification to this effect in the results and methods sections. We have also added clarification to all figure legends about whether tamoxifen was used for that specific set of analyses. The decision to forgo tamoxifen injection in MG-MitoGFP mice was motivated primarily by the observation that 80-90% of microglia showed recombination and mitoGFP expression in the absence of tamoxifen injection. With such a high percentage of cells expressing the mitoGFP tag, we did not feel that there was substantial risk that we were only sampling a subset of cells.

→ We have also carried out additional experiments to treat mice with tamoxifen and compare the properties of the mitoGFP+ population of microglia with and without tamoxifen treatment (Figure S1D-H and Reviewer Figure 2). Tamoxifen injection resulted in a slightly higher percentage of microglia expressing mitoGFP but did not result in any significant changes in GFP intensity, FSC, SSC, CX3CR1, CD45, or P2RY12, confirming that forgoing tamoxifen treatment does not result in biased sampling of only a subset of microglia.

With regard to experiments for Figure 1F using the triple transgenic *MG-MitoGFP;Ai14* animals, the presence of two cre-dependent transgenes results in sparser recombination and lower densities of mitoGFP+ only, TdTom+ only, and mitoGFP+TdTom+ cells. Although we agree with the reviewer that reasonable reconstruction of microglia can be carried out when the microglia are labeled with Iba1 staining or a universally expressed fluorophore (e.g. *CX3CR1-EGFP* mice), it can be tricky to unequivocally distinguish which terminal processes belong to which cell, particularly in brain regions where microglia present at higher density and show greater morphological complexity, such as the nucleus accumbens (NAc). Hence, we maintain that 3D reconstruction of individual cells is more rigorous when the cells are sparsely labeled.

We appreciate the clarification of tamoxifen administration in figure legends, as it makes it straightforward to follow the presented conditions. Given the additional experiments suggested by Reviewer 1 and presented in Reviewer Fig. 2, the authors have satisfied our initial comments.

iii) How relevant is the correlation between mitochondrial number and microglia complexity/number of intersections? Are these just larger cells with more mitochondria? Based on the data in Fig 1E, the data point variation for each region is small. Furthermore, how have the microglia for reconstruction been selected? What is the rationale for choosing the “maximum number of Sholl intersections”? Which radius size has been used? This can severely impact the readout. Furthermore, was this maximum always at a similar distance from the soma, or has there been some variability? This could explain potential variability in the data. The same question applies to Fig. S3E. The value used for sholl analysis in Fig. S3D is also unclear.

→ To clarify how cells were selected for 3D reconstruction the following text was added to the methods:

“To select microglia for 3D reconstruction, the brain region of interest was located and centered using only TH staining and anatomical landmarks with 20x objectives. After switching to 60x objectives, TdTom+mitoGFP+ microglia within that field of view were imaged. Slight X-Y adjustments were made, if needed, to bring a TdTom+mitoGFP+ microglial cell away from the edge of the FOV to avoid truncating processes. Cells were excluded from reconstruction if their somas fell near the top or bottom edge of the brain slice rather than being situated near the middle of the z-stack.”

Regarding maximum number of Sholl intersections - in the original submission we chose to highlight the relationship between this morphological metric and mitochondria because this nicely captures the largest degree of branching complexity that a cell displays, irrespective of whether that branching complexity is close to the soma or further away at distal processes. As the reviewer points out, each individual cell will likely achieve its maximum number of intersections at a different radius from the soma. On its own, this metric doesn't offer any special advantage over other morphological metrics, which is why we also analyze total process length and sum of branch points (obtained directly from Imaris reconstructions).

→ In the revised manuscript, we highlight total process length and sum of branch points in the main figure (Fig. 1G-J) and also present data from max # of Sholl intersections in the table in Fig. S2J. Data from PFC microglia are now also presented in addition to NAc and VTA microglia.

Regarding data presented in Fig. S3D, total number of intersections is the sum of all intersections at all radius lengths. This metric is meant to capture the total branching complexity of the cell. → To clarify this point, the text of the figure legend has been changed to “Robust regression plot showing relationship between total mitochondrial volume per cell and microglial morphological complexity (total Sholl intersections across all radii).”

Regarding the biological significance or relevance of the correlations between microglial mitochondria and morphological metrics - Nothing is known about whether energy production or any other facet of mitochondrial function plays a critical role in determining the type of morphology that a microglial cell can adopt. If the mitochondria do play some role in regulating cell morphology, one might expect to find a correlation between mitochondrial number/volume and morphological complexity. If, instead, mitochondria were *only* important for regulating microglial production and secretion of trophic/inflammatory factors, there might be no correlation between the morphological complexity of an individual cell and the abundance of mitochondria in that cell. In **Reviewer Figure 6**, above, one can appreciate that a fairly amorphous Bone Marrow Derived Macrophage (BMDM) has tons of mitochondria. But it would be difficult to relate mitochondrial abundance to morphological *complexity* of BMDMs in any fashion. In this regard, the relationship between mitochondria and microglial morphological complexity

may be more akin to what is observed in neurons, where mitochondria are linked to dendritic spine morphology and remodeling (PMID 32714603).

→ **Elaboration of some of these points has been added to the discussion.**

The addition of methodological details, microglial metrics including branch points and process length, and the further discussion of these aspects satisfy our initial comments.

Minor comment: Please clarify in the method section whether for the Sholl radius 1 um has been used.

iv) The choice for looking at the NAc and VTA needs better clarification, also in perspective for the later analysis with LPS and aging. Are those brain regions particularly vulnerable?

→ **We have added text throughout the results section to explain in greater depth our rationale for focusing on these two brain regions. These additions include:**

“In fixed tissue from young adult (2mo) double transgenic *CX3CR1^{CreER/+};mitoGFP* mice (*MG-mitoGFP* mice hereafter), numerous GFP+ structures could be observed throughout the somas and cell processes of microglia in the nucleus accumbens (NAc) and ventral tegmental area (VTA), two interconnected brain regions where we have extensively mapped microglial phenotypes across the lifespan (**Fig. 1B** and **Fig. S1A,B**)”

“Microglia show prominent differences in cell density, branching complexity, and gene expression across the NAc and VTA³⁰. To determine if these regional differences in baseline microglial attributes were accompanied by regional differences in microglial mitochondria, we carried out high resolution confocal imaging and 3D reconstruction of both microglia and their mitochondria in the NAc and VTA of young adult mice (**Fig. 1B** and **Fig. S2A**).”

“We also showed previously that microglial responses to aging vary across brain region, with VTA microglia displaying proliferative and inflammatory changes months before microglia in the NAc. Whether there are links between mitochondrial state and early VTA microglial aging has also not been investigated.”

We appreciate the changes that improved the clarity of the manuscript.

v) Based on the figure legend description in Fig 1. It is understood that microglial volume is calculated per field of view (tissue coverage)- and mitochondrial number, volume, and mass are then calculated based on this FOV microglial volume. However, the text states, “greater number of mitochondria relative to cell volume, as well as a greater total mitochondrial volume relative to cell volume (mitochondrial mass)”. If the quantification is relative to total microglial volume per FOV, and not on a per-cell basis, stating ‘cell volume’ is confusing. For example, Fig. 1D-E, is the relative ‘microglia volume’ in 1D the same denominator as ‘cell volume’ in 1E? In addition, does the number of mitochondria within a FOV bring us important information? Since many microglial processes not connected to a cell body are present within a brain volume, these averages do not indicate information on a per-cell basis. The authors should identify precise nomenclature and analysis methods, describe this in detail in the methods, and use the terminology consistently throughout the manuscript. We performed analyses using a FOV approach to obtain information about microglia as a

cell population in that particular brain region. We feel that carrying out FOV analyses AND individual cell focused analyses is more rigorous than either approach alone. FOV analyses can smooth out the effects of cell-to-cell variation and remove concerns about any unintended bias in selecting individual cells for analysis.

→ **We have updated descriptions in the methods, figure legends and text to ensure consistency and avoid confusion about the distinction between FOV-based versus individual cell-based analyses.**

These additions satisfy our initial comments.

vi) The mitochondrial field focuses more on measuring network connectivity, i.e., fragmented or connected networks. This metric is not shown throughout the manuscript but would be an essential feature to describe when considering mitochondrial structural network changes.

The reviewer is correct that measures of “fragmentation” vs interconnected “networks” are commonly seen in the mitochondrial field. The vast majority of these studies are carried out in cultured cells where genetically-encoded mitochondrial reporters and/or mitochondrial dyes can easily be used with fantastic signal-to-noise, allowing automated reconstruction and analysis of mitochondria using machine learning platforms like Avia. In fixed tissue, however, even with optimal confocal imaging, distinguishing whether a cluster of mitochondria are fully, physically interconnected or separate is much more challenging. This is particularly true in the cell soma, where mitochondria are often very closely packed. For this reason, we feel it is more accurate to analyze mitochondrial elongation, mitochondrial sphericity, mitochondrial number, and mitochondrial mass which are also widely used in the mitochondrial field and to avoid classifications like “networked” or “fragmented,” which carry functional implications.

→ **To address the reviewer’s point, we have included new material in the discussion, commenting on the limitations of light microscopy for resolving whether mitochondria are truly networked or discrete, and additional future approaches that can be used to determine whether small mitochondria are have high ROS / low TMRM and are likely to be undergoing mitophagy (which is often assumed when employing the term fragmented) versus they have undergone fission to facilitate trafficking.**

These additions satisfy our initial comments.

Fig 2:

i) The conclusion of Fig. 2C needs to be clarified. If VTA and NAc show the same motility, how can the conclusion be made that the VTA is surveying less based on their morphology? We originally advanced this interpretation because, *in vivo*, VTA microglia are less dense and less branched than NAc microglia (Fig 1C,G,I and PMID: 28689984 and PMID: 35396327). Hence, if microglia in both brain regions show similar motility, it will take VTA microglia longer to execute a complete surveillance of the surrounding tissue than the more abundant, more highly branched NAc microglia.

→ **However, given that our observations of motility are made in an acute brain section preparation, we removed this statement to address any concerns that it is too speculative.**

This amendment satisfies our initial comments.

ii) Do the motile processes contain mitochondria? Or are motile mitochondria only found in primary processes that are more stable? (also related to Fig. 3). The reviewer brings up the very interesting question of how subcellular location relates to the motility of individual microglial cell processes and the motility of the mitochondria themselves.

→ To address this point, we have carried out additional analyses focusing on individual microglial cell processes (Fig. 2J). Similar to our findings at the level of the whole microglial cell, this analysis revealed that there was not a significant correlation between the abundance of mitochondria in a primary cell branch and the motility of that branch. There was also no significant correlation between the motility of mitochondria in a branch and the motility of the branch itself (or vice versa). Hence, it is not the case that microglial process branches that have more mitochondria exhibit any heightened capacity for cell process remodeling, at least over the time frame we are analyzing. We draw the reviewer's attention to the study by Pietramale et. al (BioRxiv 2024.10.15.618505) co-submitted with our study that suggest that this result may differ if focused on even smaller branchlets. By focusing on smaller branchlets (rather than entire arbors branching out from one primary process), they found that microglial branchlets with fewer mitochondria were more motile. There could also be important brain region differences in this mitochondria-branch motility relationship. **This will be an important area for the field to investigate further, and we elaborate on some of these considerations further in the discussion.**

We appreciate the consideration.

Minor comment: The outlined description fits into a discussion and the authors should cite the co-submitted study/biorxiv Pietramale, which investigates the question in more depth.

Fig 3:

i) This data is very interesting and raises exciting new questions; however, no quantification supports the concluding statement.

We are glad that the reviewer finds these data intriguing.

→ In this resubmission, we have included quantification of microglial process motility and mitochondrial motility following laser lesion (Fig. 3F-I). These analyses revealed that abundance of microglial processes close to the lesion increases over time, whereas abundance of microglial mitochondria close to the lesion does not increase, confirming our qualitative observations from the initial manuscript submission.

We appreciate the addition of quantification as it significantly strengthens the figure.

ii) At this resolution, it seems difficult to quantify and confirm that the mitochondrial networks remain stable after focal injury. Can the authors image at the resolution seen in Fig 2 and provide data analysis of mitochondrial network motility before and after injury? Do all mitochondria stop moving, or only a few travel within the extending process?

In this set of experiments, we elected to image a larger field of view in order to sample a larger number of microglia to determine how they and their mitochondrial networks

respond to laser lesion. While repeat experiments to image smaller fields of view at higher resolution could be carried out, we felt this would best be accomplished via *in vivo* multiphoton imaging that would also permit mitochondria to be monitored over a much longer time frame. Using such *in vivo* multiphoton approaches, Pietramale et. al (BioRxiv 2024.10.15.618505) found similar responses of microglia and their mitochondria in cortex, with negligible mitochondrial remodeling within 40min of the laser lesion. Intriguingly, they *did* observe mitochondrial remodeling at 6hrs following lesion.

We acknowledge the shortcomings, which should be outlined in the discussion and referenced to the Pietramale study.

Fig 4/5:

i) If mitochondria are not linked to microglial tissue surveillance and injury, as the authors conclude in Fig. 2J and Fig. 3, respectively, how do they rationalize using LPS? Based on live imaging, mitochondria are not linked in a “direct fashion to baseline surveillance” or injury-induced damage. Why would it suddenly be related to an inflammatory response induced by LPS? We are sorry to hear that the rationale for utilizing LPS to further interrogate the relationship between mitochondria and microglial responses to tissue challenge was not clearly communicated. Laser lesion is a focal tissue injury that elicits a rapid (within minutes) response from microglia. Purinergic signaling from the injured tissue to microglial P2RY12 receptors plays a critical role in this rapid injury response. Our data indicate that rapid extension of microglial processes is not accompanied by rapid remodeling of mitochondrial networks (at least on a morphological level). Given this result, to continue testing whether mitochondria play an important role in regulating microglial phenotypic remodeling in response to pathological challenges, it was desirable to examine microglial responses that occur over a longer time scale (hours to days) as well as microglial responses to a distinct modality of challenge (inflammatory challenge rather than tissue damage).

→ As described above, we have made changes to wording of the abstract, introduction, and beginning of each section in the results to better explain rationale for study design.

We appreciate the changes that improved the clarity of the manuscript.

ii) QPCR of *Dnml1* is used to indicate fission- however, this data should also be reflected in the image analysis in Fig 5. Could the authors indicate changes in the median volume of organelles (per cell) to compare alongside the mitochondrial number and aspect ratio? If the mitochondria are indeed elongated as indicated by an aspect ratio measure, the median volume of organelles per cell should increase (aligning with a mass increase) since the number of mitochondria did not change between saline and LPS for both brain regions.

The reviewer is correct that dynamin-related protein 1 (DRP1; encoded by the gene *Dnml1*) plays a key role in promoting mitochondrial fission. In both our FACS-based analyses in Fig. 4F and Fig. 8D, the prominent increase in GFP median fluorescence intensity suggests that mitochondrial biogenesis accompanies microglial responses to LPS challenge.

The increase in mitochondrial mass detected via histology (Fig. 5B), further indicates an overall increase in mitochondria within microglia following LPS challenge. The reviewer is correct that, in general, mean volume of individual mitochondria trended toward increases

at 24hrs following LPS (**Reviewer Figure 7**), particularly in the NAc, so the FACS-based increase in GFP and the increase in mitochondrial mass via histology is likely driven by larger, longer mitochondria.

These findings don't necessarily represent an inconsistency with the increase in *Dnml1* transcript found by qPCR. Indeed, *Dnml1* may be getting upregulated precisely because there are larger, longer mitochondria and increased fission will be needed to return the mitochondrial network to its pre-LPS state.

The image analysis data support the concept that there is an increase in mitochondrial mass, accompanied by more elongated mitochondria. The support of the reviewer figure 7 helps clarify this point. The finding that mitochondria are elongated after LPS challenge is in contrast to the reported effect of LPS on microglial mitochondria in the *in vitro* setting PMID: 30637805, which is an interesting point to highlight in the discussion.

iii) Do the authors have the details of the post-hoc analysis of 5B-D?

As described in the methods in the original submission, all one-way and two-way ANOVAs were performed with Bonferroni correction for multiple comparisons being applied to all post-hoc comparisons.

→ **We have added information about post-hoc corrections to the figure legends.**

We appreciate the addition.

iv) Technical aspect related to sampling for flow cytometry. The method section must clarify whether the samples were processed sequentially or in batches. Furthermore, information related to the total time from tissue removal to sorting must be included. It has been shown that the microglia transcriptional profile adapts within 30 minutes of incubation on ice (Marsh 2022. DOI: 10.1038/s41593-022-01022-8). This can result in misleading results. Therefore, the authors should add the animal grouping strategy and total processing time to the methods and, if necessary, repeat the experiments when the timing is longer than 30 min.

The authors are correct that microglia are extremely sensitive and adaptable cells that will respond to tissue dissociation. Hence, care needs to be taken in the tissue processing workflow to avoid artifacts. We are very familiar with the Marsh et. al paper to which the reviewer refers. In this study, the key finding is that exposing microglia to elevated temperature (37°C) to carry out enzymatic digestion causes them to exhibit altered transcriptional profiles. See Figure 1C of the paper. This panel shows that you can prevent aberrant microglial transcriptional signatures by keeping the entire workflow at 4°C (i.e. on ice) OR by including transcriptional inhibitors. For all experiments in our study, the entire workflow was carried out at 4 °C as described in the methods. In our prior studies (PMID: 28689984), RNAseq confirms that this workflow does not elicit upregulation of inflammatory factors typically associated with microglial reactivity. Moreover, this workflow does not induce increased expression of the specific genes highlighted in the Marsh et. al study (**Reviewer Figure 8**).

Of note, none of the workflows in Marsh et al are completed in under 30min. All of their tissue processing workflows take 70-130min as indicated by their workflow diagrams in Figures 1 and 2. 30 minutes refers to the portion of their workflow during which microglia were exposed to elevated temperature for enzymatic dissociation *OR* maintained at 4 degrees, with the latter treatment preserving *in vivo* gene expression signatures. Moreover, their stated workflow time does not include time for sorting, but rather indicates time prior to when sorting begins.

→ As requested by the reviewer, we have added the following information to the methods regarding our sample processing:

“Total sample processing time from brain dissection to completion of FACS purification ranged from 2-4hrs depending on sort order of individual samples. Samples were collected in batches with each day of sorting including a young adult, saline injected, or TFAM-wt control mouse. Sort order of samples from specific ages, brain regions, treatment conditions, or genotypes was interleaved to avoid introduction of bias related to tissue processing length. FACS-isolation itself was carried out at 4°C. FACS-isolation of all samples for a given set of experiments was completed prior to downstream processing to allow simultaneous RNA isolation and cDNA preparation as described below.”

We appreciate the addition.

v) How was the correlation analysis performed in Fig. 4D? Cherry-picked genes were connected to putative microglial states, an aspect that is also not well-defined, introducing a bias. Also, the explanation of the “few CD68 correlations with other genes” does not seem solid (Fig. S4B) since the heatmap shows much more correlation than any of the other plots in Fig. 4.

To test for potential correlations between microglial genes reflective of overall microglial functional state and our entire panel of 14 mitochondria-relevant genes, we carried out linear regression analyses for each microglial gene and all of the mitochondrial genes. For considerations of space, we included results for 6 / 9 microglial genes in the main figure and an additional microglial gene in Figure S4. We respectfully disagree that showing data from 7/9 microglial genes constitutes “cherry-picking” genes.

It also seems that we have not done an optimal job of explaining the significance of the *Cd68* result. Expression levels of *Cd68* are indeed correlated with expression levels of multiple other microglial genes (*Tgfβ1*, *Cx3cr1*, *P2ry12*, **Fig. S4D**). However, the important point here is that this degree of correlation was not modulated by LPS challenge (i.e. these significant correlations were present at baseline and remained present at both 4hrs and 24hrs post-LPS). Also importantly, expression levels of *Tnfα* and *Il1β* were correlated with expression levels of multiple mitochondrial genes, particularly at 4hrs post LPS. Yet, no correlation was observed between *Cd68* expression levels and those of *Tnfα* or *Il1β*. Together, these findings suggest a more robust relationship between the state of mitochondria (as organelles) and acute microglial responses to LPS compared to the relationship between lysosomes and acute microglial responses to LPS. (*Please note that in response to Reviewer 2’s comment, we re- executed our linear regression analyses using Robust regression with false discovery rate control; using this more rigorous approach, some

of the correlations that were found to be significant in the original submission are no longer significant in this revised analysis).

→ To address the reviewer's comment, we have modified the text of the results:

“Moreover, examination of *Cd68*, a key microglial lysosome gene, hints that mitochondria, as organelles, may be uniquely positioned to regulate microglial functional state during inflammatory challenge. While LPS treatment increased the correlation between most mitochondria-relevant genes and key microglial function genes (*Cd45*, *Cx3cr1*, *P2ry12*, *Tgfβ1*)(**Fig. 4D**), LPS treatment did not modulate correlations between *Cd68* and *Cx3cr1*, *P2ry12*, *Tgfβ1* (correlations between *Cd68* and *Cd45* increased at 24hrs post LPS)(**Fig. S4D** and **Table S4**). Moreover, while multiple mitochondria-relevant genes were positively correlated with expression of the microglial pro-inflammatory factors *Tnfα* and *Il1β* at 4hrs post-LPS, *Cd68* expression was not correlated with *Tnfα* and *Il1β* in any of the conditions examined (**Fig. S4D**).”

We appreciate the addition.

Fig 6:

Microglia are known to show a sex-specific response also in adulthood and, especially, during aging. Therefore, the authors must precisely annotate the data plots, indicating which data points come from male or female animals.

We agree that probing for sex-differences is important for the microglial field and we included both male and female mice in all our analyses.

→ To address the reviewer's comment, we have added a supplementary figure that replicates key findings from each of our analyses with the annotation of datapoints that come from male and female mice (**Fig. S9**) We have also added commentary to the discussion that future research should continue to investigate possible sex differences in microglial mitochondria.

We appreciate the addition, which sets an essential criterion for the transparency of the research data and will enable the community to acknowledge sex-specific differences to be follow up in the future, especially in neurodevelopmental and neurodegenerative diseases.

Minor point: For sampling with at least 3males/females, it would be appreciated to see which metrics did carry significant sex-specific differences.

Fig 7:

i) Related to the mitochondrial quantification in 7D: Is this data represented as an average volume of all organelles within the cell? If yes, it would be better to use a median, as small mitochondria can skew the average value and misrepresent the data. **Figure 7D** shows the sum total volume of all the mitochondria in a field of view (FOV). This value was obtained for 3 FOVs for each mouse to compute a mouse average, which is displayed in the graph.

→ We have adjusted the label of the y axis and the description of this panel in the figure legend to clearly specify that these are FOV measurements. We have also added FOV median mitochondrial volume and FOV maximum volume of an individual mitochondrion to **Figure S7F**. This highlights that increases in the total number of mitochondria within the

FOV are driving increases in total FOV mitochondrial volume, rather than increases in the size of individual mitochondria.

ii) Mitochondrial analysis: In Fig. 7, the analysis focused on volume, number, and sphericity, whereas earlier figures describing the mitochondria in fixed tissue focused on mass, number, aspect ratio, volume, sphericity. To allow comparison between the datasets, the authors should be consistent and include all metrics for all figures. Furthermore, the methods section needs to include a detailed description of the analysis method for each parameter.

→ We have added analysis of mitochondrial mass to this figure (Fig. 7K). We have expanded the description in the methods of how these metrics were generated within Imaris analysis software.

This amendment satisfies our initial comments.

Fig 8:

How do the microglia change after TFAM KO? A two-month waiting period may allow time for compensatory mechanisms to circumvent initial damage. Understanding how TFAM KO impacts microglia is indeed the primary goal of these experiments. → In addition to the histology in the original submission (previous Fig 7I, now Fig 7H) which showed that microglial tissue coverage within the field of view (FOV) increases after microglial TFAM KO, we now show that morphological complexity of individual microglia (analyzed via Sholl analysis) increases in MG-TFAM KO mice (Fig. 7I).

→ We also show that microglial density is unchanged (Fig. 7G).

→ In addition to the qPCR in the original submission that showed multiple changes in microglial gene expression in MG-TFAM KO mice (previous Fig. 8C, now Fig. 8B), we now also show that protein level expression of CD45 and CX3CR1 (MFI during FACS) are unchanged in MG-TFAM KO microglia. Cell size (FSC) and internal complexity (SSC) are increased in microglia from MG-TFAM KO mice (Fig. 8A).

→ Finally, in this resubmission, we challenge MG-TFAM KO mice with LPS and utilize FACS and qPCR to assess the capacity of the cells to mount a mitochondrial and cellular response to this challenge (see detailed description in response to reviewer #1, point 4).

As described in the results, we elected to wait for 2 months following tamoxifen injection for the following reasons. 1) This will allow for any peripheral monocytes/macrophages impacted by this manipulation to cycle out and be replaced. 2) Mitochondrial proteins are among the most long-lived proteins in the CNS (PMID: 30315172). Hence, it is estimated that approximately 1 month would be needed for existing mitochondrial proteins to cycle out. Moreover, TFAM exerts its broad effects on the mitochondria by altering stability of the mitochondrial DNA. This impacts capacity of the mitochondria to produce key components of the electron transport chain, with additional downstream consequences for other aspects of mitochondrial function. In line with this, virtually all studies that have used cell-specific TFAM KO have either used constitutive Cre lines (PMID 31269437, PMID 17227870, PMID 9916807, PMID 25005176, PMID 23168219, PMID 18945718), or waited at least a month (PMID 35165165, PMID 40210116) . The reviewer is correct that some compensatory

responses may be elicited and some of the changes we observe in expression of nuclear-encoded mitochondrial components may represent attempts at compensation. Nonetheless, we feel that our choice of 2 months is well justified, and it will fall to future studies to tease apart direct vs compensatory effects on the organelle.

Discussion:

In line with the loose introduction, the discussion is also only superficial. It mostly repeats the results without putting the research data into the context of the literature. For example, a reported result is increased mitochondrial mass. What does increased mitochondrial mass imply? The manuscript would enormously benefit from a complete revision of the introduction and discussion to make it coherent.

We are sorry to hear that the reviewer did not find the discussion informative.

→ **We have updated the discussion to include some potential implications of key observations based on what is known in the literature from mitochondrial behavior in other cell types.** Multiple follow-up studies will be required to fully elucidate what a particular mitochondrial attribute means for microglial function in a given context. Our aim with this study was to inspire those types of follow-up studies by revealing how closely linked the general state of the organelle is with the overall state of the cell.

Overstatements:

The manuscript suffers from several overstatements and preliminary conclusions, which must be fixed. A few examples are listed below:

i) “First time” in the introduction. **We have removed this type of wording from the manuscript.**

ii) The authors regularly write “in vivo” within their manuscript, yet it becomes quickly apparent that the majority of the experiments are performed in a fixed tissue environment. This should be clarified in the text to avoid misunderstandings. **We have adjusted wording in the manuscript to avoid confusion as the reviewer suggests.**

Overall, the revised manuscript is significantly improved in clarity, making it accessible to a broader readership. It offers insights into mitochondria in microglia across a range of physiological, pathological, and aging conditions, providing the foundation for many future studies. Our specific concerns have been adequately addressed, with only a few minor points remaining.